# RelaySum for Decentralized Deep Learning on Heterogeneous Data

**Thijs Vogels**[*]
EPFL

**Lie He**[*]
EPFL

**Anastasia Koloskova**
EPFL

**Tao Lin**
EPFL

**Sai Praneeth Karimireddy**
EPFL

**Sebastian U. Stich**
EPFL

**Martin Jaggi**
EPFL

## Abstract

In decentralized machine learning, workers compute model updates on their local data. Because the workers only communicate with few neighbors without central coordination, these updates propagate progressively over the network. This paradigm enables distributed training on networks without all-to-all connectivity, helping to protect data privacy as well as to reduce the communication cost of distributed training in data centers. A key challenge, primarily in decentralized deep learning, remains the handling of differences between the workers' local data distributions. To tackle this challenge, we study the RelaySum mechanism for information propagation in decentralized learning. RelaySum uses spanning trees to distribute information exactly uniformly across all workers with finite delays depending on the distance between nodes. In contrast, the typical gossip averaging mechanism only distributes data uniformly asymptotically while using the same communication volume per step as RelaySum. We prove that RelaySGD, based on this mechanism, is independent of data heterogeneity and scales to many workers, enabling highly accurate decentralized deep learning on heterogeneous data. Our code is available at `http://github.com/epfml/relaysgd`.

## 1  Introduction

Ever-growing datasets lay at the foundation of the recent breakthroughs in machine learning. Learning algorithms therefore must be able to leverage data distributed over multiple devices, in particular for reasons of efficiency and data privacy. There are various paradigms for distributed learning, and they differ mainly in how the devices collaborate in communicating model updates with each other. In the *all-reduce* paradigm, workers average model updates with all other workers at every training step. In *federated learning* [24], workers perform local updates before sending them to a central server that returns their global average to the workers. Finally, *decentralized learning* significantly generalizes the two previous scenarios. Here, workers communicate their updates with only few directly-connected neighbors in a network, without the help of a server.

Decentralized learning offers strong promise for new applications, allowing any group of agents to collaboratively train a model while respecting the data locality and privacy of each contributor [25]. At the same time, it removes the single point of failure in centralized systems such as in federated learning [12], improving robustness, security, and privacy. Even from a pure efficiency standpoint, decentralized communication patterns can speed up training in data centers [2].

In decentralized learning, workers share their local stochastic gradient updates with the others through *gossip* communication [41]. They send their updates to their neighbors, which iteratively

---

[*]Equal contribution. Corresponding authors `thijs.vogels@epfl.ch` and `lie.he@epfl.ch`.

35th Conference on Neural Information Processing Systems (NeurIPS 2021).

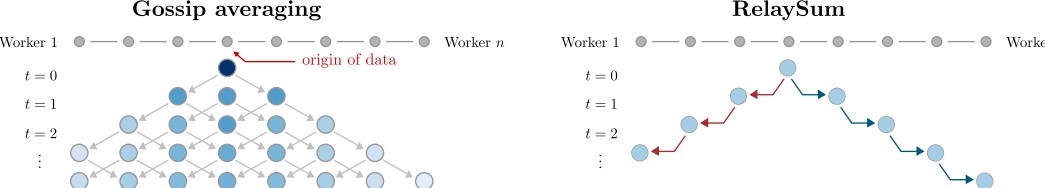

Figure 1: To spread information across a decentralized network, classical gossip averaging diffuses information slowly through the network. The left figure illustrates the spread of information originating from the fourth worker in a chain network. In RelaySum, the messages are *relayed* without reweighting, resulting in uniform delivery of the information to every worker. When multiple workers broadcast simultaneously (not pictured), RelaySum can *sum* their messages and use the same bandwidth as gossip averaging.

propagate the updates further into the network. The workers typically use iterative *gossip averaging* of their models with their neighbors, using averaging weights chosen to ensure asymptotic uniform distribution of each update across the network. It will take $\tau$ rounds of communication for an update from worker $i$ to reach a worker $j$ that is $\tau$ hops away, and when it first arrives, the update is exponentially weakened by repeated averaging with weights $< 1$. In general networks, worker $j$ will never exactly, but only asymptotically receive its uniform share of the update. The slow distribution of updates not only slows down training, but also makes decentralized learning sensitive to heterogeneity in workers' data distributions.

We study an alternative mechanism to gossip averaging, which we call RelaySum. RelaySum operates on spanning trees of the network, and distributes information exactly uniformly within a finite number of gossip steps equal to the diameter of the network. Rather than iteratively averaging models, each node acts as a 'router' that *relays* messages through the whole network without decaying their weight at every hop. While naive all-to-all routing requires $n^2$ messages to be transmitted at each step, we show that on trees, only $n$ messages (one per edge) are sufficient. This is enabled by the key observation that the routers can *merge* messages by *summation* to avoid any extra communication compared to gossip averaging. RelaySum achieves this using additional memory linear in the number of edges, and by tailoring the messages sent to different neighbors. At each time step, RelaySum workers receive a uniform average of exactly one message from each worker. Those messages just originate from different time delays depending on how many hops they travelled. The difference between gossip averaging and RelaySum is illustrated in Figure 1.

The RelaySum mechanism is structurally similar to Belief Propagation algorithms for inference in graphical models. This link was made by Zhang et al. [50], who used the same mechanism for decentralized weighted average consensus in control.

We use RelaySum in the RelaySGD learning algorithm. We theoretically show that this algorithm is not affected by differences in workers' data distributions. Compared to other algorithms that have this property [36, 31], RelaySGD does not require the selection of averaging weights, and its convergence does not depend on the spectral gap of the averaging matrix, but instead on the network diameter.

While RelaySum is formulated for trees, it can be used in any decentralized network. We use the Spanning Tree Protocol [30] to construct spanning trees of any network in a decentralized fashion. RelaySGD often performs better on any such spanning tree than gossip-based methods on the original graph. When the communication network can be chosen freely, the algorithm can use double binary trees [33]. While these trees have logarithmic diameter and scale to many workers, RelaySGD in this setup uses only constant memory equivalent to two extra copies of the model parameters and sends and receives only two models per iteration.

Surprisingly, in deep learning with highly heterogeneous data, prior methods that are theoretically independent of data heterogeneity [36, 31], perform worse than heuristic methods that do not have this property, but use cleverly designed time-varying communication topologies [2]. In extensive tests on image- and text classification, RelaySGD performs better than both kinds of baselines at equal communication budget.

## 2 Related work

Out of the multitude of decentralized optimization methods, first-order algorithms that interleave local gradient updates with a form of gossip averaging [29, 11] show most promise for deep learning. Such algorithms are theoretically analyzed for convex and non-convex objectives in [28, 11, 29], and [19, 36, 2, 20] demonstrate that gossip-based methods can perform well in deep learning.

In a gossip averaging step, workers average their local models with the models of their direct neighbors. The corresponding 'mixing matrix' is a central object of study. The matrix can be doubly-stochastic [29, 19, 16], column-stochastic [38, 26, 39, 2], row-stochastic [40, 44], or a combination [42, 43, 32]. Column-stochastic methods use the *push-sum* consensus mechanism [13] and can be used on directed graphs. Our analysis borrows from the theory developed for those methods.

While gossip averages in general requires an infinite number of steps to reach exact consensus, another line of work identifies mixing schemes that yield exact consensus in finite steps. For some graphs, this is possible with time-independent averaging weights [15, 6]. One can also achieve finite-time consensus with time-varying mixing matrices. On trees, for instance, exact consensus can be achieved by routing updates to a root node and back, in exactly diameter number of steps [15, 6]. On some graphs, tighter bounds can be established [8]. For fully-connected networks with $n$ workers, Assran et al. [2] design a sparse time-varying communication scheme that yields exact consensus in a cycle of $\log n$ averaging steps and performs well in deep learning.

The 'relay' mechanism of RelaySGD was previously used by Zhang et al. [50] in the control community for the decentralized weighted average consensus problem, but they do not use it in the context of optimization. Zhang et al. also introduce a modified algorithm for loopy graphs, but this modification makes the achieved consensus inexact. The 'relay' mechanism effectively turns a sparse graph into a fully-connected graph with communication delays. Work on delayed consensus [27] and optimization [37, 1] analyzes such schemes for centralized distributed algorithms. Those consensus schemes are, however, not directly applicable to decentralized optimization.

A fundamental challenge in decentralized learning is dealing with data that is not identically distributed among workers. Because, in this case, workers pursue different optima, workers may drift [29] and this can harm convergence. There is a large family of algorithms that introduce update corrections that provably mitigate such data heterogeneity. Examples applicable to non-convex problems are exact diffusion [45], Gradient Tracking [22, 31, 48], D$^2$ [36], PushPull [32]. To tackle the same challenge, Lin et al. [20], Yuan et al. [46] propose modifications to local momentum to empirically improve performance in deep learning, but without provable guarantees. Lu and De Sa [23] propose DeTAG which overlaps multiple consecutive gossip steps and gradient computations to accelerate information diffusion. This technique could be applied to the RelaySum mechanism, too.

## 3 Method

**Setup**   We consider standard decentralized optimization with data distributed over $n \geq 1$ nodes:

$$f^\star := \min_{\mathbf{x} \in \mathbb{R}^d} \left[ f(\mathbf{x}) = \frac{1}{n} \sum_{i=1}^n \left[ f_i(\mathbf{x}) := \mathbb{E}_{\xi \sim \mathcal{D}_i} F_i(\mathbf{x}, \xi_i) \right] \right] .$$

Here $\mathcal{D}_i$ denotes the distribution of the data on node $i$ and $f_i \colon \mathbb{R}^d \to \mathbb{R}$ the local optimization objectives. Workers are connected by a network respecting a graph topology $\mathcal{G} = (\mathcal{V}, \mathcal{E})$, where $\mathcal{V} = \{1, \dots, n\}$ denotes the set of workers, and $\mathcal{E}$ the set of undirected communication links between them (without self loops). Each worker $i$ can only directly communicate with its neighbors $\mathcal{N}_i \subset \mathcal{V}$.

**Decentralized learning with gossip**   We consider synchronous first-order algorithms that interleave local gradient-based updates

$$\mathbf{x}_i^{(t+1/2)} = \mathbf{x}_i^{(t)} + \mathbf{u}_i^{(t)}$$

with message exchange between connected workers. For SGD with typical gossip averaging (DP-SGD [19]), the local updates can be written as $\mathbf{u}_i^{(t)} = -\gamma \nabla f_i(\mathbf{x}_i^{(t)}, \xi_i^{(t)})$, and the messages exchanged between pairs of connected workers $(i, j)$ are $\mathbf{m}_{i \to j}^{(t)} = \mathbf{x}_i^{(t+1/2)} \in \mathbb{R}^d$. Each timestep, the workers average their model with received messages,

$$\mathbf{x}_i^{(t+1)} = \mathbf{W}_{ii} \mathbf{x}_i^{(t+1/2)} + \sum_{j \in \mathcal{N}_i} \mathbf{W}_{ij} \mathbf{m}_{j \to i}^{(t)}, \tag{DP-SGD}$$

using averaging weights defined by a *gossip matrix* $\mathbf{W} \in \mathbb{R}^{n \times n}$.

In this scheme, an update $\mathbf{u}_i^{(t_1)}$ from any worker $i$ will be linearly incorporated into the model $\mathbf{x}_j^{(t_2)}$ at a later timestep $t_2$ with weight $(\mathbf{W}^{t_2 - t_1})_{ij}$. The gossip matrix must be chosen such that these weights asymptotically converge to $\frac{1}{n}$, distributing all updates uniformly over the workers. This setup appears in, for example, [19, 16].

**Uniform model averaging**    If the graph topology is fully-connected, any worker can communicate with any other worker, and it is ideal to use 'all-reduce averaging',

$$\mathbf{x}_i^{(t+1)} = \tfrac{1}{n} \sum_{j=1}^n \mathbf{x}_j^{(t+1/2)}.$$

Contrary to the decentralized scheme (DP-SGD), this algorithm does not degrade in performance if data is distributed heterogeneously across workers. In sparsely connected networks, however, all-reduce averaging requires routing messages through the network. On arbitrary networks, such a routing protocol requires at least a number of communication steps equal to the network diameter $\tau_{\max}$—the minimum number of hops some messages have to travel.

**RelaySGD**    In this paper, we approximate the all-reduce averaging update as

$$\mathbf{x}_i^{(t+1)} = \tfrac{1}{n} \sum_{j=1}^n \mathbf{x}_j^{(t - \tau_{ij} + 1/2)}, \tag{RelaySGD}$$

where $\tau_{ij}$ is minimum number of network hops between workers $i$ and $j$ (and $\tau_{ii} = 0$). Since it takes $\tau_{ij}$ steps to route a message from worker $i$ to $j$, this scheme could be implemented using a peer-to-peer routing protocol like Ethernet. Of course, this naive implementation drastically increases the bandwidth used compared to gossip averaging. The key insight of this paper is that, on tree networks, the RelaySGD update rule can be implemented while using the same communication volume per step as gossip averaging, using additional memory linear in the number of a worker's direct neighbors.

**RelaySum**    To implement RelaySGD, we require a communication mechanism that delivers sums of delayed 'parcels' $s_w^{(t)} = \sum_{j=1}^n p_j^{(t - \tau_{wj})}$ to each worker $w$ in a tree network, where the parcel $p_j^{(t)}$ is created by worker $j$ at time $t$. To simplify the exposition, let us first consider the simplest type of tree network: a chain. In a chain, a worker $w$ is connected to workers $w - 1$ and $w + 1$, if those exist, and the delays are $\tau_{ij} = |i - j|$. We can then decompose

$$s_w^{(t)} = \sum_{j=1}^n p_j^{(t - \tau_{wj})} = p_w^{(t)} + \underbrace{\sum_{j=1}^{w-1} p_j^{(t - \tau_{wj})}}_{\text{parcels from the 'left'}} + \underbrace{\sum_{j=w+1}^n p_j^{(t - \tau_{wj})}}_{\text{parcels from the 'right'}}.$$

The sum of parcels from the 'left' will be sent as one message $m_{(w-1) \to w}$ from worker $w - 1$ to $w$, and the sum of data from the 'right' will be sent as one message $m_{(w+1) \to w}$ from $w + 1$ to $w$. Neighboring workers can compute these messages from the messages they received from their neighbors in the previous timestep. Compared to typical gossip averaging, RelaySum requires additional memory linear in the number of neighbors, but it uses the same volume of communication.

Algorithm 1 shows how this scheme is generalized to general tree networks and incorporated into RelaySGD. Along with the model parameters, we send scalar counters that are used in the first few iterations of the algorithm $t \le \tau_{\max}$ to correct for messages that have not yet arrived.

**Spanning trees**    RelaySGD is formulated on tree networks, but it can be used on any communication graph by constructing a spanning tree. In a truly decentralized setting, we can use the Spanning Tree Protocol [30] used in Ethernet to find such trees in a decentralized fashion. The protocol elects a leader as the root of the tree, after which every other node finds the fastest path to this leader.

On the other hand, when the decentralized paradigm is used in a data center to reduce communication, RelaySGD can run on double binary trees [33] used in MPI and NCCL [10]. The key idea of double binary trees is to use two different communication topologies for different parts of the model. We communicate odd coordinates using a balanced binary tree $A$, and communicate the even coordinates with a complimentary tree $B$. The trees $A$ and $B$ are chosen such that internal nodes (with 3 edges)

**Algorithm 1** RelaySGD

**Input:** $\forall i$, $\mathbf{x}_i^{(0)} = \mathbf{x}^{(0)}$; $\forall i, j, \mathbf{m}_{i \to j}^{(-1)} = \mathbf{0}$, counts $c_{i \to j}^{(-1)} = 0$, learning rate $\gamma$, tree network
1: **for** $t = 0, 1, \ldots$ **do**
2:     **for** node $i$ **in parallel**
3:         $\mathbf{x}_i^{(t+1/2)} = \mathbf{x}_i^{(t)} - \gamma \nabla f_i(\mathbf{x}_i^{(t)})$                                   (or Adam/momentum)
4:         **for** each neighbor $j \in \mathcal{N}_i$ **do**
5:             Send $\mathbf{m}_{i \to j}^{(t)} = x_i^{(t+1/2)} + \sum_{k \in \mathcal{N}_i \backslash j} \mathbf{m}_{k \to i}^{(t-1)}$    (relay messages from other neighbors)
6:             Send corresponding counters $c_{i \to j}^{(t)} = 1 + \sum_{k \in \mathcal{N}_i \backslash j} c_{k \to i}^{(t-1)}$
7:             Receive $(\mathbf{m}_{j \to i}^{(t)}, c_{j \to i}^{(t)})$ from node $j$
8:         **end for**
9:         $\bar{n}_i^{(t+1)} = 1 + \sum_{j \in \mathcal{N}_i} c_{j \to i}^{(t)}$                        ($\bar{n}$ converges to the total number of workers)
10:        $\mathbf{x}_i^{t+1} = \frac{1}{\bar{n}_i^{(t+1)}} \left( \mathbf{x}_i^{(t+1/2)} + \sum_{j \in \mathcal{N}_i} \mathbf{m}_{j \to i}^{(t)} \right)$        $\left( = \frac{1}{n} \sum_{j=1}^{n} \mathbf{x}_j^{(t - \tau_{ij} + 1/2)} \right)$
11:     **end for**
12: **end for**

in one tree are leaves (with only 1 edge) in the other. Using the combination of two trees, RelaySGD requires only constant extra memory equivalent to at most 2 model copies (just like the Adam optimizer [14]), and it sends and receives the equivalent of 2 models (just like on a ring).

## 4 Theoretical analysis

Since RelaySGD updates worker's models at time step $t+1$ using models from (at most) the past $\tau_{\max}$ steps, we conveniently reformulate RelaySGD in the following way: Let $\mathbf{Y}^{(t)}, \mathbf{G}^{(t)} \in \mathbb{R}^{n(\tau_{\max}+1) \times d}$ denote stacked worker models and gradients whose row vectors at index $n \cdot \tau + i$ represent

$$\left[ \mathbf{Y}^{(t)} \right]_{n\tau+i}^{\top} = \begin{cases} \mathbf{x}_i^{(t-\tau)} & t \geq \tau \\ \mathbf{x}^{(0)} & \text{otherwise} \end{cases}, \qquad \left[ \mathbf{G}^{(t)} \right]_{n\tau+i}^{\top} = \begin{cases} \nabla F_i(\mathbf{x}_i^{(t-\tau)}; \xi_i^{(t-\tau)}) & t \geq \tau \\ \mathbf{x}^{(0)} & \text{otherwise} \end{cases}$$

for all times $t \geq 0$, delay $\tau \in [0, \tau_{\max}]$ and worker $i \in [n]$. Then (RelaySGD) can be written as

$$\mathbf{Y}^{(t+1)} = \mathbf{W} \mathbf{Y}^{(t)} - \gamma \tilde{\mathbf{W}} \mathbf{G}^{(t)}$$

where $\mathbf{W}, \tilde{\mathbf{W}} \in \mathbb{R}^{n(\tau_{\max}+1) \times n(\tau_{\max}+1)}$ are non-negative matrices whose elements are

$$[\mathbf{W}]_{n\tau+i, n\tau'+j} = \begin{cases} \frac{1}{n} & \tau = 0 \text{ and } \tau' = \tau_{ij} \\ 1 & i = j \text{ and } \tau = \tau' + 1 \\ 0 & \text{otherwise} \end{cases}, \qquad \left[ \tilde{\mathbf{W}} \right]_{n\tau+i, n\tau'+j} = \begin{cases} \frac{1}{n} & \tau = 0 \text{ and } \tau' = \tau_{ij} \\ 0 & \text{otherwise} \end{cases}$$

for all $\tau, \tau' \in [0, \tau_{\max}]$ and $i, j \in [n]$. The matrix $\mathbf{W}$ can be interpreted as the mixing matrix of an 'augmented graph' [27] with additional virtual 'forwarding nodes'. $\mathbf{W}$ is row stochastic and its largest eigenvalue is 1. The vector of all ones $\mathbf{1}_{n(\tau_{\max}+1)} \in \mathbb{R}^{n(\tau_{\max}+1)}$ is a right eigenvector of $\mathbf{W}$ and let $\boldsymbol{\pi} \in \mathbb{R}^{n(\tau_{\max}+1)}$ be the left eigenvector such that $\boldsymbol{\pi}^{\top} \mathbf{1}_{n(\tau_{\max}+1)} = 1$.

We characterize the convergence rate of the consensus distance in the following key lemma:

**Lemma 1** (Key lemma). *There exists an integer $m = m(\mathbf{W}) > 0$ such that for any $\mathbf{X} \in \mathbb{R}^{n(\tau_{\max}+1) \times d}$ we have*

$$\| \mathbf{W}^m \mathbf{X} - \mathbf{1} \boldsymbol{\pi}^{\top} \mathbf{X} \|^2 \leq (1-p)^{2m} \| \mathbf{X} - \mathbf{1} \boldsymbol{\pi}^{\top} \mathbf{X} \|^2,$$

*where $p = \frac{1}{2}(1 - |\lambda_2(\mathbf{W})|)$ is a constant.*

All the following optimization convergence results will only depend on the *effective spectral gap* $\rho := \frac{p}{m}$ of $\mathbf{W}$. We empirically observe that $\rho = \Theta(1/n)$ for a variety of network topologies (see Figure 5 in Appendix A).

**Remark 2.** *The above key lemma is similar to [16, Assumption 4] for gossip-type averaging with symmetric matrices. However, in our case $\mathbf{W}$ is just a row stochastic matrix, and its spectral norm*

$\|\mathbf{W}\|_2 > 1$. *In general, the consensus distance can increase after just one single communication step (multiplication by* $\mathbf{W}$*). That is why we need* $m > 1$*. The proof of the Lemma relies on a Perron-Frobenius type theorem, and holds over several steps* $m$ *instead of a single iteration. It means RelaySum defines a consensus algorithm with linear convergence rate which pulls models closer.*

Our main convergence results hold under the following common assumptions, as e.g. [16].

**Assumption A** (L-smoothness). *For each* $i \in [n]$*,* $F_i(\mathbf{x}, \xi) : \mathbb{R}^D \times \Omega_i \to \mathbb{R}$ *is differentiable for each* $\xi \in supp(\mathcal{D}_i)$ *and there exists a constant* $L \geq 0$ *such that for each* $\mathbf{x}, \mathbf{y} \in \mathbb{R}^d$*,* $\xi \in supp(\mathcal{D}_i)$*:*

$$\|\nabla F_i(\mathbf{x}, \xi) - \nabla F_i(\mathbf{y}, \xi)\| \leq L\|\mathbf{x} - \mathbf{y}\|.$$

**Assumption B** (Uniform bounded noise). *There exists constant* $\bar{\sigma}$*, such that for all* $\mathbf{x} \in \mathbb{R}^d$*,* $i \in [n]$*,*

$$\mathbb{E}_\xi \|\nabla F_i(\mathbf{x}, \xi) - \nabla f_i(\mathbf{x})\|^2 \leq \bar{\sigma}^2.$$

**Assumption C** ($\mu$-convexity). *For* $i \in [n]$*, each function* $f_i : \mathbb{R}^d \to \mathbb{R}$ *is* $\mu$*-(strongly) convex for constant* $\mu \geq 0$*. That is,* $\forall \, \mathbf{x}, \mathbf{y} \in \mathbb{R}^d$

$$f_i(\mathbf{x}) - f_j(\mathbf{y}) + \frac{\mu}{2}\|\mathbf{x} - \mathbf{y}\|_2^2 \leq \nabla f_i(\mathbf{x})^\top (\mathbf{x} - \mathbf{y}).$$

**Theorem I** (RelaySGD). *For any target accuracy* $\varepsilon > 0$ *and an optimal solution* $\mathbf{x}^\star$*,*
**(Convex:)** *under Assumptions A, B and C with* $\mu \geq 0$*, it holds that* $\frac{1}{T+1}\sum_{t=0}^{T}\left(f(\overline{\mathbf{x}}^{(t)}) - f(\mathbf{x}^\star)\right) \leq \varepsilon$
*after*

$$\mathcal{O}\left(\frac{\bar{\sigma}^2}{n\varepsilon^2} + \frac{C\sqrt{L}\bar{\sigma}}{\varepsilon^{3/2}} + \frac{CL}{\varepsilon}\right) R_0^2$$

*iterations. Here* $\overline{\mathbf{x}}^{(t)} := \boldsymbol{\pi}^\top \mathbf{Y}^{(t)}$ *averages past models,* $R_0^2 = \|\mathbf{x}^0 - \mathbf{x}^\star\|^2$*, and* $C = \mathcal{O}(\frac{1}{\rho}\tau_{\max}^{3/2})$*.*
**(Non-convex:)** *under Assumptions A and B, it holds that* $\frac{1}{T+1}\sum_{t=0}^{T}\|\nabla f(\overline{\mathbf{x}}^{(t)})\|^2 \leq \varepsilon$ *after*

$$\mathcal{O}\left(\frac{\bar{\sigma}^2}{n\varepsilon^2} + \frac{C\bar{\sigma}}{\varepsilon^{3/2}} + \frac{C}{\varepsilon}\right) LF_0$$

*iterations, where* $F_0 := f(\overline{\mathbf{x}}^{(0)}) - f(\mathbf{x}^\star)$*.*

The dominant term in our convergence result, $\mathcal{O}\left(\frac{\bar{\sigma}^2}{n\varepsilon^2}\right)$ matches with the dominant term in the convergence rate of centralized ('all-reduce') mini-batch SGD, and thus can not be improved.

In contrast to other methods, the presented convergence result of RelaySGD is independent of the data heterogeneity $\zeta^2$ in [16, Assumption 3b].

**Definition D** (Data heterogeneity). *There exists a constant* $\zeta^2$ *such that* $\forall \, i \in [n], \mathbf{x} \in \mathbb{R}^d$

$$\|\nabla f_i(\mathbf{x}) - \nabla f(\mathbf{x})\|_2^2 \leq \zeta^2.$$

**Remark 3.** *For convex objectives, Assumptions B and D can be relaxed to only hold at the optimum* $\mathbf{x}^\star$*. A weaker variant of Assumption A only uses L-smoothness of* $f_i$ *[16, Assumption 1b].*

Comparing to gossip averaging for convex $f_i$ which has complexity $\mathcal{O}(\frac{\bar{\sigma}^2}{n\varepsilon^2} + (\frac{\zeta}{\rho} + \frac{\bar{\sigma}}{\sqrt{\rho}})\frac{\sqrt{L}}{\varepsilon^{3/2}} + \frac{L}{\rho\varepsilon})R_0^2$, our rate for RelaySGD does not depend on $\zeta^2$ and has same leading term $\mathcal{O}(\frac{\bar{\sigma}}{n\varepsilon^2})$ as $D^2$.

## 5 Experimental analysis and practical properties

### 5.1 Effect of network topology

**Random quadratics** To efficiently investigate the scalability of RelaySGD with respect to the number of workers, and to study the benefits of binary tree topologies over chains, we introduce a family of synthetic functions. We study *random quadratics* with local cost functions $f_i(\mathbf{x}) = \|\mathbf{A}_i\mathbf{x} - \mathbf{b}_i^\top\mathbf{x}\|^2$ to precisely control all constants that appear in our theoretical analysis. The Hessians $\mathbf{A}_i$ are initialized randomly, and their spectrum is scaled to achieve a desired smoothness $L$ and strong convexity $\mu$. The offsets $\mathbf{b}_i$ ensure a desired level of heterogeneity $\zeta^2$ and distance between optimum and initialization $r_0$. Appendix B.4 describes the generation of these quadratics in detail.

**Scalability on rings and trees** Using these quadratics, Figure 2 studies the number of steps required to reach a suboptimality $f(\bar{\mathbf{x}}) - f(\mathbf{x}^\star) \leq \varepsilon$ with tuned constant learning rates. On *ring* topologies with uniform (1/3) gossip weights (and chains for RelaySum), all compared methods require steps at least linear in the number of workers to reach the target quality. RelaySGD and $D^2$ empirically scale significantly better than Gradient Tracking, these methods are all independent of data heterogeneity. On a *balanced binary tree network* with Metropolis-Hastings weights [41], both $D^2$ and Gradient Tracking notably do not scale better than on a ring, while RelaySGD on these trees requires only a number of steps logarithmic in the number of workers. SGP with their time-varying exponential topology scales well, too, but it requires more steps on more heterogeneously distributed data.

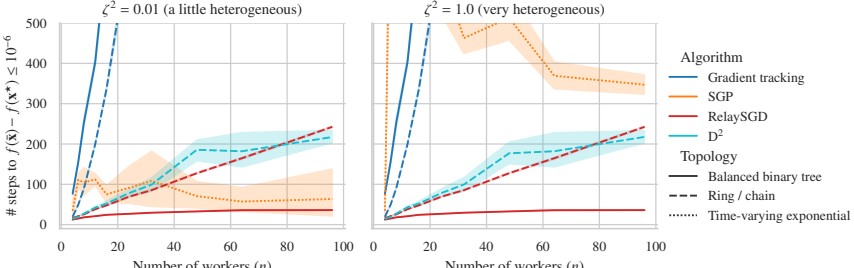

Figure 2: Time required to optimize random quadratics ($\sigma^2 = 0, r_0 = 10, L = 1, \mu = 0.5$) to suboptimality $\leq 10^{-6}$ with varying numbers of workers with tuned constant learning rates. On a ring (- - -), ■ $D^2$ and ■ RelaySGD require steps linear in the number of workers, and this number is *independent of the data heterogeneity*. RelaySGD reduces this to $\log n$ on a balanced tree topology (——), but trees do not improve ■ $D^2$ or ■ Gradient Tracking. For ■ SGP with time-varying exponential topology (·····), the number of steps does not consistently grow with more workers, but this number becomes higher with more heterogeneity (left v.s. right plot).

## 5.2 Spanning trees compared to other topologies

RelaySGD cannot utilize all available edges in arbitrary networks to communicate, but is restricted to a spanning tree of the graph. We empirically find that this restriction is not limiting. In Figure 3, we take an organic social network topology based on the Davis Southern Women graph [4] from NetworkX [7], and construct random spanning trees found by the Spanning Tree Protocol [30]. On any such spanning tree, RelaySGD optimizes random heterogeneous quadratics as fast as $D^2$ on the full graph with Metropolis-Hastings weights [41], significantly faster than DP-SGD.

For decentralized learning used in a fully-connected data center for communication efficiency, the deep learning experiments below show that RelaySGD on double binary trees outperforms the most popular non-tree-based communication scheme used in decentralized deep learning [2].

## 5.3 Effect of data heterogeneity in decentralized deep learning

We study the performance of RelaySGD in deep-learning based image- and text classification. While the algorithm is theoretically independent of dissimilarities in training data, other methods ($D^2$, RelaySGD/Grad) that have the same property often lose accuracy in the presence of high data heterogeneity [20]. To study the dependence of RelaySGD in practical deep learning, we partition training data strictly across 16 workers and distribute the classes using a Dirichlet process [47, 20]. The Dirichlet parameter $\alpha$ controls the heterogeneity of the data across workers.

We compare RelaySGD against a variety of other algorithms. DP-SGD [19] is the most natural combination of SGD with gossip averaging, and we chose $D^2$ [36] to represent the class of previous work that is theoretically robust to heterogeneity. We extend $D^2$ to allow varying step sizes and local momentum, according to Appendix D.4, and make it suitable for practical deep learning. Although Stochastic Gradient Push [2] is not theoretically independent of data heterogeneity, it is a popular choice in the data center setting, where they use a time-varying exponential scheme on $2^d$ workers that mixes exactly uniformly in $d$ rounds (Appendix D.6). We also compare to DP-SGD with quasi-global momentum [20], a practical method recently introduced to increase robustness to heterogeneous data.

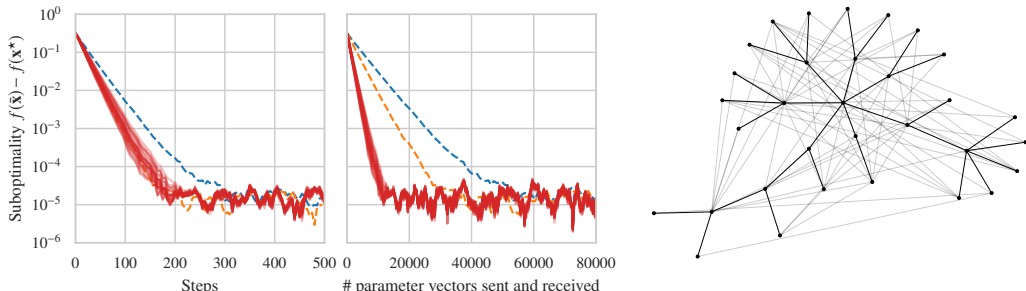

Figure 3: Performance of ■ RelaySGD on spanning trees of the Social Network graph (32 nodes) found using Spanning Tree Protocol, compared to ■ DP-SGD and ■ $D^2$ on the full network. Solid lines (——) indicate spanning trees while dashed lines (- - -) indicate the full graph. The figure on the right shows one spanning tree on top of the original network. Learning rates are tuned to reach suboptimality $\leq 10^{-5}$ on random quadratics ($\zeta^2 = 0.1, \sigma^2 = 0.1, r_0 = 1, L = 1, \mu = 0.5$). ■ RelaySGD on spanning trees converges as fast as ■ $D^2$ on the full network, while the total communication on spanning trees is smaller than on the full graph.

Table 1: Cifar-10 [17] test accuracy with the VGG-11 architecture. We vary the data heterogeneity $\alpha$ [20] between 16 workers. Each method sends/receives 2 models per iteration. We use a ring topology for DP-SGD and $D^2$ because they perform better on rings than on trees. RelaySum with momentum achieves the best results across all levels of data heterogeneity.

| Algorithm | Topology (optimal c.f. Table 2) | $\alpha = 1.00$ (most homogeneous) | $\alpha = 0.1$ | $\alpha = .01$ (most heterogeneous) |
|---|---|---|---|---|
| All-reduce (baseline) | fully connected | 87.0% | 87.0% | 87.0% |
| +momentum | | 90.2% | 90.2% | 90.2% |
| **RelaySGD** | binary trees | 87.4% | 86.9% | 84.6% |
| **+local momentum** | | 90.2% | 89.5% | 89.1% |
| DP-SGD [19] | ring | 87.4% | 79.9% | 53.9% |
| +quasi-global mom. [20] | | 89.5% | 84.8% | 63.3% |
| $D^2$ [36] | ring | 87.2% | 84.0% | 38.2% |
| +local momentum | | 88.2% | 88.5% | 61.0% |
| Stochastic gradient push [2] | time-varying exponential [2] | 87.4% | 86.7% | 86.7% |
| +local momentum | | 89.5% | 89.2% | 87.5% |

Table 1 evaluates RelaySGD in the fully-connected data center setting where we limit the communication budget per iteration to two models. We use 16-workers on Cifar-10, following the experimental details outlined in Appendix B and hyper-parameter tuning procedure from Appendix C. For this experiment, we consider three topologies: (1) double binary trees as described in section 3, (2) rings, and (3) the time-varying exponential scheme of Stochastic Gradient Push (SGP) [2]. Because SGP normally sends/receives only one model per communication round, we execute two synchronous communication steps per gradient update, increasing its latency. The various algorithms compared have different optimal topology choices. In Table 1 we only include the optimal choice for each algorithm. Table 2 qualitatively compares the possible combinations. We opt for the VGG-11 architecture because it does not feature BatchNorm [9]. BatchNorm poses particular challenges to data heterogeneity, and the search for alternatives is an active, and orthogonal, area of research [21].

Even though RelaySGD does not use a time-varying topology, it performs as well as or better than SGP, and RelaySGD with momentum suffers minimal accuracy loss up to heterogeneity $\alpha = 0.01$, a level higher than considered in previous work [20]. While $D^2$ is theoretically independent of data heterogeneity, and while some of its random repetitions yield good results, it is unstable in the very heterogeneous setting. Moreover, Figure 4 shows that workers with RelaySGD achieve high test accuracies quicker during training than with other algorithms.

These findings are confirmed on ImageNet [5] with the ResNet-20-EvoNorm architecture [21] in Table 3. On the BERT fine-tuning task from [20], Table 4 demonstrates that RelaySGD with the Adam optimizer, customary for such NLP tasks, outperforms all compared algorithms.

Table 2: Motivation of topology choices. For each algorithm, we compare 4 topologies configured to send/receive 2 models at each SGD iteration. The algorithms have different optimal topologies.

| Algorithm | Ring | Chain (= spanning tree of ring) | Double binary trees | Time-varying exponential [2] |
|---|---|---|---|---|
| RelaySGD | Unsupported | Worse than double b. trees (E.1) | **Best result** | Unsupported |
| DP-SGD | **Best result** | Worse than ring | Worse than ring (E.1) | Unsupported |
| D$^2$ | **Best result** | Worse than ring | Worse than ring (E.1) | Unsupported |
| SGP | Equivalent to DP-SGD | Equivalent to DP-SGD | Equivalent to DP-SGD | **Best result** |

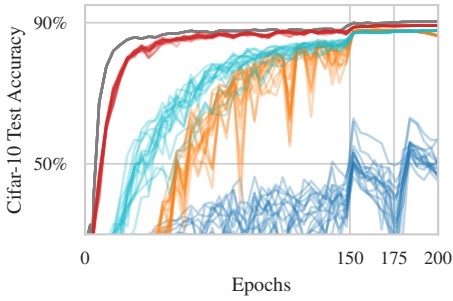

Figure 4: Test accuracy during training of 16 workers with heterogeneous data ($\alpha = 0.01$) on Cifar-10. Like, with the ■ all-reduce baseline, all workers in ■ RelaySGD on double binary trees quickly reach good accuracy, while this takes longer for ■ SGP with time-varying exponential topology and ■ D$^2$ on a ring. ■ DP-SGD does not reach good accuracy with such heterogeneous data.

Table 3: Test accuracies on ImageNet, using 16 workers with heterogeneous data ($\alpha = 0.1$). Even when communicating over a simple chain network, RelaySGD performs similarly to SGP with their time-varying exponential communicating scheme. Methods use default learning rates (Appendix C.2).

| Algorithm | Topology | Top-1 Accuracy |
|---|---|---|
| Centralized (baseline) | fully-connected | 69.7% |
| **RelaySGD w/ momentum** | double binary trees | 60.0% |
| DP-SGD [19] w/ quasi-global momentum [20] | ring | 55.8% |
| D$^2$ [36] w/ momentum | ring | diverged at epoch 65, at 49.5% |
| SGP [2] w/ momentum | time-varying exponential [2] | 58.5% |

| Algorithm | Topology | Top-1 Accuracy |
|---|---|---|
| Centralized Adam | fully-connected | 94.2% ± 0.1% |
| **Relay-Adam** | double binary trees | 93.2% ± 0.6% |
| DP-SGD Adam | ring | 87.3% ± 0.6% |
| Quasi-global Adam [20] | ring | 88.3% ± 0.7% |
| SGP [2] Adam | time-varying exp. | 88.3% ± 0.3% |

Table 4: DistilBERT [34] fine-tuning on AG news data [49] using 16 nodes with heterogeneous data ($\alpha = 0.1$). Transformers are usually trained with Adam, and RelaySGD naturally supports Adam updates. (Appendix B.3).

Table 5: Robustness to unreliable networks. On Cifar-10/VGG-11 with 16 workers and heterogeneous data ($\alpha = 0.01$), we compare momentum versions of the best-performing algorithms from Table 1. Like gossip-based algorithms, RelaySGD with the robust update rule 1 can tolerate up to 10% dropped messages and converge to full test accuracy. Without modification, D$^2$ does not share this property.

| Algorithm | Topology | Reliable network | 1% dropped messages | 10% dropped messages |
|---|---|---|---|---|
| RelaySGD w/ momentum | trees | 89.2% | 89.3% | 89.3% |
| DP-SGD [19] w/ quasi-global mom. [20] | ring | 78.3% | 76.2% | 76.9% |
| D$^2$ [36] w/ momentum | ring | 87.4% | diverges | diverges |
| SGP [2] w/ momentum | time-varying | 88.5% | 88.6% | 88.1% |

## 5.4 Robustness to unreliable communication

Peer-to-peer applications are a central use case for decentralized learning. Decentralized learning algorithms must therefore be robust to workers joining and leaving, and to unreliable communication between workers. Gossip averaging naturally features such robustness, but for methods like $D^2$, that correct for local data biases, achieving such robustness is non-trivial. As a proxy for these challenges, in Table 5, we verify that RelaySGD can tolerate randomly dropped messages. The algorithm achieves this by reliably counting the number of models summed up in each message. For this experiment, we use an extended version of Algorithm 1, where line 10 is replaced by

$$\mathbf{x}_i^{(t+1)} = \tfrac{1}{n}\left( \mathbf{x}_i^{(t+1/2)} + \sum_{j \in \mathcal{N}_i} \mathbf{m}_{j \to i}^{(t)} + (n - \bar{n}_i^{(t+1)})\mathbf{x}_i^{(t)} \right). \tag{1}$$

We count the number of models received as $\bar{n}$, and substitute any missing models ($< n$) by the previous state $\mathbf{x}_i^{(t)}$. RelaySGD trains reliably to good test accuracy with up to 10% deleted messages. This behavior is on par with a similarly modified SGP [2] that corrects for missing energy. In contrast, $D^2$ becomes unstable with undelivered messages and diverges.

# 6 Conclusion

Decentralized learning has great promise as a building block in the democratization of deep learning. Deep learning relies on large datasets, and while large companies can afford those, many individuals together can, too. Of course, their data does not follow the exact same distribution, calling for robustness of decentralized learning algorithms to data heterogeneity. Algorithms with this property have been proposed and analyzed theoretically, but they do not always perform well in deep learning.

In this paper, we propose RelaySGD for distributed optimization over decentralized networks with heterogeneous data. Unlike algorithms based on gossip averaging, RelaySGD *relays* models through spanning trees of a network without decaying their magnitude. This yields an algorithm that is both theoretically independent of data heterogeneity, but also high performing in actual deep learning tasks. With its demonstrated robustness to unreliable communication, RelaySGD makes an attractive choice for peer-to-peer deep learning and applications in large-scale data centers.

## Acknowledgments and Disclosure of Funding

This project was supported by SNSF grant 200020_200342, as well as EU project DIGIPREDICT, and a Google PhD Fellowship.

We thank Yatin Dandi and Lenka Zdeborová for pointing out the similarities between this algorithm and Belief Propagation during a poster session. This discussion helped us find the strongly related article by Zhang et al. [50] that we missed initially.

We thank Renee Vogels for proofreading of the manuscript.

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
