# Contents of the Appendix

## A   Convergence Analysis of RelaySGD

The structure of this section is as follows: Appendix A.1 describes the notations used in the proof; Appendix A.2 introduces the properties of mixing matrix **W** and useful inequalities and lemmas; Appendix A.3 elaborates the results of Theorem I for non-convex, convex, and strongly convex objectives, all of the technical details are deferred to Appendix A.4, Appendix A.5 and Appendix A.6.

### A.1   Notation

We use upper case, bold letters for matrices and lower case, bold letters for vectors. By default, let $\|\cdot\|$ and $\|\cdot\|_F$ be the spectral norm and Frobenius norm for matrices and 2-norm $\|\cdot\|_2$ be the Euclidean norm for vectors.

Let $\tau_{ij}$ be the delay between node $i$ and node $j$ and let $\tau_{\max} = \max_{ij} \tau_{ij}$. Let

$$\mathbf{Z}^{(t)} = [\mathbf{x}_1^{(t)}, \ldots, \mathbf{x}_n^{(t)}]^\top \in \mathbb{R}^{n \times d}$$

be the state at time $t$ and let

$$\nabla \mathbf{F}^{(t)} = [\nabla F_1(\mathbf{x}_1^{(t)}; \xi_1^{(t)}), \ldots, \nabla F_n(\mathbf{x}_n^{(t)}; \xi_n^{(t)})]^\top \in \mathbb{R}^{n \times d}$$

be the worker gradients at time $t$. Denote $\mathbf{Y}^{(t)}$ and $\mathbf{G}^{(t)}$ as the state (models) and gradients respectively, of all nodes, from time $t - \tau_{\max}$ to $t$.

$$\mathbf{Y}^{(t)} = \begin{bmatrix} \mathbf{Z}^{(t)} \\ \mathbf{Z}^{t-1} \\ \vdots \\ \mathbf{Z}^{t-\tau_{\max}} \end{bmatrix} \in \mathbb{R}^{n(\tau_{\max}+1) \times d}, \qquad \mathbf{G}^{(t)} = \begin{bmatrix} \nabla \mathbf{F}^{(t)} \\ \nabla \mathbf{F}^{t-1} \\ \vdots \\ \nabla \mathbf{F}^{t-\tau_{\max}} \end{bmatrix} \in \mathbb{R}^{n(\tau_{\max}+1) \times d}.$$

The mixing matrix $\mathbf{W}$ can be alternatively defined as follows

**Definition E** (Mixing matrix $\mathbf{W}$). *Define $\mathbf{W}, \tilde{\mathbf{W}} \in \mathbb{R}^{n(\tau_{\max}+1) \times n(\tau_{\max}+1)}$ such that RelaySGD can be reformulated as*

$$\mathbf{Y}^{(t+1)} = \underbrace{\begin{bmatrix} \mathbf{W}_0 & \mathbf{W}_1 & \cdots & \mathbf{W}_{\tau_{\max}-1} & \mathbf{W}_{\tau_{\max}} \\ \mathbf{I} & \mathbf{0} & \cdots & \mathbf{0} & \mathbf{0} \\ \vdots & & \ddots & \ddots & \vdots \\ \mathbf{0} & \cdots & \cdots & \mathbf{I} & \mathbf{0} \end{bmatrix}}_{\mathbf{W}} \mathbf{Y}^{(t)} - \gamma \underbrace{\begin{bmatrix} \mathbf{W}_0 & \mathbf{W}_1 & \cdots & \mathbf{W}_{\tau_{\max}-1} & \mathbf{W}_{\tau_{\max}} \\ \mathbf{0} & \mathbf{0} & \cdots & \mathbf{0} & \mathbf{0} \\ \vdots & & \ddots & \ddots & \vdots \\ \mathbf{0} & \cdots & \cdots & \mathbf{0} & \mathbf{0} \end{bmatrix}}_{\tilde{\mathbf{W}}} \mathbf{G}^{(t)}$$

*where $\sum_{i=1}^n \mathbf{W}_i = \frac{1}{n} \mathbf{1}_n \mathbf{1}_n^\top$.*

### A.2 Technical Preliminaries

#### A.2.1 Properties of W.

In this part, we show that $\mathbf{W}$ enjoys similar properties as Perron-Frobenius Theorem in Theorem II and its left dominant eigenvector $\boldsymbol{\pi}$ has specific structure in Lemma 4. Then we use the established tools to prove the key Lemma 1. Finally, we define constants $C$ and $C_1$ in Definition G which are used to simplify the convergence results in Appendix A.3.

**Definition F** (Spectral radius.). *Let $\lambda_1, \ldots, \lambda_n$ be the eigenvalues of a matrix $\mathbf{A} \in \mathbb{C}^{n \times n}$. Then its spectral radius $\rho(\mathbf{A})$ is defined as:*

$$\rho(\mathbf{A}) = \max\{|\lambda_1|, \ldots, |\lambda_n|\}.$$

**Lemma 4.** *The $\mathbf{W}$ in Definition E satisfies*

1. *The spectral radius $\rho(\mathbf{W}) = 1$ and 1 is an eigenvalue of $\mathbf{W}$ and $\mathbf{1}_{n(\tau_{\max}+1)} \in \mathbb{R}^{n(\tau_{\max}+1)}$ is its right eigenvector.*
2. *The left eigenvector $\boldsymbol{\pi} \in \mathbb{R}^{n(\tau_{\max}+1)}$ of eigenvalue 1 is nonnegative and $[\boldsymbol{\pi}]_i = \pi_0 > 0, \forall i \in [n]$ and $\boldsymbol{\pi}^\top \mathbf{1}_{n(\tau_{\max}+1)} = 1$.*

*Proof.* Since $\mathbf{W}$ is a row stochastic matrix, the Gershgorin Circle Theorem asserts the spectral radius

$$\rho(\mathbf{W}) = |\lambda_1(\mathbf{W})| \leq 1.$$

It is clear that 1 is an eigenvalue of $\mathbf{W}$ and $\mathbf{1}_{n(\tau_{\max}+1)}$ is its right eigenvector, we have $\rho(\mathbf{W}) = 1$.

Let $\boldsymbol{\pi} \in \mathbb{R}^{n(\tau_{\max}+1)}$ be the left eigenvector corresponding to 1 and denote it as

$$\boldsymbol{\pi} = \begin{bmatrix} \boldsymbol{\pi}_0 \\ \boldsymbol{\pi}_1 \\ \vdots \\ \boldsymbol{\pi}_{\tau_{\max}} \end{bmatrix} \in \mathbb{R}^{n(\tau_{\max}+1)}$$

where $\boldsymbol{\pi}_i \in \mathbb{R}^n, \forall\, i = 0, 1, \ldots, \tau_{\max}$. Since $\boldsymbol{\pi} = \mathbf{W}^\top \boldsymbol{\pi}$, we have

$$
\begin{bmatrix} \boldsymbol{\pi}_0 \\ \boldsymbol{\pi}1 \\ \vdots \\ \boldsymbol{\pi}_{\tau_{\max}} \end{bmatrix} = \boldsymbol{\pi} = \mathbf{W}^\top \boldsymbol{\pi} = \begin{bmatrix} \mathbf{W}_0^\top \boldsymbol{\pi_0} + \boldsymbol{\pi_1} \\ \mathbf{W}_1^\top \boldsymbol{\pi_0} + \boldsymbol{\pi_2} \\ \vdots \\ \mathbf{W}_{\tau_{\max}-1}^\top \boldsymbol{\pi_0} + \boldsymbol{\pi_{\tau_{\max}}} \\ \mathbf{W}_{\tau_{\max}}^\top \boldsymbol{\pi_0} \end{bmatrix}
$$

which holds true in each block. Then summing up all blocks yields

$$
\sum_{i=0}^{\tau_{\max}} \boldsymbol{\pi_i} = \left( \sum_{i=0}^{\tau_{\max}} \mathbf{W}_i^\top \right) \boldsymbol{\pi_0} + \sum_{i=1}^{\tau_{\max}} \boldsymbol{\pi_i} = \frac{1}{n} \mathbf{1}_n \mathbf{1}_n^\top \boldsymbol{\pi_0} + \sum_{i=1}^{\tau_{\max}} \boldsymbol{\pi_i}
$$

which means $\boldsymbol{\pi}_0 = \frac{1}{n} \mathbf{1}_n \mathbf{1}_n^\top \boldsymbol{\pi}_0$ and therefore $\boldsymbol{\pi}_0 = \pi_0 \mathbf{1}_n$ is a vector of same value.

Other coordinate blocks of $\boldsymbol{\pi}$ can be derived as

$$
\boldsymbol{\pi}_i = \left( \sum_{k=i}^{\tau_{\max}} \mathbf{W}_k^\top \right) \boldsymbol{\pi}_0 \qquad \forall\, i = 1, \ldots, \tau_{\max}.
$$

Since $\mathbf{W}_i$ are nonnegative matrices, we can scale $\boldsymbol{\pi}$ such that $\pi_0 > 0$ and $\mathbf{1}^\top \boldsymbol{\pi} = 1$. Therefore $\boldsymbol{\pi}$ is a nonnegative vector. $\qquad \square$

**Lemma 5.** *If $\lambda \in \mathbb{C}$ is an eigenvalue of $\mathbf{W}$ and $|\lambda| = \rho(\mathbf{W}) = 1$, then $\lambda = 1$ and its geometric multiplicity is 1.*

*Proof.* Let $\boldsymbol{v} \in \mathbb{C}^{n(\tau_{\max}+1)}$ be a right eigenvector corresponding to eigenvalue $\lambda \in \mathbb{C}$ which $|\lambda| = 1$.

Denote $\boldsymbol{v}$ as

$$
\boldsymbol{v} = \begin{bmatrix} \boldsymbol{v}_0 \\ \boldsymbol{v}_1 \\ \vdots \\ \boldsymbol{v}_{\tau_{\max}} \end{bmatrix} \in \mathbb{C}^{n(\tau_{\max}+1)}.
$$

where $\boldsymbol{v}_i \in \mathbb{C}^n, \forall\, i = 0, \ldots, \tau_{\max}$. Then $\mathbf{W}\boldsymbol{v} = \lambda \boldsymbol{v}$ implies

$$
\mathbf{W}\boldsymbol{v} = \begin{bmatrix} \sum_{i=0}^{\tau_{\max}} \mathbf{W}_i \boldsymbol{v_i} \\ \boldsymbol{v}_0 \\ \vdots \\ \boldsymbol{v}_{\tau_{\max}-2} \\ \boldsymbol{v}_{\tau_{\max}-1} \end{bmatrix} = \lambda \boldsymbol{v} = \begin{bmatrix} \lambda \boldsymbol{v}_0 \\ \lambda \boldsymbol{v}1 \\ \vdots \\ \lambda \boldsymbol{v}_{\tau_{\max}} \end{bmatrix}.
$$

The last $\tau$ equations ensures $\boldsymbol{v}_i = \lambda^{-i} \boldsymbol{v}_0$ and thus the first equality becomes

$$
\left( \sum_{i=0}^{\tau_{\max}} \mathbf{W}_i \lambda^{-i} \right) \boldsymbol{v}_0 = \lambda \boldsymbol{v}_0
$$

Denote $\boldsymbol{v}_0 = [x_1, x_2, \ldots, x_n]^\top \in \mathbb{C}^n$, then $\forall\, i = 1, \ldots, n$

$$
\sum_{j=1}^n \frac{1}{n} \lambda^{-\tau_{ij}} x_j = \lambda x_i. \tag{2}
$$

Pick $i$ such that $|\lambda x_i| = \max_j |\lambda x_j|$, then

$$
|\lambda x_i| = \left| \sum_{j=1}^n \frac{1}{n} \lambda^{-\tau_{ij}} x_j \right| \le \frac{1}{n} \sum_{j=1}^n |\lambda^{-\tau_{ij}} x_j| = \frac{1}{n} \sum_{j=1}^n |\lambda^{-\tau_{ij}}||x_j| = \frac{1}{n} \sum_{j=1}^n |x_j| \le |x_i|
$$

where we use the triangular inequality $|a + b| \le |a| + |b|$ and $|ab| = |a||b|$ for all $a, b \in \mathbb{C}$.

Note that as $|\lambda x_i| = |\lambda||x_i| = |x_i|$, the triangular inequality is in fact an equality which means $\lambda^{-\tau_{ij}} x_j$ could be written as

$$
\lambda^{-\tau_{ij}} x_j = a_{ij} \xi \qquad \forall\, j \in [n].
$$

where $a_{ij} \geq 0$ and $\xi \in \mathbb{C}$. Here $\xi \neq 0$, otherwise $\boldsymbol{v} = \boldsymbol{0}$ which contradicts to $\boldsymbol{v}$ is an eigenvector. Then (2) becomes

$$\frac{1}{n} \sum_{j=1}^{n} a_{ij}\xi = \lambda a_{ii}\xi.$$

which implies $|\frac{1}{n} \sum_{j=1}^{n} a_{ij}| = |a_{ii}|$. As $|\lambda x_i| = \max_j |\lambda x_j|$, we know $a_{ii} \geq a_{ij}$ for all $j$, thus

$$a_{i1} = \ldots = a_{in} = a \geq 0,$$

moreover, $a > 0$ as $a = 0$ again leads to $\boldsymbol{v} = \boldsymbol{0}$. Then (2) becomes

$$\lambda a\xi = \lambda x_i = \frac{1}{n} \sum_{j=1}^{n} \lambda^{-\tau_{ij}} x_j = \frac{1}{n} \sum_{j=1}^{n} a\xi = a\xi$$

which shows $\lambda = 1$ as $a > 0$ and $\xi \neq 0$.

Therefore, $\boldsymbol{v}_0 = a\boldsymbol{1}_n \in \mathbb{R}^n$ and $\boldsymbol{v} = a\boldsymbol{1}_{n(\tau_{\max}+1)} \in \mathbb{R}^{n(\tau_{\max}+1)}$. It mean the eigenspace of 1 is one-dimensional and thus its geometric multiplicity is 1. $\qquad\square$

**Lemma 6.** *The algebraic multiplicity of eigenvalue 1 of $\mathbf{W}$ is 1.*

*Proof.* Proof by contradiction. Let $\mathbf{P} \in \mathbb{R}^{n(\tau_{\max}+1) \times n(\tau_{\max}+1)}$ be the invertible matrix which transform $\mathbf{W}$ to its Jordan normal form $\mathbf{J}$ by

$$\mathbf{P}^{-1}\mathbf{W}\mathbf{P} = \mathbf{J} = \begin{bmatrix} \mathbf{J}_1 & & \\ & \ddots & \\ & & \mathbf{J}_p \end{bmatrix}$$

where $\mathbf{J}_1$ is the block for eigenvalue 1. If we assume the algebraic multiplicity of 1 greater equal than 2, and use the Lemma 5 that its geometric multiplicity is 1, then $\mathbf{J}_1$ should look like

$$\mathbf{J}_1 = \begin{bmatrix} 1 & 1 & & \\ & 1 & \ddots & \\ & & \ddots & 1 \\ & & & 1 \end{bmatrix}$$

which is a square matrix of at least 2 columns. Denote the first two columns of $\mathbf{P}$ as $\mathbf{p}_1$ and $\mathbf{p}_2$. We can see that $\mathbf{p}_1 = \boldsymbol{1}_{n(\tau_{\max}+1)}$. Then inspecting $\mathbf{P}^{-1}\mathbf{W}\mathbf{P} = \mathbf{J}$ for $\mathbf{p}_2$ yields

$$\mathbf{W}\mathbf{p}_2 = \mathbf{p}_1 + \mathbf{p}_2 = \boldsymbol{1}_{n(\tau_{\max}+1)} + \mathbf{p}_2.$$

Multiply both sides by $\boldsymbol{\pi}^\top$ gives

$$\boldsymbol{\pi}^\top \mathbf{W}\mathbf{p}_2 = \boldsymbol{\pi}^\top \boldsymbol{1}_{n(\tau_{\max}+1)} + \boldsymbol{\pi}^\top \mathbf{p}_2$$
$$\boldsymbol{\pi}^\top \mathbf{p}_2 = \boldsymbol{\pi}^\top \boldsymbol{1}_{n(\tau_{\max}+1)} + \boldsymbol{\pi}^\top \mathbf{p}_2$$
$$0 = \boldsymbol{\pi}^\top \boldsymbol{1}_{n(\tau_{\max}+1)}$$

which contradicts Lemma 4 that $\boldsymbol{\pi}^\top \boldsymbol{1}_{n(\tau_{\max}+1)} = 1$. Thus the algebraic multiplicity of 1 is 1. $\quad\square$

**Theorem II** (Perron-Frobenius Theorem for $\mathbf{W}$). *The mixing $\mathbf{W}$ of RelaySGD satisfies*

1. *(Positivity) $\rho(\mathbf{W}) = 1$ is an eigenvalue of $\mathbf{W}$.*
2. *(Simplicity) The algebraic multiplicity of 1 is 1.*
3. *(Dominance) $\rho(\mathbf{W}) = |\lambda_1(\mathbf{W})| > |\lambda_2(\mathbf{W})| \geq \ldots \geq |\lambda_{n(\tau_{\max}+1)}(\mathbf{W})|$.*
4. *(Nonnegativity) The $\mathbf{W}$ has a nonnegative left eigenvector $\boldsymbol{\pi}$ and right eigenvector $\boldsymbol{1}_{n(\tau_{\max}+1)}$.*

*Proof.* Statements 1 and 4 follow from Lemma 4. Statement 2 follows from Lemma 6. Statement 3 follows from Lemma 5 and Lemma 6. $\qquad\square$

**Lemma 7** (Gelfand's formula). *For any matrix norm $\|\cdot\|$, we have*

$$\rho(\mathbf{A}) = \lim_{k \to \infty} \|\mathbf{A}^k\|^{\frac{1}{k}}.$$

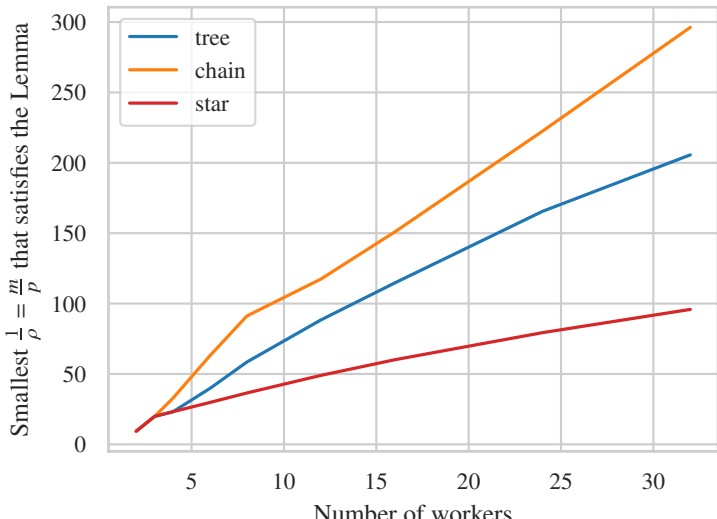

Figure 5: Optimal ratios for $\rho = p/m$ for Lemma 1 computed empirically for three common types of graph topologies.

We characterize the convergence rate of the consensus distance in the following key lemma:

**Lemma' 1** (Key lemma). *Given $\mathbf{W}$ and $\boldsymbol{\pi}$ as before. There exists an integer $m = m(\mathbf{W}) > 0$ such that for any $\mathbf{X} \in \mathbb{R}^{n(\tau_{\max}+1) \times d}$ we have*

$$\|\mathbf{W}^m \mathbf{X} - \mathbf{1}\boldsymbol{\pi}^\top \mathbf{X}\|^2 \leq (1-p)^{2m}\|\mathbf{X} - \mathbf{1}\boldsymbol{\pi}^\top \mathbf{X}\|^2,$$

*where $p = \frac{1}{2}(1 - |\lambda_2(\mathbf{W})|)$ is a constant.*

All the following optimization convergence results will only depend on the *effective spectral gap* $\rho := \frac{p}{m}$ of $\mathbf{W}$. We empirically observe that $\rho = \Theta(1/n)$ for a variety of network topologies, as shown in Figure 5.

*Proof of key lemma 1.* First, let $\{\lambda_i\}$ and $\{\boldsymbol{v}_i\}$ be the eigenvalues and right eigenvectors of $\mathbf{W}$ where $\lambda_1 = 1$ and $\boldsymbol{v}_1 = \mathbf{1}_{n(\tau_{\max}+1)}$, then

$$\begin{aligned}
(\mathbf{W} - \mathbf{1}\boldsymbol{\pi}^\top)\boldsymbol{v}_1 &= (\mathbf{W} - \mathbf{1}\boldsymbol{\pi}^\top)\mathbf{1} = \mathbf{0} \\
(\mathbf{W} - \mathbf{1}\boldsymbol{\pi}^\top)\boldsymbol{v}_i &= \mathbf{W}\boldsymbol{v}_i - \mathbf{1}\boldsymbol{\pi}^\top \boldsymbol{v}_i = \mathbf{W}\boldsymbol{v}_i = \lambda_i \boldsymbol{v}_i \qquad \forall\, i > 1
\end{aligned}$$

where $\boldsymbol{\pi}^\top \boldsymbol{v}_i = 0$ because

$$(1 - \lambda_i)\boldsymbol{\pi}^\top \boldsymbol{v}_i = \boldsymbol{\pi}^\top \boldsymbol{v}_i - \lambda_i \boldsymbol{\pi}^\top \boldsymbol{v}_i = (\boldsymbol{\pi}^\top \mathbf{W})\boldsymbol{v}_i - \boldsymbol{\pi}^\top(\mathbf{W}\boldsymbol{v}_i) = 0.$$

The spectrum of $\mathbf{W} - \mathbf{1}\boldsymbol{\pi}^\top$ are

$$\{0, \lambda_2, \ldots, \lambda_{n(\tau_{\max}+1)}\},$$

and thus the spectral radius of $\mathbf{W} - \mathbf{1}\boldsymbol{\pi}^\top$ is $|\lambda_2| < 1$. Since

$$\mathbf{W}^m - \mathbf{1}\boldsymbol{\pi}^\top = (\mathbf{W} - \mathbf{1}\boldsymbol{\pi}^\top)^m,$$

then $\mathbf{W}^m - \mathbf{1}\boldsymbol{\pi}^\top$ has a spectral radius of $|\lambda_2|^m < 1$.

Then, we apply Gelfand's formula (Lemma 7) with $\mathbf{A} = \mathbf{W} - \mathbf{1}\boldsymbol{\pi}^\top$ and can conclude that for a given $\varepsilon \in (0, 1 - |\lambda_2|)$, there exists a large enough integer $m > 0$ such that

$$\|\mathbf{W}^m - \mathbf{1}\boldsymbol{\pi}^\top\| = \|(\mathbf{W} - \mathbf{1}\boldsymbol{\pi}^\top)^m\| \leq (\rho(\mathbf{W} - \mathbf{1}\boldsymbol{\pi}^\top) + \varepsilon)^m = (|\lambda_2| + \varepsilon)^m < 1.$$

Thus

$$\|\mathbf{W}^m \mathbf{X} - \mathbf{1}\boldsymbol{\pi}^\top \mathbf{X}\|^2 \leq \|\mathbf{W}^m - \mathbf{1}\boldsymbol{\pi}^\top\|^2 \|\mathbf{X} - \mathbf{1}\boldsymbol{\pi}^\top \mathbf{X}\|^2 \leq (1-p)^{2m}\|\mathbf{X} - \mathbf{1}\boldsymbol{\pi}^\top \mathbf{X}\|^2$$

where $p \in (0, 1 - |\lambda_2|)$. $\qquad\square$

**Definition G.** *Given* $\mathbf{W}$ *and* $m$, *and* $\tilde{\mathbf{I}} \in \mathbb{R}^{n(\tau_{\max}+1) \times n(\tau_{\max}+1)}$ *is a matrix which satisfies*

$$[\tilde{\mathbf{I}}]_{ij} = \begin{cases} 1 & i = j \leq n \\ 0 & Otherwise. \end{cases}$$

*We define constants* $C_1^2 := \max_{i=0,\dots,m-1} \|\mathbf{W}^i \tilde{\mathbf{I}}\|^2$ *and* $C = C(\mathbf{W})$ *such that*

$$C^2 := \frac{C_1^2}{\|\mathbf{W}^\infty \tilde{\mathbf{I}}\|^2}.$$

*where* $\mathbf{W}^\infty := \mathbf{1}\boldsymbol{\pi}^\top$.

In addition, the $\|\mathbf{1}\boldsymbol{\pi}^\top \tilde{\mathbf{I}}\|^2$ can be computed as follows.

**Lemma 8.** *Given* $\tilde{\mathbf{I}}$ *in Definition G, we have the following estimate*

$$\|\mathbf{1}\boldsymbol{\pi}^\top \tilde{\mathbf{I}}\|^2 = n^2(\tau_{\max}+1)\pi_0^2 \leq n^3\pi_0^2.$$

*Proof.* For rank $r$ matrix $\|A\|^2 \leq \|A\|_F^2 \leq r\|A\|^2$. Since $\mathbf{1}\boldsymbol{\pi}^\top \tilde{\mathbf{I}}$ is a rank 1 matrix, we know that

$$\|\mathbf{1}\boldsymbol{\pi}^\top \tilde{\mathbf{I}}\|^2 = \|\mathbf{1}\boldsymbol{\pi}^\top \tilde{\mathbf{I}}\|_F^2.$$

As the first n entries of $\boldsymbol{\pi}$ are $\pi_0$, we can compute that

$$\|\mathbf{1}\boldsymbol{\pi}^\top \tilde{\mathbf{I}}\|_F^2 = n^2(\tau_{\max}+1)\pi_0^2.$$

$\square$

### A.2.2 Useful inequalities and lemmas

For convex objective, the noise in Assumption B can be defined only at the minimizer $\mathbf{x}^\star$ which leads to Assumption H. This assumption is used in the proof of Proposition III.

**Assumption H** (Bounded noise at the optimum). *Let* $\mathbf{x}^\star = \arg\min f(\mathbf{x})$ *and define*

$$\zeta_i^2 := \|\nabla f_i(\mathbf{x}^\star)\|^2, \qquad \bar{\zeta}^2 := \frac{1}{n}\sum_{i=1}^n \zeta_i^2. \tag{3}$$

*Further, define*

$$\sigma_i^2 := \mathbb{E}_{\xi_i} \|\nabla F_i(\mathbf{x}^\star, \xi_i) - \nabla f_i(\mathbf{x}^\star)\|^2$$

*and similarly as above,* $\bar{\sigma}^2 := \frac{1}{n}\sum_{i=1}^n \sigma_i^2$. *We assume that* $\bar{\sigma}^2$ *and* $\bar{\zeta}^2$ *are bounded.*

**Lemma 9** (Cauchy-Schwartz inequality). *For arbitrary set of* $n$ *vectors* $\{\mathbf{a}_i\}_{i=1}^n$, $a_i \in \mathbb{R}^d$

$$\left\|\sum_{i=1}^n \mathbf{a}_i\right\|^2 \leq n\sum_{i=1}^n \|\mathbf{a}_i\|^2. \tag{4}$$

**Lemma 10.** *If function* $g(\mathbf{x})$ *is L-smooth, then*

$$\|\nabla g(\mathbf{x}) - \nabla g(\mathbf{y})\|^2 \leq 2L(g(\mathbf{x}) - g(\mathbf{y}) - \langle \mathbf{x} - \mathbf{y}, \nabla g(\mathbf{y})\rangle), \qquad \forall\, \mathbf{x}, \mathbf{y} \in \mathbb{R}^d. \tag{5}$$

**Lemma 11.** *Let* $\mathbf{A}$ *be a matrix with* $\{\mathbf{a}_i\}_{i=1}^n$ *as its columns and* $\bar{\mathbf{a}} = \frac{1}{n}\sum_{i=1}^n \mathbf{a}_i$, $\bar{\mathbf{A}} = \bar{\mathbf{a}}\mathbf{1}^\top$ *then*

$$\|\mathbf{A} - \bar{\mathbf{A}}\|_F^2 = \sum_{i=1}^n \|\mathbf{a}_i - \bar{\mathbf{a}}\|^2 \leq \sum_{i=1}^n \|\mathbf{a}_i\|^2 = \|\mathbf{A}\|_F^2. \tag{6}$$

**Lemma 12.** *Let* $\mathbf{A}, \mathbf{B}$ *be two matrices*

$$\|\mathbf{AB}\|_F^2 \leq \|\mathbf{A}\|_F^2 \|\mathbf{B}\|^2. \tag{7}$$

## A.3 Results of Theorem I

In this subsection, we summarize the precise results of Theorem I for convex, strongly convex and non-convex cases. The complete proofs for each case are then given in the following Appendix A.4, Appendix A.5 and Appendix A.6.

**Theorem' I.** *Given mixing matrix* $\mathbf{W}$ *and* $\tilde{\mathbf{W}}$, *constant* $m$, $p$ *defined in Lemma 1,* $C$, $C_1$ *defined in Definition G. Under Assumption A and B, then for any target accuracy* $\varepsilon > 0$,

***Non-convex:*** *if the objective is non-convex, then* $\frac{1}{T+1}\sum_{t=0}^{T}\|\nabla f(\bar{\mathbf{x}}^{(t)})\|^2 \leq \varepsilon$ *after*

$$\mathcal{O}\left(\frac{\bar{\sigma}^2}{n\varepsilon^2} + \frac{Cm\bar{\sigma}}{\sqrt{p}\varepsilon^{3/2}} + \frac{C_1 m}{p\varepsilon}\right)Lr_0$$

*iterations, where* $r_0 = f(\mathbf{x}^{(0)}) - f^\star$.

***Convex:*** *if the objective is convex and* $\mathbf{x}^\star$ *is the minimizer, then* $\frac{1}{T+1}\sum_{t=0}^{T}\left(f(\bar{\mathbf{x}}^{(t)}) - f(\mathbf{x}^\star)\right) \leq \varepsilon$ *after*

$$\mathcal{O}\left(\frac{\bar{\sigma}^2}{n\varepsilon^2} + \frac{Cm\sqrt{L}\bar{\sigma}}{\sqrt{p}\varepsilon^{3/2}} + \frac{Lm\sqrt{n}C}{p\varepsilon}\right)r_0$$

*iterations, where* $r_0 = \|\mathbf{x}^0 - \mathbf{x}^\star\|^2$.

***Strongly-convex:*** *if the objective is* $\mu$ *strongly convex and* $\mathbf{x}^\star$ *is the minimizer, then* $\frac{1}{W_T}\sum_{t=0}^{T}w_t(\mathbb{E}\,f(\bar{\mathbf{x}}^{(t)}) - f^\star) + \mu\,\mathbb{E}\,\|\bar{\mathbf{x}}^{(T+1)} - \mathbf{x}^\star\|^2 \leq \varepsilon$ *after*

$$\tilde{\mathcal{O}}\left(\frac{\bar{\sigma}^2}{\mu n\varepsilon^2} + \frac{Lm^2C^2\bar{\sigma}^2}{\mu np^2\varepsilon} + \frac{s}{a}\log\frac{bsr_0}{\varepsilon}\right)$$

*iterations, where* $r_0 = \|\mathbf{x}^0 - \mathbf{x}^\star\|^2$, $w_t = (1 - \frac{\mu\gamma n\pi_0}{2})^{-(t+1)}$ *and* $W_T = \sum_{t=0}^{T}w_t$ *and* $a = \frac{\mu n\pi_0}{2}$, $b = \frac{2}{n\pi_0}$, $s = \frac{aT}{\ln\max\{\frac{ba^2T^2r_0}{\pi_0\bar{\sigma}^2},2\}}$.

In all three cases, the convergence rate is independent of the heterogeneity $\zeta^2$.

## A.4 Proof of Theorem I in the convex case

Let $\bar{\mathbf{x}}^{(t)} := \left(\boldsymbol{\pi}^\top\mathbf{Y}^{(t)}\right)^\top$ and $\bar{\mathbf{Y}}^{(t)} := \mathbf{1}\boldsymbol{\pi}^\top\mathbf{Y}^{(t)}$. Let $\mathbf{x}^\star$ be the minimizer of $f$ and define the following iterates

- $r_t := \|\bar{\mathbf{x}}^{(t)} - \mathbf{x}^\star\|^2$,
- $e_t := f(\bar{\mathbf{x}}^{(t)}) - f(\mathbf{x}^\star)$,
- $\Xi_t := \frac{1}{n}\|\bar{\mathbf{Y}}^{(t)} - \mathbf{Y}^{(t)}\|_F^2$.

The consensus distance $\Xi_t$ can be written as follows

$$\Xi_t = \frac{1}{n}\sum_{i=1}^{n}\sum_{\tau=0}^{\tau_{\max}}\|\bar{\mathbf{x}}^{(t)} - \mathbf{x}_i^{(t-\tau)}\|^2. \tag{8}$$

There is a related term $\sum_{i=1}^{n}\sum_{j=1}^{n}\|\bar{\mathbf{x}}^{(t)} - \mathbf{x}_i^{(t-\tau_{ij})}\|^2$ which will be used frequently in the proof. The next lemma explains their relations.

**Lemma 13.** *For all* $t \geq 0$

$$\sum_{i=1}^{n}\sum_{j=1}^{n}\|\bar{\mathbf{x}}^{(t)} - \mathbf{x}_i^{(t-\tau_{ij})}\|^2 \leq n^2\Xi_t.$$

*where* $\mathbf{x}^{(0)} = \mathbf{x}^{(-1)} = \ldots = \mathbf{x}^{(-\tau_{\max})}$.

*Proof.* Rewrite the $\tau_{ij}$ as an indicator function

$$\sum_{i=1}^{n}\sum_{j=1}^{n}\|\bar{\mathbf{x}}^{(t)} - \mathbf{x}_i^{(t-\tau_{ij})}\|^2 = \sum_{i=1}^{n}\sum_{j=1}^{n}\sum_{\tau=0}^{\tau_{\max}}\mathbf{1}_{\{\tau=\tau_{ij}\}}\|\bar{\mathbf{x}}^{(t)} - \mathbf{x}_i^{(t-\tau)}\|^2.$$

This term can be relaxed by removing the indicator function

$$\sum_{i=1}^{n}\sum_{j=1}^{n}\|\bar{\mathbf{x}}^{(t)} - \mathbf{x}_i^{(t-\tau_{ij})}\|^2 \leq n\sum_{i=1}^{n}\sum_{\tau=0}^{\tau_{\max}}\|\bar{\mathbf{x}}^{(t)} - \mathbf{x}_i^{(t-\tau)}\|^2.$$

Then applying (8) for the consensus distance in vector form completes the proof. $\qquad\square$

The next two propositions upper bound the difference between stochastic gradients and full gradients.

**Proposition III.** *Under Assumption A and B. Then for $t \geq 0$,*

$$\mathbb{E}\left\|\boldsymbol{\pi}^\top \tilde{\mathbf{W}}(\mathbb{E}\,\mathbf{G}^{(t)} - \mathbf{G}^{(t)})\right\|^2 \leq 3n\pi_0^2(L^2\Xi_t + 2Le_t + \bar{\sigma}^2).$$

*Proof.* Use $T_0$ to denote the left hand side quantity

$$T_0 \quad := \quad \mathbb{E}\left\|\boldsymbol{\pi}^\top \tilde{\mathbf{W}}(\mathbb{E}\,\mathbf{G}^{(t)} - \mathbf{G}^{(t)})\right\|^2$$

$$= \quad \mathbb{E}\left\|\frac{\pi_0}{n}\sum_{i=1}^{n}\sum_{j=1}^{n}(\nabla f_j(\mathbf{x}_j^{(t-\tau_{ij})}) - \nabla F_j(\mathbf{x}_j^{(t-\tau_{ij})};\xi_j^{(t-\tau_{ij})}))\right\|^2$$

$$\overset{\text{Cauchy-Schwartz (4)}}{\leq} \quad \frac{\pi_0^2}{n}\sum_{i=1}^{n}\mathbb{E}\left\|\sum_{j=1}^{n}(\nabla f_j(\mathbf{x}_j^{(t-\tau_{ij})}) - \nabla F_j(\mathbf{x}_j^{(t-\tau_{ij})};\xi_j^{(t-\tau_{ij})}))\right\|^2.$$

Since the randomness inside the norm are independent, we have

$$T_0 \leq \frac{\pi_0^2}{n}\sum_{i=1}^{n}\sum_{j=1}^{n}\mathbb{E}\left\|\nabla f_j(\mathbf{x}_j^{(t-\tau_{ij})}) - \nabla F_j(\mathbf{x}_j^{(t-\tau_{ij})};\xi_j^{(t-\tau_{ij})})\right\|^2.$$

Inside the vector norm, we can add and subtract terms the same terms and apply Cauchy-Schwartz (4)

$$T_0 \leq \frac{3\pi_0^2}{n}\sum_{i=1}^{n}\sum_{j=1}^{n}\mathbb{E}\left\|\nabla F_j(\mathbf{x}_j^{(t-\tau_{ij})};\xi_j^{(t-\tau_{ij})}) - \nabla F_j(\bar{\mathbf{x}}^{(t)};\xi_j^{(t-\tau_{ij})}) + \nabla f_j(\mathbf{x}_j^{(t-\tau_{ij})}) - \nabla f_j(\bar{\mathbf{x}}^{(t)})\right\|^2$$

$$+ \frac{3\pi_0^2}{n}\sum_{i=1}^{n}\sum_{j=1}^{n}\mathbb{E}\left\|\nabla F_j(\bar{\mathbf{x}}^{(t)};\xi_j^{(t-\tau_{ij})}) - \nabla F_j(\mathbf{x}^\star;\xi_j^{(t-\tau_{ij})}) + \nabla f_j(\bar{\mathbf{x}}^{(t)}) - \nabla f_j(\mathbf{x}^\star)\right\|^2$$

$$+ \frac{3\pi_0^2}{n}\sum_{i=1}^{n}\sum_{j=1}^{n}\mathbb{E}\left\|\nabla F_j(\mathbf{x}^\star;\xi_j^{(t-\tau_{ij})})) - \nabla f_j(\mathbf{x}^\star)\right\|^2.$$

Use the inequality that for $a = \mathbb{E}\,Y$, $\mathbb{E}\|Y - a\|^2 = \mathbb{E}\|Y\|^2 - \|a\|^2 \leq \mathbb{E}\|Y\|^2$, then we have

$$T_0 \leq \frac{3\pi_0^2}{n}\sum_{i=1}^{n}\sum_{j=1}^{n}\mathbb{E}\left\|\nabla F_j(\mathbf{x}_j^{(t-\tau_{ij})};\xi_j^{(t-\tau_{ij})}) - \nabla F_j(\bar{\mathbf{x}}^{(t)};\xi_j^{(t-\tau_{ij})})\right\|^2$$

$$+ \frac{3\pi_0^2}{n}\sum_{i=1}^{n}\sum_{j=1}^{n}\mathbb{E}\left\|\nabla F_j(\bar{\mathbf{x}}^{(t)};\xi_j^{(t-\tau_{ij})}) - \nabla F_j(\mathbf{x}^\star;\xi_j^{(t-\tau_{ij})})\right\|^2$$

$$+ \frac{3\pi_0^2}{n}\sum_{i=1}^{n}\sum_{j=1}^{n}\mathbb{E}\left\|\nabla F_j(\mathbf{x}^\star;\xi_j^{(t-\tau_{ij})}) - \nabla f_j(\mathbf{x}^\star)\right\|^2.$$

Applying Assumption A, Smoothness (5), and Assumption B (or Assumption H) to the three terms gives

$$T_0 \leq \frac{3L^2\pi_0^2}{n}\sum_{i=1}^{n}\sum_{j=1}^{n}\left\|\mathbf{x}_j^{(t-\tau_{ij})} - \bar{\mathbf{x}}^{(t)}\right\|^2 + 6Ln\pi_0^2(f(\bar{\mathbf{x}}^{(t)}) - f(\mathbf{x}^\star)) + 3\pi_0^2 n\bar{\sigma}^2$$

$$\overset{\text{Lemma 13}}{\leq} 3n\pi_0^2(L^2\Xi_t + 2Le_t + \bar{\sigma}^2).$$

where in the last line we have used our previous Lemma 13. $\qquad\square$

The next proposition is very similar to the Proposition III except that it considers the matrix form instead of the projection onto $\boldsymbol{\pi}$.

**Proposition IV.** *Under Assumption A and B. Then for $t \geq 0$,*

$$\mathbb{E} \left\| \tilde{\mathbf{W}}(\mathbb{E}\, \mathbf{G}^{(t)} - \mathbf{G}^{(t)}) \right\|_F^2 \leq 3(L^2 \Xi_t + 2L e_t + \bar{\sigma}^2).$$

*Proof.*

$$\mathbb{E} \left\| \tilde{\mathbf{W}}(\mathbb{E}\, \mathbf{G}^{(t)} - \mathbf{G}^{(t)}) \right\|_F^2$$

$$= \sum_{i=1}^n \mathbb{E} \left\| \frac{1}{n} \sum_{j=1}^n (\nabla F(\mathbf{x}_j^{(t-\tau_{ij})}; \xi_j^{(t-\tau_{ij})}) - \nabla f_j(\mathbf{x}_j^{(t-\tau_{ij})})) \right\|^2$$

$$\leq \frac{1}{n^2} \sum_{i=1}^n \sum_{j=1}^n \mathbb{E} \left\| \nabla F(\mathbf{x}_j^{(t-\tau_{ij})}; \xi_j^{(t-\tau_{ij})}) - \nabla f_j(\mathbf{x}_j^{(t-\tau_{ij})}) \right\|^2$$

The rest of the proof is identical to the one of Proposition III. $\qquad\qquad\square$

**Lemma 14.** *(Descent lemma for convex objective.) If $\gamma \leq \frac{1}{10Ln\pi_0}$, then*

$$r_{t+1} \leq (1 - \tfrac{\gamma\mu n\pi_0}{2})r_t - \gamma n\pi_0 e_t + 4\gamma Ln\pi_0 \Xi_t + 3\gamma^2 n\pi_0^2 \bar{\sigma}^2.$$

*Proof.* Expand $r_{t+1} = \mathbb{E} \left\| \bar{\mathbf{x}}^{(t+1)} - \mathbf{x}^\star \right\|^2$ as follows

$$\mathbb{E} \left\| \bar{\mathbf{x}}^{(t+1)} - \mathbf{x}^\star \right\|^2 = \mathbb{E} \left\| \bar{\mathbf{x}}^{(t)} - \gamma\boldsymbol{\pi}^\top \tilde{\mathbf{W}} \mathbf{G}^{(t)} - \mathbf{x}^\star \right\|^2$$

$$= \mathbb{E} \left\| \bar{\mathbf{x}}^{(t)} - \gamma\boldsymbol{\pi}^\top \tilde{\mathbf{W}} \mathbb{E}\, \mathbf{G}^{(t)} - \mathbf{x}^\star + \gamma\boldsymbol{\pi}^\top \tilde{\mathbf{W}}(\mathbb{E}\, \mathbf{G}^{(t)} - \mathbf{G}^{(t)}) \right\|^2$$

Directly expand it into three terms

$$\mathbb{E} \left\| \bar{\mathbf{x}}^{(t+1)} - \mathbf{x}^\star \right\|^2 = \mathbb{E} \left( \left\| \bar{\mathbf{x}}^{(t)} - \gamma\boldsymbol{\pi}^\top \tilde{\mathbf{W}} \mathbb{E}\, \mathbf{G}^{(t)} - \mathbf{x}^\star \right\|^2 + \gamma^2 \| \boldsymbol{\pi}^\top \tilde{\mathbf{W}}(\mathbb{E}\, \mathbf{G}^{(t)} - \mathbf{G}^{(t)}) \|^2 \right.$$

$$\left. + \left\langle \bar{\mathbf{x}}^{(t)} - \gamma\boldsymbol{\pi}^\top \tilde{\mathbf{W}} \mathbb{E}\, \mathbf{G}^{(t)} - \mathbf{x}^\star, \gamma\boldsymbol{\pi}^\top \tilde{\mathbf{W}}(\mathbb{E}\, \mathbf{G}^{(t)} - \mathbf{G}^{(t)}) \right\rangle \right)$$

where the 3rd term is 0 and the second term is bounded in Proposition III. The first term is independent of the randomness

$$\left\| \bar{\mathbf{x}}^{(t)} - \gamma\boldsymbol{\pi}^\top \tilde{\mathbf{W}} \mathbb{E}\, \mathbf{G}^{(t)} - \mathbf{x}^\star \right\|^2$$

$$= \left\| \bar{\mathbf{x}}^{(t)} - \mathbf{x}^\star \right\|^2 + \gamma^2 \underbrace{\left\| \boldsymbol{\pi}^\top \tilde{\mathbf{W}} \mathbb{E}\, \mathbf{G}^{(t)} \right\|^2}_{=:T_1} - 2\gamma \underbrace{\left\langle \boldsymbol{\pi}^\top \tilde{\mathbf{W}} \mathbb{E}\, \mathbf{G}^{(t)}, \bar{\mathbf{x}}^{(t)} - \mathbf{x}^\star \right\rangle}_{=:T_2}.$$

Since $\boldsymbol{\pi}^\top \tilde{\mathbf{W}} \mathbb{E}\, \mathbf{G}^{(t)} = \frac{\pi_0}{n} \sum_{i=1}^n \sum_{j=1}^n \nabla f_i(\mathbf{x}_i^{(t-\tau_{ij})})$, first bound $T_1$

$$T_1 = \qquad \pi_0^2 \left\| \frac{1}{n} \sum_{i=1}^n \sum_{j=1}^n \nabla f_i(\mathbf{x}_i^{(t-\tau_{ij})}) \right\|^2$$

$$= \qquad \pi_0^2 \left\| \frac{1}{n} \sum_{i=1}^n \sum_{j=1}^n (\nabla f_i(\mathbf{x}_i^{(t-\tau_{ij})}) - \nabla f_i(\bar{\mathbf{x}}^{(t)}) + \nabla f_i(\bar{\mathbf{x}}^{(t)}) - \nabla f_i(\mathbf{x}^\star)) \right\|^2$$

$$\leq \qquad 2\pi_0^2 \left( \left\| \frac{1}{n} \sum_{i=1}^n \sum_{j=1}^n (\nabla f_i(\mathbf{x}_i^{(t-\tau_{ij})}) - \nabla f_i(\bar{\mathbf{x}}^{(t)})) \right\|^2 + \left\| \sum_{i=1}^n (\nabla f_i(\bar{\mathbf{x}}^{(t)}) - \nabla f_i(\mathbf{x}^\star)) \right\|^2 \right)$$

$$\leq \qquad 2\pi_0^2 L^2 \sum_{i=1}^n \sum_{j=1}^n \left\| \mathbf{x}_i^{(t-\tau_{ij})} - \bar{\mathbf{x}}^{(t)} \right\|^2 + 2n\pi_0^2 \sum_{i=1}^n \left\| \nabla f_i(\bar{\mathbf{x}}^{(t)}) - \nabla f_i(\mathbf{x}^\star) \right\|^2$$

$$\overset{\text{Smoothness (5)}}{\leq} \quad 2\pi_0^2 L^2 \sum_{i=1}^n \sum_{j=1}^n \left\| \mathbf{x}_i^{(t-\tau_{ij})} - \bar{\mathbf{x}}^{(t)} \right\|^2 + 4Ln^2\pi_0^2(f(\bar{\mathbf{x}}^{(t)}) - f(\mathbf{x}^\star)),$$

Using again Lemma 13 we have

$$T_1 \leq 2L^2 n^2 \pi_0^2 \Xi_t + 4Ln^2 \pi_0^2 e_t.$$

Then bound $T_2$

$$
\begin{aligned}
T_2 =\quad & \frac{\pi_0}{n} \sum_{i=1}^n \sum_{j=1}^n \langle \nabla f_i(\mathbf{x}_i^{(t-\tau_{ij})}), \bar{\mathbf{x}}^{(t)} - \mathbf{x}^\star \rangle \\
=\quad & \frac{\pi_0}{n} \sum_{i=1}^n \sum_{j=1}^n (\langle \nabla f_i(\mathbf{x}_i^{(t-\tau_{ij})}), \bar{\mathbf{x}}^{(t)} - \mathbf{x}_i^{(t-\tau_{ij})} \rangle + \langle \nabla f_i(\mathbf{x}_i^{(t-\tau_{ij})}), \mathbf{x}_i^{(t-\tau_{ij})} - \mathbf{x}^\star \rangle) \\
\geq\quad & \frac{\pi_0}{n} \sum_{i=1}^n \sum_{j=1}^n (f_i(\bar{\mathbf{x}}^{(t)}) - f_i(\mathbf{x}_i^{(t-\tau_{ij})}) - \tfrac{L}{2}\|\bar{\mathbf{x}}^{(t)} - \mathbf{x}_i^{(t-\tau_{ij})}\|^2 \\
& + f_i(\mathbf{x}_i^{(t-\tau_{ij})}) - f_i(\mathbf{x}^\star) + \tfrac{\mu}{2}\|\mathbf{x}_i^{(t-\tau_{ij})} - \mathbf{x}^\star\|^2) \\
=\quad & n\pi_0(f(\bar{\mathbf{x}}^{(t)}) - f(\mathbf{x}^\star)) + \frac{\pi_0}{n} \sum_{i=1}^n \sum_{j=1}^n (\tfrac{\mu}{2}\|\mathbf{x}_i^{(t-\tau_{ij})} - \mathbf{x}^\star\|^2 - \tfrac{L}{2}\|\bar{\mathbf{x}}^{(t)} - \mathbf{x}_i^{(t-\tau_{ij})}\|^2) \\
\geq\quad & n\pi_0(f(\bar{\mathbf{x}}^{(t)}) - f(\mathbf{x}^\star)) + \frac{\pi_0}{n} \sum_{i=1}^n \sum_{j=1}^n (\tfrac{\mu}{4}\|\bar{\mathbf{x}}^{(t)} - \mathbf{x}^\star\|^2 - \tfrac{\mu+L}{2}\|\bar{\mathbf{x}}^{(t)} - \mathbf{x}_i^{(t-\tau_{ij})}\|^2) \\
\overset{\text{Lemma 13}}{\geq}\quad & n\pi_0 e_t + \tfrac{n\mu\pi_0}{4} r_t - nL\pi_0 \Xi_t
\end{aligned}
$$

where the first inequality and the second inequality uses the $L$-smoothness and $\mu$-convexity of $f_i$.
Combine both $T_1$, $T_2$ and Proposition III we have

$$
\begin{aligned}
r_{t+1} \leq & r_t + \gamma^2 n^2 \pi_0^2 (2L^2 \Xi_t + 4Le_t) - 2\gamma n\pi_0(e_t + \tfrac{\mu}{4} r_t - L\Xi_t) \\
& + \gamma^2 n (3L^2 \pi_0^2 \Xi_t + 6L\pi_0^2 e_t + 3\pi_0^2 \bar{\sigma}^2) \\
= & (1 - \tfrac{\gamma\mu n\pi_0}{2}) r_t - (2\gamma n\pi_0 - 4L\gamma^2 n^2 \pi_0^2 - 6L\gamma^2 n\pi_0^2) e_t \\
& + (2\gamma^2 L^2 n^2 \pi_0^2 + 2\gamma Ln\pi_0 + 3L^2 \gamma^2 n\pi_0^2) \Xi_t + 3\gamma^2 n\pi_0^2 \bar{\sigma}^2
\end{aligned}
$$

In addition if $\gamma \leq \frac{1}{10Ln\pi_0}$, then we can simplify the coefficient of $e_t$ and $\Xi_t$

$$
\begin{aligned}
4L\gamma^2 n^2 \pi_0^2 + 6L\gamma^2 n\pi_0^2 & \leq \gamma n\pi_0 \\
2\gamma^2 L^2 n^2 \pi_0^2 + 2\gamma Ln\pi_0 + 3L^2 \gamma^2 n\pi_0^2 & \leq 4\gamma Ln\pi_0
\end{aligned}
$$

Then

$$r_{t+1} \leq (1 - \tfrac{\gamma\mu n\pi_0}{2}) r_t - \gamma n\pi_0 e_t + 4\gamma Ln\pi_0 \Xi_t + 3\gamma^2 n\pi_0^2 \bar{\sigma}^2$$

**Lemma 15.** *For $\gamma \leq \frac{p}{10LmC_1}$ we have*

$$\frac{1}{T+1} \sum_{t=0}^T \Xi_t \leq C_1^2 \gamma^2 m^2 \frac{24}{p} \frac{\bar{\sigma}^2}{n} + \frac{80Lm^2}{p^2} C_1^2 \gamma^2 \frac{1}{T+1} \sum_{t=0}^T e_t$$

*where $C_1$ is defined in Definition G.*

*Proof.* First bound the consensus distance as follows:

$$
\begin{aligned}
n\Xi_t = \mathbb{E}\|\mathbf{Y}^{(t)} - \bar{\mathbf{Y}}^{(t)}\|_F^2 & \leq \mathbb{E}\|(\mathbf{Y}^{(t)} - \bar{\mathbf{Y}}^{(t-m)}) - (\bar{\mathbf{Y}}^{(t)} - \bar{\mathbf{Y}}^{(t-m)})\|_F^2 \\
& \leq \mathbb{E}\|\mathbf{Y}^{(t)} - \bar{\mathbf{Y}}^{(t-m)}\|_F^2
\end{aligned}
$$

where the last inequality we use the simple matrix inequality (6). For $t \geq m$ unroll to $t - m$.

$$n\Xi_t \leq \mathbb{E}\left\| \mathbf{W}^m \mathbf{Y}^{(t-m)} - \gamma \sum_{k=t-m}^{t-1} \mathbf{W}^{t-1-k} \tilde{\mathbf{W}} \mathbf{G}^{(k)} - \bar{\mathbf{Y}}^{(t-m)} \right\|_F^2$$

Separate the stochastic part and deterministic part.

$$n\Xi_t \leq \left\| \mathbf{W}^m \mathbf{Y}^{(t-m)} - \gamma \sum_{k=t-m}^{t-1} \mathbf{W}^{t-1-k} \tilde{\mathbf{W}} \, \mathbb{E} \, \mathbf{G}^{(k)} - \bar{\mathbf{Y}}^{(t-m)} \right\|_F^2$$

$$+ \mathbb{E} \left\| \gamma \sum_{k=t-m}^{t-1} \mathbf{W}^{t-1-k} \tilde{\mathbf{W}} (\mathbb{E} \, \mathbf{G}^{(k)} - \mathbf{G}^{(k)}) \right\|_F^2$$

$$\leq \left\| \mathbf{W}^m \mathbf{Y}^{(t-m)} - \gamma \sum_{k=t-m}^{t-1} \mathbf{W}^{t-1-k} \tilde{\mathbf{W}} \, \mathbb{E} \, \mathbf{G}^{(k)} - \bar{\mathbf{Y}}^{(t-m)} \right\|_F^2$$

$$+ \gamma^2 m \sum_{k=t-m}^{t-1} \mathbb{E} \left\| \mathbf{W}^{t-1-k} \tilde{\mathbf{W}} (\mathbb{E} \, \mathbf{G}^{(k)} - \mathbf{G}^{(k)}) \right\|_F^2$$

Given $\tilde{\mathbf{I}}$ and $C_1$ in defined in Definition G, we know that $\tilde{\mathbf{W}} = \tilde{\mathbf{I}} \tilde{\mathbf{W}}$. Then use (7) and Proposition IV

$$n\Xi_t \leq \left\| \mathbf{W}^m \mathbf{Y}^{(t-m)} - \gamma \sum_{k=t-m}^{t-1} \mathbf{W}^{t-1-k} \tilde{\mathbf{W}} \, \mathbb{E} \, \mathbf{G}^{(k)} - \bar{\mathbf{Y}}^{(t-m)} \right\|_F^2$$

$$+ C_1^2 \gamma^2 m \sum_{k=t-m}^{t-1} \mathbb{E} \left\| \tilde{\mathbf{W}} (\mathbb{E} \, \mathbf{G}^{(k)} - \mathbf{G}^{(k)}) \right\|_F^2$$

$$\leq \left\| \mathbf{W}^m \mathbf{Y}^{(t-m)} - \gamma \sum_{k=t-m}^{t-1} \mathbf{W}^{t-1-k} \tilde{\mathbf{W}} \, \mathbb{E} \, \mathbf{G}^{(k)} - \bar{\mathbf{Y}}^{(t-m)} \right\|_F^2$$

$$+ C_1^2 \gamma^2 m \sum_{k=t-m}^{t-1} 3(L^2 \Xi_k + 2Le_k + \bar{\sigma}^2)$$

Separate the first term as

$$n\Xi_t \leq (1+\alpha) \left\| \mathbf{W}^m \mathbf{Y}^{(t-m)} - \bar{\mathbf{Y}}^{(t-m)} \right\|_F^2 + \left(1 + \frac{1}{\alpha}\right) \left\| \gamma \sum_{k=t-m}^{t-1} \mathbf{W}^{t-1-k} \tilde{\mathbf{W}} \, \mathbb{E} \, \mathbf{G}^{(k)} \right\|_F^2$$

$$+ C_1^2 \gamma^2 m \sum_{k=t-m}^{t-1} 3(L^2 \Xi_k + 2Le_k + \bar{\sigma}^2)$$

$$\leq (1+\alpha)(1-p)^{2m} \left\| \mathbf{Y}^{(t-m)} - \bar{\mathbf{Y}}^{(t-m)} \right\|_F^2 + \left(1 + \frac{1}{\alpha}\right) \left\| \gamma \sum_{k=t-m}^{t-1} \mathbf{W}^{t-1-k} \tilde{\mathbf{W}} \, \mathbb{E} \, \mathbf{G}^{(k)} \right\|_F^2$$

$$+ C_1^2 \gamma^2 m \sum_{k=t-m}^{t-1} 3(L^2 \Xi_k + 2Le_k + \bar{\sigma}^2)$$

where the first inequality uses $(a+b)^2 \leq (1+\varepsilon)a^2 + (1+\frac{1}{\varepsilon})b^2$ and take $\varepsilon = (\frac{2-p}{2-2p})^{2m} - 1$.

$$1 + \tfrac{1}{\varepsilon} \leq 1 + \tfrac{1-p}{mp} \leq 1 + \tfrac{1}{mp} \leq \tfrac{2}{p}.$$

Then by applying our key lemma (Lemma 1) we have

$$n\Xi_t \leq \left(1 - \frac{p}{2}\right)^{2m} \left\| \mathbf{Y}^{(t-m)} - \bar{\mathbf{Y}}^{(t-m)} \right\|_F^2 + \frac{2m}{p} C_1^2 \gamma^2 \sum_{k=t-m}^{t-1} \left\| \tilde{\mathbf{W}} \, \mathbb{E} \, \mathbf{G}^{(k)} \right\|_F^2$$

$$+ C_1^2 \gamma^2 m \sum_{k=t-m}^{t-1} 3(L^2 \Xi_k + 2Le_k + \bar{\sigma}^2)$$

Next we bound $\mathbb{E}\|\tilde{\mathbf{W}}\mathbf{G}^{(t')}\|_F^2$,

$$
\begin{aligned}
\mathbb{E}\|\tilde{\mathbf{W}}\,\mathbb{E}\,\mathbf{G}^{(k)}\|_F^2 &= \sum_{i=1}^n \mathbb{E}\|\tfrac{1}{n}\sum_{j=1}^n \nabla f_j(\mathbf{x}_j^{(k-\tau_{ij})})\|^2 \\
&= \sum_{i=1}^n \mathbb{E}\|\tfrac{1}{n}\sum_{j=1}^n (\nabla f_j(\mathbf{x}_j^{(k-\tau_{ij})}) - \nabla f_j(\bar{\mathbf{x}}^{(k)}) + \nabla f_j(\bar{\mathbf{x}}^{(k)}) - \nabla f_j(\mathbf{x}^\star))\|^2 \\
&\leq \tfrac{2}{n}\sum_{i=1}^n \sum_{j=1}^n (\|\nabla f_j(\mathbf{x}_j^{(k-\tau_{ij})}) - \nabla f_j(\bar{\mathbf{x}}^{(k)})\|^2 + \|\nabla f_j(\bar{\mathbf{x}}^{(k)}) - \nabla f_j(\mathbf{x}^\star))\|^2) \\
&\leq \tfrac{2}{n}\sum_{i=1}^n \sum_{j=1}^n (L^2\|\mathbf{x}_j^{(k-\tau_{ij})} - \bar{\mathbf{x}}^{(k)}\|^2 + \|\nabla f_j(\bar{\mathbf{x}}^{(k)}) - \nabla f_j(\mathbf{x}^\star))\|^2) \\
&\overset{\text{Lemma 13}}{\leq} 2L^2 n\Xi_k + 2\sum_{j=1}^n \|\nabla f_j(\bar{\mathbf{x}}^{(k)}) - \nabla f_j(\mathbf{x}^\star)\|^2 \\
&\overset{\text{Smoothness (5)}}{\leq} 2L^2 n\Xi_k + 4nLe_k.
\end{aligned}
$$

Then

$$
n\Xi_t \leq (1-\tfrac{p}{2})^{2m} n\Xi_{t-m} + \frac{2m}{p}C_1^2\gamma^2 \sum_{k=t-m}^{t-1}(2L^2 n\Xi_k + 4nLe_k) + C_1^2\gamma^2 m \sum_{k=t-m}^{t-1} 3(L^2\Xi_k + 2Le_k + \bar{\sigma}^2)
$$

Then

$$
\Xi_t \leq (1-\tfrac{p}{2})^{2m}\Xi_{t-m} + \frac{2m}{p}C_1^2\gamma^2 \sum_{k=t-m}^{t-1}(5L^2\Xi_k + 10Le_k) + 3C_1^2\gamma^2 m^2 \frac{\bar{\sigma}^2}{n}.
$$

**Unroll for $t < m$.** We can apply similar steps

$$
\begin{aligned}
n\Xi_t &\leq \mathbb{E}\left\|\mathbf{W}^{(t)}\mathbf{Y}^{(0)} - \gamma\sum_{k=0}^{t-1}\mathbf{W}^{t-1-k}\tilde{\mathbf{W}}\mathbf{G}^{(k)} - \bar{\mathbf{Y}}^{(0)}\right\|_F^2 = \mathbb{E}\left\|\gamma\sum_{k=0}^{t-1}\mathbf{W}^{t-1-k}\tilde{\mathbf{W}}\mathbf{G}^{(k)}\right\|_F^2 \\
&\leq C_1^2\gamma^2 m\sum_{k=0}^{t-1}\mathbb{E}\left\|\tilde{\mathbf{W}}\mathbf{G}^{(k)}\right\|_F^2 \leq 2C_1^2\gamma^2 m\sum_{k=0}^{t-1}(5L^2 n\Xi_k + 10nLe_k + 3\bar{\sigma}^2)
\end{aligned}
$$

**Merge two parts together and sum over $t$.**

$$
\begin{aligned}
\frac{1}{T+1}\sum_{t=0}^T \Xi_t &\leq \left(1-\frac{p}{2}\right)^{2m}\frac{1}{T+1}\sum_{t=m}^T \Xi_{t-m} + 6C_1^2\gamma^2 m^2\frac{\bar{\sigma}^2}{n} \\
&\quad + \frac{2m}{p}C_1^2\gamma^2\frac{1}{T+1}\left(\sum_{t=m}^T\sum_{k=t-m}^{t-1}(5L^2\Xi_k + 10Le_k) + \sum_{t=0}^{m-1}\sum_{k=t-m}^{t-1}(5L^2\Xi_k + 10Le_k)\right) \\
&\leq \left(1-\frac{p}{2}\right)^{2m}\frac{1}{T+1}\sum_{t=0}^T \Xi_t + 6C_1^2\gamma^2 m^2\frac{\bar{\sigma}^2}{n} + \frac{2m^2}{p}C_1^2\gamma^2\frac{1}{T+1}\sum_{t=0}^T(5L^2\Xi_t + 10Le_t)
\end{aligned}
$$

By taking $\gamma \leq \frac{p}{10CLm}$, then $\frac{10L^2 m^2}{p}C_1^2\gamma^2 \leq \frac{p}{4}$.

$$
\frac{1}{T+1}\sum_{t=0}^T \Xi_t \leq C_1^2\gamma^2 m^2\frac{24}{p}\frac{\bar{\sigma}^2}{n} + \frac{80Lm^2}{p^2}C_1^2\gamma^2\frac{1}{T+1}\sum_{t=0}^T e_t. \qquad \square
$$

**Lemma 16** (Identical to [16, Lemma 15]). *For any parameters $r_0 \geq 0, a \geq 0, b \geq 0, c \geq 0$ there exists constant stepsizes $\gamma \leq \frac{1}{c}$ such that*

$$
\Psi_T := \frac{r_0}{\gamma(T+1)} + a\gamma + b\gamma^2 \leq 2\left(\frac{ar_0}{T+1}\right)^{\frac{1}{2}} + 2b^{\frac{1}{3}}\left(\frac{r_0}{T+1}\right)^{\frac{2}{3}} + \frac{cr_0}{T+1}.
$$

**Theorem V.** *If $\gamma \leq \frac{p}{30LmC_1}$, then*

$$
\frac{1}{T+1}\sum_{t=0}^T \left(f(\bar{\mathbf{x}}^{(t)}) - f(\mathbf{x}^\star)\right) \leq 8\left(\frac{\bar{\sigma}^2 r_0}{n(T+1)}\right)^{\frac{1}{2}} + 2\left(\frac{16Cm\sqrt{L}\bar{\sigma}r_0}{\sqrt{p}(T+1)}\right)^{\frac{2}{3}} + \frac{30Lm\sqrt{n}Cr_0}{p(T+1)}.
$$

*where $r_0 = \|\mathbf{x}^0 - \mathbf{x}^\star\|^2$ and $C = C(\mathbf{W})$ is defined in Definition G.*

*Proof.* Reorganize Lemma 14 and average over time

$$\frac{1}{T+1}\sum_{t=0}^{T} e_t \le \frac{1}{T+1}\sum_{t=0}^{T}\left(\frac{r_t}{\gamma n\pi_0} - \frac{r_{t+1}}{\gamma n\pi_0}\right) + \frac{4L}{T+1}\sum_{t=0}^{T}\Xi_t + 3\gamma\pi_0\bar{\sigma}^2.$$

Combining with Lemma 15 gives

$$\frac{1}{T+1}\sum_{t=0}^{T} e_t \le \frac{1}{T+1}\frac{r_0}{\gamma n\pi_0} + 4L\left(C_1^2\gamma^2 m^2\frac{24}{p}\frac{\bar{\sigma}^2}{n} + \frac{80Lm^2}{p^2}C_1^2\gamma^2\frac{1}{T+1}\sum_{t=0}^{T} e_k\right) + 3\gamma\pi_0\bar{\sigma}^2$$

Select $\gamma \le \frac{p}{30LmC_1}$ such that $\frac{320L^2}{p^2}\gamma^2 m^2 C_1^2 \le \frac{1}{2}$

$$\frac{1}{T+1}\sum_{t=0}^{T} e_t \le \frac{2}{T+1}\frac{r_0}{\gamma n\pi_0} + 6\gamma\pi_0\bar{\sigma}^2 + \frac{96L}{p}\gamma^2 m^2 C_1^2\frac{\bar{\sigma}^2}{n}.$$

Applying Lemma 16 gives

$$\frac{1}{T+1}\sum_{t=0}^{T} e_t \le 40\left(\frac{\bar{\sigma}^2 r_0}{n(T+1)}\right)^{\frac{1}{2}} + 2\left(\frac{\sqrt{mL}\bar{\sigma}r_0}{\sqrt{p}(T+1)}\frac{16C_1\sqrt{m}}{n\pi_0\sqrt{n}}\right)^{\frac{2}{3}} + \frac{dr_0}{n\pi_0(T+1)}$$

where $d = \max\{\frac{30LmC_1}{p}, 10Ln\pi_0\} = \frac{30LmC_1}{p}$. As in Lemma 8,

$$C_1 = C\|\mathbf{1}\boldsymbol{\pi}^\top\tilde{\mathbf{I}}\| = Cn\sqrt{\tau_{\max}+1}\pi_0 \le Cn\sqrt{n}\pi_0.$$

We can further simplify it as

$$\frac{1}{T+1}\sum_{t=0}^{T} e_t \le 40\left(\frac{\bar{\sigma}^2 r_0}{n(T+1)}\right)^{\frac{1}{2}} + 2\left(\frac{16Cm\sqrt{L}\bar{\sigma}r_0}{\sqrt{p}(T+1)}\right)^{\frac{2}{3}} + \frac{30Lm\sqrt{n}Cr_0}{p(T+1)} \quad\square$$

### A.5 Proof of Theorem I in the strongly convex case

The proof for strongly convex objective follows similar lines as [35]:

**Theorem VI.** *Let* $a = \frac{\mu n\pi_0}{2}$, $b = \frac{2}{n\pi_0}$, $c = 6\pi_0\bar{\sigma}^2$, $A = \frac{400L}{p^2}m^2 C_1^2\bar{\sigma}^2$, *and let* $\gamma = \frac{1}{s} \le \frac{1}{aT}\ln\max\{\frac{ba^2T^2r_0}{c}, 2\}$, *then*

$$\frac{1}{W_T}\sum_{t=0}^{T} w_t e_t + \mu r_{T+1} \le \tilde{\mathcal{O}}\left(bsr_0\exp\left[-\frac{a(T+1)}{s}\right] + \frac{c}{a(T+1)} + \frac{A}{a^2(T+1)^2}\right)$$

*where* $w_t = (1 - \frac{\mu\gamma n\pi_0}{2})^{-(t+1)}$.

*Proof.* From Lemma 14 we know that if $\gamma \le \frac{1}{10Ln\pi_0}$, then

$$r_{t+1} \le (1 - \frac{\gamma\mu n\pi_0}{2})r_t - \gamma n\pi_0 e_t + 4\gamma Ln\pi_0\Xi_t + 3\gamma^2 n\pi_0^2\bar{\sigma}^2.$$

Then

$$e_t \le \frac{1}{\gamma n\pi_0}(1 - \frac{\mu\gamma n\pi_0}{2})r_t - \frac{1}{\gamma n\pi_0}r_{t+1} + 4L\Xi_t + 3\gamma\pi_0\bar{\sigma}^2.$$

Multiply $w_t$ and sum over $t = 0$ to $T$ and divided by $W_T$

$$\frac{1}{W_T}\sum_{t=0}^{T} w_t e_t \le \frac{1}{W_T}\sum_{t=0}^{T}\left(\frac{1 - \frac{\mu\gamma n\pi_0}{2}}{\gamma n\pi_0}w_t r_t - \frac{w_t}{\gamma n\pi_0}r_{t+1}\right) + \frac{4L}{W_T}\sum_{t=0}^{T} w_t\Xi_t + 3\gamma\pi_0\bar{\sigma}^2.$$

Set $(1 - \frac{\mu\gamma n\pi_0}{2})w_{t+1} = w_t$, then

$$\frac{1}{W_T}\sum_{t=0}^{T} w_t e_t \le \frac{1}{W_T}\left(\frac{1 - \frac{\mu\gamma n\pi_0}{2}}{\gamma n\pi_0}w_0 r_0 - \frac{1 - \frac{\mu\gamma n\pi_0}{2}}{\gamma n\pi_0}w_{T+1}r_{T+1}\right) + \frac{4L}{W_T}\sum_{t=0}^{T} w_t\Xi_t + 3\gamma\pi_0\bar{\sigma}^2.$$

Then using Lemma 15 we have

$$\frac{1}{W_T}\sum_{t=0}^{T} w_t e_t + \frac{1 - \frac{\mu\gamma n\pi_0}{2}}{\gamma n\pi_0 W_T} w_{T+1} r_{T+1}$$

$$\leq \frac{1}{W_T}\frac{1 - \frac{\mu\gamma n\pi_0}{2}}{\gamma n\pi_0} w_0 r_0 + 4L\left(\frac{80C_1^2 Lm^2}{p^2}\gamma^2\frac{1}{W_T}\sum_{t'=0}^{T} w_t \mathbf{e}_{t'} + \frac{24}{p}\gamma^2 m^2 C_1^2\frac{\bar{\sigma}^2}{n}\right) + 3\gamma\pi_0\bar{\sigma}^2$$

By taking $\gamma \leq \frac{p}{30LmC_1}$ we have $\frac{320L^2 m^2 C_1^2\gamma^2}{p^2} \leq \frac{1}{2}$, then

$$\frac{1}{W_T}\sum_{t=0}^{T} w_t e_t + \frac{1 - \frac{\mu\gamma n\pi_0}{2}}{\gamma n\pi_0 W_T} 2w_{T+1} r_{T+1} \leq \frac{1}{W_T}\frac{1 - \frac{\mu\gamma n\pi_0}{2}}{\gamma n\pi_0} 2w_0 r_0 + 6\gamma\pi_0\bar{\sigma}^2 + \frac{400L}{p^2}\gamma^2 m^2 C_1^2\bar{\sigma}^2$$

Since $W_T \geq w_T = (1 - \frac{\mu\gamma n\pi_0}{2})^{-(T+1)}$ and $W_T \leq \frac{2w_T}{\mu\gamma n\pi_0}$

$$\frac{1}{W_T}\sum_{t=0}^{T} w_t e_t + \mu r_{T+1} \leq \frac{(1 - \frac{\mu\gamma n\pi_0}{2})^{T+1}}{\gamma n\pi_0} 2w_0 r_0 + 6\gamma\pi_0\bar{\sigma}^2 + \frac{400L}{p^2}\gamma^2 m^2 C_1^2\bar{\sigma}^2$$

$$\leq \frac{e^{-\frac{\mu\gamma n\pi_0}{2}(T+1)}}{\gamma n\pi_0} 2w_0 r_0 + 6\gamma\pi_0\bar{\sigma}^2 + \frac{400L}{p^2}\gamma^2 m^2 C_1^2\bar{\sigma}^2$$

Let $a = \frac{\mu n\pi_0}{2}$, $b = \frac{2}{n\pi_0}$, $c = 6\pi_0\bar{\sigma}^2$, $A = \frac{400L}{p^2}m^2 C_1^2\bar{\sigma}^2$, then

$$\frac{1}{W_T}\sum_{t=0}^{T} w_t e_t + \mu r_{T+1} \leq \frac{br_0}{\gamma}\exp[-a\gamma(T+1)] + c\gamma + A\gamma^2$$

**Tuning stepsize.** Let $\gamma = \frac{1}{d} \leq \frac{1}{aT}\ln\max\{\frac{ba^2 T^2 r_0}{c}, 2\}$, then

$$\frac{1}{W_T}\sum_{t=0}^{T} w_t e_t + \mu r_{T+1} \leq \tilde{\mathcal{O}}\left(bsr_0\exp[-\frac{a(T+1)}{s}] + \frac{c}{a(T+1)} + \frac{A}{a^2(T+1)^2}\right). \qquad \square$$

### A.6 Proof of Theorem I in the non-convex case

Let $\bar{\mathbf{x}}^{(t)} := (\boldsymbol{\pi}^\top \mathbf{Y}^{(t)})^\top$ and $\bar{\mathbf{Y}}^{(t)} := \mathbf{1}\boldsymbol{\pi}^\top \mathbf{Y}^{(t)}$. Let $f^\star$ be the optimal objective value at critical points. We can define the following iterates

1. $r_t := \mathbb{E} f(\bar{\mathbf{x}}^{(t)}) - f^\star$ is the *expected function suboptimality*.
2. $e_t := \|\nabla f(\bar{\mathbf{x}}^{(t)})\|^2$
3. $\Xi_t := \frac{1}{n}\|\bar{\mathbf{Y}}^{(t)} - \mathbf{Y}^{(t)}\|_F^2$ is the *consensus distance*.

where the expectation is taken with respect to $\boldsymbol{\xi}^{(t)} \in \mathbb{R}^n$ the randomness across all workers at time $t$. Note that Lemma 13 still holds.

Proposition VII and Proposition VIII bound the stochastic noise of the gradient.

**Proposition VII.** *Under Assumption B, we have*

$$\mathbb{E}\|\boldsymbol{\pi}^\top\tilde{\mathbf{W}}(\mathbf{G}^{(t)} - \mathbb{E}\,\mathbf{G}^{(t)})\|^2 \leq n\pi_0^2\bar{\sigma}^2. \tag{9}$$

*Proof.* Denote $\mathbb{E} = \mathbb{E}_\xi$. Use Cauchy-Schwartz inequality Equation (4)

$$\mathbb{E}\|\boldsymbol{\pi}^\top\tilde{\mathbf{W}}(\mathbf{G}^{(t)} - \mathbb{E}\,\mathbf{G}^{(t)})\|^2 = \mathbb{E}\left\|\frac{\pi_0}{n}\sum_{i=1}^{n}\sum_{j=1}^{n}(\nabla F_j(\mathbf{x}_j^{(t-\tau_{ij})};\xi_j^{(t-\tau_{ij})}) - \nabla f_j(\mathbf{x}_j^{(t-\tau_{ij})}))\right\|^2$$

$$\leq \frac{\pi_0^2}{n}\sum_{i=1}^{n}\mathbb{E}\left\|\sum_{j=1}^{n}\nabla F_j(\mathbf{x}_j^{(t-\tau_{ij})};\xi_j^{(t-\tau_{ij})}) - \nabla f_j(\mathbf{x}_j^{(t-\tau_{ij})})\right\|^2$$

Now the randomness inside the norm are independent

$$\mathbb{E}\|\boldsymbol{\pi}^\top\tilde{\mathbf{W}}(\mathbf{G}^{(t)} - \mathbb{E}\,\mathbf{G}^{(t)})\|^2 \mathbb{E}\|\boldsymbol{\pi}^\top\tilde{\mathbf{W}}(\mathbf{G}^{(t)} - \mathbb{E}\,\mathbf{G}^{(t)})\|^2 \leq n\pi_0^2\bar{\sigma}^2. \qquad \square$$

**Proposition VIII.** *Under Assumption B, we have*

$$\mathbb{E}\|\tilde{\mathbf{W}}(\mathbf{G}^{(t)} - \mathbb{E}\,\mathbf{G}^{(t)})\|_F^2 \le \bar{\sigma}^2. \tag{10}$$

Next we establish the recursion of $r_t$

**Lemma 17** (Descent lemma for non-convex case). *Under Assumption A and B. Let $\gamma \le \frac{1}{8Ln\pi_0}$, then*

$$r_{t+1} \le r_t - \frac{\gamma n\pi_0}{4}e_t + 2\gamma L^2 n\pi_0 \Xi_t + 2\gamma^2 Ln\pi_0^2 \bar{\sigma}^2.$$

*Proof.* Since $f$ is $L$-smooth,

$$\mathbb{E}\,f(\bar{\mathbf{x}}^{(t+1)}) = \mathbb{E}\,f(\bar{\mathbf{x}}^{(t)} - \gamma\boldsymbol{\pi}^\top\tilde{\mathbf{W}}\mathbf{G}^{(t)})$$
$$\le f(\bar{\mathbf{x}}^{(t)}) - \gamma\underbrace{\langle\nabla f(\bar{\mathbf{x}}^{(t)}), \boldsymbol{\pi}^\top\tilde{\mathbf{W}}\,\mathbb{E}\,\mathbf{G}^{(t)}\rangle}_{:=T_1} + \tfrac{\gamma^2 L}{2}\underbrace{\mathbb{E}\|\boldsymbol{\pi}^\top\tilde{\mathbf{W}}\mathbf{G}^{(t)}\|^2}_{:=T_2}$$

The first-order term $T_1$ has a lower bound

$$T_1 = n\pi_0\langle\nabla f(\bar{\mathbf{x}}^{(t)}), \tfrac{1}{n\pi_0}\boldsymbol{\pi}^\top\tilde{\mathbf{W}}\,\mathbb{E}\,\mathbf{G}^{(t)}\rangle$$
$$= n\pi_0\left(\|\nabla f(\bar{\mathbf{x}}^{(t)})\|^2 + \langle\nabla f(\bar{\mathbf{x}}^{(t)}), \tfrac{1}{n\pi_0}\boldsymbol{\pi}^\top\tilde{\mathbf{W}}\,\mathbb{E}\,\mathbf{G}^{(t)} - \nabla f(\bar{\mathbf{x}}^{(t)})\rangle\right)$$
$$\ge n\pi_0\left(\tfrac{1}{2}\|\nabla f(\bar{\mathbf{x}}^{(t)})\|^2 - \tfrac{1}{2}\|\tfrac{1}{n\pi_0}\boldsymbol{\pi}^\top\tilde{\mathbf{W}}\,\mathbb{E}\,\mathbf{G}^{(t)} - \nabla f(\bar{\mathbf{x}}^{(t)})\|^2\right)$$
$$= n\pi_0\left(\tfrac{1}{2}e_t - \tfrac{1}{2n^4}\|\textstyle\sum_{i=1}^n\sum_{j=1}^n(\nabla f_j(\mathbf{x}_j^{(t-\tau_{ij})}) - \nabla f_j(\bar{\mathbf{x}}^{(t)}))\|^2\right)$$
$$\ge n\pi_0\left(\tfrac{1}{2}e_t - \tfrac{L^2}{2n^2}\textstyle\sum_{i=1}^n\sum_{j=1}^n\|\mathbf{x}_j^{(t-\tau_{ij})} - \bar{\mathbf{x}}^{(t)}\|^2\right)$$
$$\ge n\pi_0\left(\tfrac{1}{2}e_t - \tfrac{L^2}{2}\Xi_t\right)$$

as $a^2 - \langle a, b\rangle \ge \frac{a^2}{2} - \frac{b^2}{2}$ for $a, b \ge 0$.

On the other hand, separate the stochastic part and deterministic part of $T_2$ we have

$$T_2 \le 2\,\mathbb{E}\|\boldsymbol{\pi}^\top\tilde{\mathbf{W}}(\mathbf{G}^{(t)} - \mathbb{E}\,\mathbf{G}^{(t)})\|^2 + 2\|\boldsymbol{\pi}^\top\tilde{\mathbf{W}}\,\mathbb{E}\,\mathbf{G}^{(t)}\|^2.$$

Under Assumption B and Proposition VII, we know the first term

$$\mathbb{E}\|\boldsymbol{\pi}^\top\tilde{\mathbf{W}}(\mathbf{G}^{(t)} - \mathbb{E}\,\mathbf{G}^{(t)})\|^2 \le n\pi_0^2\bar{\sigma}^2.$$

Consider the second term

$$\|\boldsymbol{\pi}^\top\tilde{\mathbf{W}}\,\mathbb{E}\,\mathbf{G}^{(t)}\|^2 = \left\|\frac{\pi_0}{n}\sum_{i=1}^n\sum_{j=1}^n\nabla f_j(\mathbf{x}_j^{(t-\tau_{ij})})\right\|^2$$
$$= n^2\pi_0^2\left\|\frac{1}{n^2}\sum_{i=1}^n\sum_{j=1}^n\nabla f_j(\mathbf{x}_j^{(t-\tau_{ij})}) - \nabla f(\bar{\mathbf{x}}^{(t)}) + \nabla f(\bar{\mathbf{x}}^{(t)})\right\|^2$$
$$\le 2n^2\pi_0^2\left\|\frac{1}{n^2}\sum_{i=1}^n\sum_{j=1}^n(\nabla f_j(\mathbf{x}_j^{(t-\tau_{ij})}) - \nabla f_j(\bar{\mathbf{x}}^{(t)}))\right\|^2 + 2n^2\pi_0^2\left\|\nabla f(\bar{\mathbf{x}}^{(t)})\right\|^2$$
$$\le 2\pi_0^2\sum_{i=1}^n\sum_{j=1}^n\left\|\nabla f_j(\mathbf{x}_j^{(t-\tau_{ij})}) - \nabla f_j(\bar{\mathbf{x}}^{(t)})\right\|^2 + 2n^2\pi_0^2\left\|\nabla f(\bar{\mathbf{x}}^{(t)})\right\|^2$$

Combine Assumption B we have

$$\|\boldsymbol{\pi}^\top\tilde{\mathbf{W}}\,\mathbb{E}\,\mathbf{G}^{(t)}\|^2 \le 2n^2\pi_0^2(L^2\Xi_t + e_t).$$

Therefore, the $T_2$ can be bounded as follows

$$T_2 \le 4n^2\pi_0^2(\tfrac{\bar{\sigma}^2}{n} + L^2\Xi_t + e_t). \tag{11}$$

Gathering everything together

$$r_{t+1} \leq r_t - \frac{\gamma n \pi_0}{2}(e_t - L^2\Xi_t) + 2\gamma^2 Ln^2\pi_0^2(\frac{\bar{\sigma}^2}{n} + L^2\Xi_t + e_t)$$
$$\leq r_t - \frac{\gamma n \pi_0}{2}(1 - 4\gamma Ln\pi_0)e_t + \gamma L^2 n\pi_0(1 + 2\gamma Ln\pi_0)\Xi_t + 2\gamma^2 Ln\pi_0^2\bar{\sigma}^2$$

Let $\gamma \leq \frac{1}{8Ln\pi_0}$, then

$$r_{t+1} \leq r_t - \frac{\gamma n\pi_0}{4}e_t + 2\gamma L^2 n\pi_0\Xi_t + 2\gamma^2 Ln\pi_0^2\bar{\sigma}^2. \qquad \square$$

Next we bound the consensus distance

**Lemma 18** (Bounded consensus distance). *Under Assumption B,*

$$\frac{1}{T+1}\sum_{t=0}^{T}\Xi_t \leq \frac{16C^2m^2}{p^2}\gamma^2\bar{\sigma}^2 + \frac{16C^2m^2}{p^2}\gamma^2\frac{1}{T+1}\sum_{t=0}^{T}e_k\,.$$

*Proof.* First bound the consensus distance by inserting $\bar{\mathbf{Y}}^{(t-m)}$

$$n\Xi_t = \mathbb{E}\|\bar{\mathbf{Y}}^{(t)} - \mathbf{Y}^{(t)}\|_F^2 \leq \mathbb{E}\|(\bar{\mathbf{Y}}^{(t)} - \bar{\mathbf{Y}}^{(t-m)}) - (\mathbf{Y}^{(t)} - \bar{\mathbf{Y}}^{(t-m)})\|_F^2$$
$$\leq \mathbb{E}\|\mathbf{Y}^{(t)} - \bar{\mathbf{Y}}^{(t-m)}\|_F^2$$

where we used $\|A - \bar{A}\|_F^2 = \sum_{i=1}^{n}\|\mathbf{a}_i - \bar{\mathbf{a}}\|^2 \leq \sum_{i=1}^{n}\|\mathbf{a}_i\|^2 = \|A\|_F^2$.

For $t \geq m$ unroll $\mathbf{Y}^{(t)}$ until $t - m$.

$$n\Xi_t \leq \mathbb{E}\left\|\mathbf{W}^m\mathbf{Y}^{(t-m)} - \gamma\sum_{k=t-m}^{t-1}\mathbf{W}^{t-1-k}\tilde{\mathbf{W}}\mathbf{G}^{(k)} - \bar{\mathbf{Y}}^{(t-m)}\right\|_F^2$$

Separate stochastic part and deterministic part

$$n\Xi_t \leq \left\|\mathbf{W}^m\mathbf{Y}^{(t-m)} - \gamma\sum_{k=t-m}^{t-1}\mathbf{W}^{t-1-k}\tilde{\mathbf{W}}\,\mathbb{E}\,\mathbf{G}^{(k)} - \bar{\mathbf{Y}}^{(t-m)}\right\|_F^2$$
$$+ \mathbb{E}\left\|\gamma\sum_{k=t-m}^{t-1}\mathbf{W}^{t-1-k}\tilde{\mathbf{W}}(\mathbb{E}\,\mathbf{G}^{(k)} - \mathbf{G}^{(k)})\right\|_F^2$$

then let $C_1^2$ defined in Definition G and use $\|AB\|_F^2 \leq \|A\|_F^2\|B\|^2$ and (10)

$$n\Xi_t \leq \left\|\mathbf{W}^m\mathbf{Y}^{(t-m)} - \gamma\sum_{k=t-m}^{t-1}\mathbf{W}^{t-1-k}\tilde{\mathbf{W}}\,\mathbb{E}\,\mathbf{G}^{(k)} - \bar{\mathbf{Y}}^{(t-m)}\right\|_F^2$$
$$+ C_1^2\gamma^2 m\sum_{k=t-m}^{t-1}\mathbb{E}\left\|\tilde{\mathbf{W}}(\mathbb{E}\,\mathbf{G}^{(k)} - \mathbf{G}^{(k)})\right\|_F^2$$
$$\leq \left\|\mathbf{W}^m\mathbf{Y}^{(t-m)} - \gamma\sum_{k=t-m}^{t-1}\mathbf{W}^{t-1-k}\tilde{\mathbf{W}}\,\mathbb{E}\,\mathbf{G}^{(k)} - \bar{\mathbf{Y}}^{(t-m)}\right\|_F^2 + C_1^2\gamma^2 m^2\bar{\sigma}^2$$

Apply Cauchy-Schwartz inequality with $\alpha > 0$

$$n\Xi_t \leq (1+\alpha)\left\|\mathbf{W}^m\mathbf{Y}^{(t-m)} - \bar{\mathbf{Y}}^{(t-m)}\right\|_F^2 + (1+\tfrac{1}{\alpha})\left\|\gamma\sum_{k=t-m}^{t-1}\mathbf{W}^{t-1-k}\tilde{\mathbf{W}}\,\mathbb{E}\,\mathbf{G}^{(k)}\right\|_F^2 + C_1^2\gamma^2 m^2\bar{\sigma}^2$$

Applying Lemma 1 to the first term

$$n\Xi_t \leq (1+\alpha)(1-p)^{2m}\|\mathbf{Y}^{(t-m)} - \bar{\mathbf{Y}}^{(t-m)}\|_F^2 + (1+\tfrac{1}{\alpha})\left\|\gamma\sum_{k=t-m}^{t-1}\mathbf{W}^{t-1-k}\tilde{\mathbf{W}}\,\mathbb{E}\,\mathbf{G}^{(k)}\right\|_F^2 + C_1^2\gamma^2 m^2\bar{\sigma}^2$$

Take $\alpha = (\frac{2-p}{2-2p})^{2m} - 1 = (1 + \frac{p}{2-2p})^{2m} - 1 \geq \frac{mp}{1-p}$ and use

$$1 + \frac{1}{\alpha} \leq 1 + \frac{1-p}{mp} \leq 1 + \frac{1}{mp} \leq \frac{2}{p},$$

then use $\|AB\|_F^2 \leq \|A\|_F^2 \|B\|^2$

$$n\Xi_t \leq \left(1 - \frac{p}{2}\right)^{2m} \|\mathbf{Y}^{(t-m)} - \bar{\mathbf{Y}}^{(t-m)}\|_F^2 + \frac{2}{p}\left\|\gamma \sum_{k=t-m}^{t-1} \mathbf{W}^{t-1-k}\tilde{\mathbf{W}}\,\mathbb{E}\,\mathbf{G}^{(k)}\right\|_F^2 + C_1^2\gamma^2 m^2\bar{\sigma}^2$$

$$\leq \left(1 - \frac{p}{2}\right)^{2m} \|\mathbf{Y}^{(t-m)} - \bar{\mathbf{Y}}^{(t-m)}\|_F^2 + \frac{2C_1^2 m}{p}\gamma^2 \sum_{k=t-m}^{t-1}\left\|\tilde{\mathbf{W}}\,\mathbb{E}\,\mathbf{G}^{(k)}\right\|_F^2 + C_1^2\gamma^2 m^2\bar{\sigma}^2.$$

where the second term can be expanded by

$$\|\tilde{\mathbf{W}}\,\mathbb{E}\,\mathbf{G}^{(k)}\|_F^2 = \sum_{i=1}^n \left\|\frac{1}{n}\sum_{j=1}^n \nabla f_j(\mathbf{x}_j^{(k-\tau_{ij})})\right\|^2$$

$$= \sum_{i=1}^n \left\|\frac{1}{n}\sum_{j=1}^n \nabla f_j(\mathbf{x}_j^{(k-\tau_{ij})}) - \nabla f(\bar{\mathbf{x}}^{(k)}) + \nabla f(\bar{\mathbf{x}}^{(k)})\right\|^2$$

$$\leq 2\sum_{i=1}^n \left\|\frac{1}{n}\sum_{j=1}^n (\nabla f_j(\mathbf{x}_j^{(k-\tau_{ij})}) - \nabla f_j(\bar{\mathbf{x}}^{(k)}))\right\|^2 + 2n\left\|\nabla f(\bar{\mathbf{x}}^{(k)})\right\|^2$$

$$\leq \frac{2}{n}\sum_{i=1}^n \sum_{j=1}^n \left\|\nabla f_j(\mathbf{x}_j^{(k-\tau_{ij})}) - \nabla f_j(\bar{\mathbf{x}}^{(k)})\right\|^2 + 2n\left\|\nabla f(\bar{\mathbf{x}}^{(k)})\right\|^2$$

$$\leq 2nL^2\Xi_k + 2ne_k$$

Combine and reduce the $n$ on both sides

$$\Xi_t \leq \left(1 - \frac{p}{2}\right)^{2m} \Xi_{t-m} + 2C_1^2 m^2\gamma^2 \frac{\bar{\sigma}^2}{n} + \frac{4C_1^2 m}{p}\gamma^2 \sum_{k=t-m}^{t-1} (L^2\Xi_k + e_k).$$

**Unroll for $t < m$.** For $t < m$, we can apply similar steps

$$n\Xi_t \leq \mathbb{E}\left\|\mathbf{W}^{(t)}\mathbf{Y}^{(0)} - \gamma\sum_{k=0}^{t-1}\mathbf{W}^{t-1-k}\tilde{\mathbf{W}}\mathbf{G}^{(k)} - \bar{\mathbf{Y}}^{(0)}\right\|_F^2 = \mathbb{E}\left\|\gamma\sum_{k=0}^{t-1}\mathbf{W}^{t-1-k}\tilde{\mathbf{W}}\mathbf{G}^{(k)}\right\|_F^2$$

$$\leq C_1^2\gamma^2 m \sum_{k=0}^{t-1}\mathbb{E}\left\|\tilde{\mathbf{W}}\mathbf{G}^{(k)}\right\|_F^2 \leq 2C_1^2 m\gamma^2 \sum_{k=0}^{t-1}(\bar{\sigma}^2 + nL^2\Xi_k + ne_k).$$

**Finally, sum over $t$**

$$\frac{1}{T+1}\sum_{t=0}^T \Xi_t \leq \left(1 - \frac{p}{2}\right)^{2m}\frac{1}{T+1}\sum_{t=m}^T \Xi_{t-m} + 2C_1^2 m^2\gamma^2\frac{\bar{\sigma}^2}{n}$$

$$+ \frac{4C_1^2 m}{p}\gamma^2\frac{1}{T+1}\left(\sum_{t=m}^T\sum_{k=t-m}^{t-1}(L^2\Xi_k + e_k) + \sum_{t=0}^{m-1}\sum_{k=0}^{t-1}(L^2\Xi_k + e_k)\right)$$

$$\leq \left(1 - \frac{p}{2}\right)^{2m}\frac{1}{T+1}\sum_{t=0}^T \Xi_t + 2C_1^2 m^2\gamma^2\frac{\bar{\sigma}^2}{n} + \frac{4C_1^2 m^2}{p}\frac{\gamma^2}{T+1}\sum_{t=0}^T(L^2\Xi_k + e_k).$$

by taking $\gamma \leq \frac{p}{4CLm}$ we have $\frac{4C_1^2 m^2}{p}\gamma^2 L^2 \leq \frac{p}{4}$, then rearrange the all of the $\Xi$ terms

$$\frac{1}{T+1}\sum_{t=0}^T \Xi_t \leq \frac{16C_1^2 m^2}{p}\frac{\bar{\sigma}^2}{n}\gamma^2 + \frac{16C_1^2 m^2}{p^2}\gamma^2\frac{1}{T+1}\sum_{t=0}^T e_k \qquad \square$$

We can use the lemmas for recursion and the descent in the consensus distance to conclude the following theorem.

**Theorem IX.** *Under Assumption A and Assumption B. For* $\gamma \leq \frac{p}{16C_1Lm}$

$$\frac{1}{T+1}\sum_{t=0}^{T}\|\nabla f(\bar{\mathbf{x}}^{(t)})\|^2 \leq 16\left(\frac{2L\bar{\sigma}^2 r_0}{n(T+1)}\right)^{\frac{1}{2}} + 2\left(\frac{16CLm\bar{\sigma}}{\sqrt{p}}\frac{8r_0}{T+1}\right)^{\frac{2}{3}} + \frac{16C_1Lm}{p}\frac{r_0}{T+1}$$

*where* $C = C(\mathbf{W})$ *is defined in Definition G and* $r_0 = f(\mathbf{x}^{(0)}) - f^\star$. *Alternatively, for any target accuracy* $\varepsilon$, $\frac{1}{T+1}\sum_{t=0}^{T}\|\nabla f(\bar{\mathbf{x}}^{(t)})\|^2 \leq \varepsilon$ *after*

$$\mathcal{O}\left(\frac{\bar{\sigma}^2}{n\varepsilon^2} + \frac{Cm\bar{\sigma}}{\sqrt{p}\varepsilon^{3/2}} + \frac{C_1m}{p\varepsilon}\right)Lr_0$$

*iterations.*

**Remark 19.** *For gossip averaging [16], the rate with* $\zeta^2 = 0$ *is*

$$\mathcal{O}\left(\frac{\bar{\sigma}^2}{n\varepsilon^2} + \frac{\sqrt{m}\bar{\sigma}}{\sqrt{p}\varepsilon^{3/2}} + \frac{m}{p\varepsilon}\right)Lr_0.$$

*Proof.* From Lemma 17 we know that for $\gamma \leq \frac{1}{8Ln\pi_0}$

$$r_{t+1} \leq r_t - \frac{\gamma n\pi_0}{4}e_t + 2\gamma L^2 n\pi_0\Xi_t + 2\gamma^2 Ln\pi_0^2\bar{\sigma}^2.$$

Rearrange the terms and average over $t$

$$\frac{1}{T+1}\sum_{t=0}^{T}e_t \leq \frac{1}{T+1}\sum_{t=0}^{T}(\frac{4r_t}{\gamma n\pi_0} - \frac{4r_{t+1}}{\gamma n\pi_0}) + \frac{8L^2}{T+1}\sum_{t=0}^{T}\Xi_t + 8L\pi_0\gamma\bar{\sigma}^2$$

$$\leq \frac{1}{T+1}\frac{4r_0}{\gamma n\pi_0} + \frac{8L^2}{T+1}\sum_{t=0}^{T}\Xi_t + 8L\pi_0\gamma\bar{\sigma}^2$$

On the other hand, from Lemma 18 for $\gamma \leq \frac{p}{4C_1Lm}$ we have

$$\frac{1}{T+1}\sum_{t=0}^{T}\Xi_t \leq \frac{16C_1^2m^2}{p}\frac{\bar{\sigma}^2}{n}\gamma^2 + \frac{16C_1^2m^2}{p^2}\gamma^2\frac{1}{T+1}\sum_{t=0}^{T}e_k.$$

Then

$$\frac{1}{T+1}\sum_{t=0}^{T}e_t \leq \frac{1}{T+1}\frac{4r_0}{\gamma n\pi_0} + 8L^2\frac{16C_1^2m^2}{p^2}\gamma^2\left(\frac{p\bar{\sigma}^2}{n} + \frac{1}{T+1}\sum_{t=0}^{T}e_k\right) + 8L\pi_0\gamma\bar{\sigma}^2$$

By taking $\gamma \leq \frac{p}{16C_1Lm}$ such that $8L^2\frac{16C_1^2m^2}{p^2}\gamma^2 \leq \frac{1}{2}$, then

$$\frac{1}{T+1}\sum_{t=0}^{T}e_t \leq \frac{1}{T+1}\frac{8r_0}{\gamma n\pi_0} + 16L\pi_0\gamma\bar{\sigma}^2 + \frac{16^2L^2C_1^2m^2}{np}\gamma^2\bar{\sigma}^2$$

Then applying Lemma 16 we have

$$\frac{1}{T+1}\sum_{t=0}^{T}e_t \leq 32\left(\frac{L\bar{\sigma}^2 r_0}{n(T+1)}\right)^{\frac{1}{2}} + 2\left(\frac{16C_1Lm\bar{\sigma}}{\sqrt{np}}\frac{8r_0}{n\pi_0(T+1)}\right)^{\frac{2}{3}} + \frac{dr_0}{T+1}$$

where $d = \max\{\frac{16C_1Lm}{p}, 8Ln\pi_0\} = \frac{16C_1Lm}{p}$. As in Lemma 8,

$$C_1 = C\|\mathbf{1}\boldsymbol{\pi}^\top\tilde{\mathbf{I}}\| = Cn\sqrt{\tau_{\max}+1}\pi_0 \leq Cn\sqrt{n}\pi_0.$$

We can further simplify it as

$$\frac{1}{T+1}\sum_{t=0}^{T}e_t \leq 32\left(\frac{L\bar{\sigma}^2 r_0}{n(T+1)}\right)^{\frac{1}{2}} + 2\left(\frac{16CLm\bar{\sigma}}{\sqrt{p}}\frac{8r_0}{T+1}\right)^{\frac{2}{3}} + \frac{dr_0}{T+1}. \qquad \square$$

Table 6: Default experimental settings for Cifar-10/VGG-11

| | |
|---|---|
| Dataset | Cifar-10 [18] |
| Data augmentation | random horizontal flip and random $32 \times 32$ cropping |
| Architecture | VGG-11 [17] |
| Training objective | cross entropy |
| Evaluation objective | top-1 accuracy |
| Number of workers | 16 |
| Topology | SGP: time-varying exponential, RelaySGD: double binary trees, baselines: best of ring or double binary trees |
| Gossip weights | Metropolis-Hastings (1/3 for ring) |
| Data distribution | Heterogeneous, not shuffled, according to Dirichlet sampling procedure from [20] |
| Batch size | 32 patches per worker |
| Momentum | 0.9 (Nesterov) |
| Learning rate | Tuned c.f. subsection C.1 |
| LR decay | $/10$ at epoch 150 and 180 |
| LR warmup | Step-wise linearly within 5 epochs, starting from 0 |
| # Epochs | 200 |
| Weight decay | $10^{-4}$ |
| Normalization scheme | no normalization layer |
| Repetitions | 3, with varying seeds |
| Reported metric | Worst result of any worker of the worker's mean test accuracy over the last 5 epochs |

# B   Detailed experimental setup

## B.1   Cifar-10

Table 6

## B.2   ImageNet

Table 7

Table 7: Default experimental settings for ImageNet

| | |
|---|---|
| Dataset | ImageNet [5] |
| Data augmentation | random resized crop ($224 \times 224$), random horizontal flip |
| Architecture | ResNet-20-EvoNorm [21, 20] |
| Training objective | cross entropy |
| Evaluation objective | top-1 accuracy |
| Number of workers | 16 |
| Topology | SGP: time-varying exponential, RelaySGD: double binary trees, baselines: best of ring or double binary trees |
| Gossip weights | Metropolis-Hastings (1/3 for ring) |
| Data distribution | Heterogeneous, not shuffled, according to Dirichlet sampling procedure from [20] |
| Batch size | 32 patches per worker |
| Momentum | 0.9 (Nesterov) |
| Learning rate | based on centralized training (scaled to $0.1 \times \frac{32*16}{256}$) |
| LR decay | $/10$ at epoch $30, 60, 80$ |
| LR warmup | Step-wise linearly within 5 epochs, starting from 0.1 |
| # Epochs | 90 |
| Weight decay | $10^{-4}$ |
| Normalization layer | EvoNorm [21] |
| Repetitions | Just one |
| Reported metric | Mean of all worker's test accuracies over the last 5 epochs |

## B.3   BERT finetuning

Table 8

## B.4   Random quadratics

We generate quadratics $\frac{1}{n} \sum_{i=1}^{n} f_i(\mathbf{x})$ of $\mathbf{x} \in \mathbb{R}^d$ where

$$f_i(\mathbf{x}) = \|\mathbf{A}_i \mathbf{x} + \mathbf{b}_i\|_2^2.$$

Here the local Hessian $\mathbf{A}_i \in \mathbb{R}^{d \times d}$ control the shape of worker $i$'s local objective functions and the offset $\mathbf{b}_i \in \mathbb{R}^d$ allows for shifting the worker's optimum. The generation procedure is as follows:

Table 8: Default experimental settings for BERT finetuning

| | |
|---|---|
| Dataset | AG News [49] |
| Data augmentation | none |
| Architecture | DistilBERT [34] |
| Training objective | cross entropy |
| Evaluation objective | top-1 accuracy |
| Number of workers | 16 |
| Topology | restricted to a ring (chain for RelaySGD) |
| Gossip weights | Metropolis-Hastings (1/3 for ring) |
| Data distribution | Heterogeneous, not shuffled, according to Dirichlet sampling procedure from [20] |
| Batch size | 32 patches per worker |
| Adam $\beta_1$ | 0.9 |
| Adam $\beta_2$ | 0.999 |
| Adam $\varepsilon$ | $10^{-8}$ |
| Learning rate | Tuned c.f. subsection C.3 |
| LR decay | constant learning rate |
| LR warmup | no warmup |
| # Epochs | 5 |
| Weight decay | 0 |
| Normalization layer | LayerNorm [3] |
| Repetitions | 3, with varying seeds |
| Reported metric | Mean of all worker's test accuracies over the last 5 epochs |

1. Sample $\mathbf{A}_i \in \mathbb{R}^{d \times d}$ from an i.i.d. element-wise standard normal distribution, independently for each worker.

2. Control the smoothness $L$ and strong-convexity constant $\mu$. Decompose $\mathbf{A}_i = \mathbf{U}_i \mathbf{S}_i \mathbf{V}_i^\top$ using Singular Value Decomposition, and replace $\mathbf{A}_i$ with $\mathbf{A}_i \leftarrow \mathbf{U}_i \tilde{\mathbf{S}}_i \mathbf{V}_i^\top$, where $\tilde{\mathbf{S}}_i \in \mathbb{R}^{d \times d}$ is a diagonal matrix with diagonal entries $[\mu, \frac{d-2}{d-1}\mu + \frac{1}{d-1}L, \ldots, L]$.

3. Control the heterogeneity $\zeta_2$ by shifting worker's optima into random directions.

   (a) Sample random directions $\mathbf{d}_i \in \mathbb{R}^d$ from an i.i.d. element-wise standard normal distributions, independently for each worker.

   (b) Instantiate a scalar $s \leftarrow 1$ and optimize it using binary search:

   (c) Move local optima by $s\mathbf{d}_i$ by setting $\mathbf{b}_i \leftarrow \mathbf{A}_i s \mathbf{d}_i$.

   (d) Move all optima $\mathbf{b}_i \leftarrow \mathbf{b}_i - \mathbf{A}_i \mathbf{x}^\star$ such that the global optimum $\mathbf{x}^\star$ remains at zero.

   (e) Evaluate $\zeta^2 = \frac{1}{n}\sum_{i=1}^n \|\nabla f_i(\mathbf{x}^\star)\|_2^2$ and adjust the scale factor $s$ until $\zeta^2$ is as desired. Repeat from step (c).

4. Control the initial distance to the optimum $r_0$. Sample a random vector for the optimum $\mathbf{x}^\star$ from an i.i.d. element-wise normal distribution and scale it to have norm $r_0$. Shift all worker's optima in this direction by updating $\mathbf{b}_i \leftarrow \mathbf{b}_i + \mathbf{A}_i \mathbf{x}^\star$.

## C  Hyper-parameters and tuning details

### C.1  Cifar-10

For our image classification experiments on Cifar-10, we have independently tuned learning rates for each algorithm, at each data heterogeneity level $\alpha$, and separately for SGD with and without momentum. We followed the following procedure:

1. We found an appropriate learning rate for centralized (all-reduce) training (by using the procedure below)

2. Start the search from this learning rate. For RelaySGD, we apply a correction computed as in subsection D.1.

3. Grid-search the learning rate by multiplying and dividing by powers of two. Try larger and smaller learning rates, until the best result found so far is sandwiched between two learning rates that gave worse results.

4. Repeat the experiment with 3 random seeds.

5. If any of those replicas diverged, reduce the learning rate by a factor two until it does.

For the experiments in Table 1, we used the learning rates listed in Table 9.

Table 9: Learning rates used for Cifar-10/ VGG-11. Numbers between parentheses indicate the number of converged replications with this learning rate.

| Algorithm | Topology | $\alpha = 1.00$ (most homogeneous) | $\alpha = 0.1$ | $\alpha = .01$ (most heterogeneous) |
|---|---|---|---|---|
| All-reduce | fully connected | 0.100 (3) | 0.100 (3) | 0.100 (3) |
| +momentum | | 0.100 (3) | 0.100 (3) | 0.100 (3) |
| **RelaySGD** | binary trees | 1.200 (3) | 0.600 (3) | 0.300 (3) |
| **+local momentum** | | 0.600 (3) | 0.300 (3) | 0.150 (3) |
| DP-SGD [19] | ring | 0.400 (3) | 0.100 (3) | 0.200 (3) |
| +quasi-global mom. [20] | | 0.100 (3) | 0.025 (3) | 0.050 (3) |
| $D^2$ [36] | ring | 0.200 (3) | 0.200 (3) | 0.100 (3) |
| +local momentum | | 0.050 (3) | 0.050 (3) | 0.013 (3) |
| Stochastic gradient push [2] | time-varying exponential [2] | 0.400 (3) | 0.200 (3) | 0.200 (3) |
| +local momentum | | 0.100 (3) | 0.100 (3) | 0.025 (3) |

## C.2 ImageNet

Due to the high resource requirements, we did not tune the learning rate for our ImageNet experiments. We identified a suitable learning rate based on prior work, and used this for all experiments. For RelaySGD, we used the analytically computed learning rate correction from subsection D.1.

## C.3 BERT finetuning

For DistilBERT fine-tuning experiments on AG News, we have independently tuned learning rate for each algorithm. We search the learning rate in the grid of $\{1e-5, 3e-5, 5e-5, 7e-5, 9e-5\}$ and we extend the grid to ensure that the best hyper-parameter lies in the middle of our search grids, otherwise we extend our search grid.

For the experiments in Table 4, we used the learning rates listed in Table 10.

Table 10: Tuned learning rates used for AG News / DistilBERT (Table 4)

| Algorithm | Topology | Learning rate |
|---|---|---|
| Centralized Adam | fully-connected | 3e-5 |
| **Relay-Adam** | chain | 9e-4 |
| DP-SGD Adam | ring | 1e-6 |
| Quasi-global Adam [20] | ring | 1e-6 |

## C.4 Random quadratics

For Figures 2 and 3, we tuned the learning rate for each compared method to reach a desired quality level as quickly as possible, using binary search. We made a distinction between methods that are expected to converge linearly, and methods that are expected to reach a plateau. For experiments with stochastic noise, we tuned a learning rate without noise first, and then lowered the learning rate if needed to reach a desirable plateau. Please see the supplied code for implementation details.

# D   Algorithmic details

## D.1   Learning-rate correction for RelaySGD

In DP-SGD as well as all other algorithms we compared to, a gradient-based update $\mathbf{u}_i^{(t)}$ from worker $i$ at time $t$ will eventually, as $t \to \infty$ distribute uniformly with weights $\frac{1}{n}$ over all workers. In RelaySGD, the update also distributes uniformly (typically much quicker), but it will converge to a weight $\alpha \leq \frac{1}{n}$. The constant $\alpha$ is fixed throughout training and depends only on the network topology used. To correct for this loss in energy, you can scale the learning rate by a factor $\frac{1}{\alpha n}$.

Experimentally, we pre-compute $\alpha$ for each architecture by initialing a *scalar* model for each worker to zero, updating the models to 1, and running RelaySGD until convergence with no further model

updates. The worker will converge to the value $\alpha$. The correction factors that result from this procedure are illustrated in Figure 6.

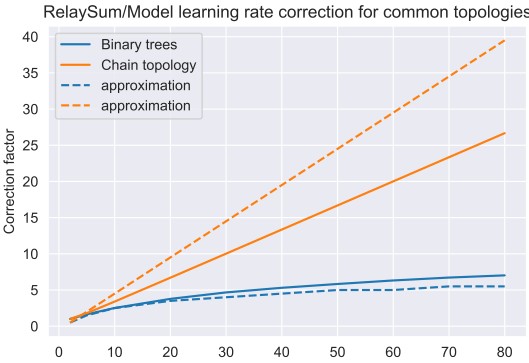

Figure 6: This network-topology-dependent correction factor is computed as follows: Each worker initializes a scalar model to 0 and sends a single fixed value 1 as gradient update through the RelaySGD algorithm. For DP-SGD and all-reduce, workers would converge to 1, but for RelaySGD, we lose some of this energy. If the workers converge to a value $\alpha$, we will scale the learning rate with $1/\alpha$ for RelaySGD compared to all-reduce.

In our deep learning experiments, we find that for each learning rate were centralized SGD converges, RelaySGD with the corrected learning rate converges too. Note that this learning rate correction is only useful if you already have a tuned learning rate from centralized experiments, or experiments with algorithms such as DP-SGD. If you start from scratch, tuning the learning rate for RelaySGD is no different form tuning the learning rate for any of the other algorithms.

## D.2 RelaySGD with momentum

RelaySGD follows Algorithm 1, but replaces the local update in line 3 with a local momentum. For Nesterov momentum with momentum-parameter $\alpha$, this is:

$$\mathbf{m}_i^{(t)} = \alpha \, \mathbf{m}_i^{(t-1)} + \nabla f_i(\mathbf{x}_i^{(t)}) \quad \text{(initialize } \mathbf{m}_i^0 = 0\text{)}$$
$$\mathbf{x}_i^{(t+1/2)} = \mathbf{x}_i^{(t)} - \gamma \left( \nabla f_i(\mathbf{x}_i^{(t)}) + \alpha \, \mathbf{m}_i^{(t)} \right).$$

## D.3 RelaySGD with Adam

Modifiying RelaySGD (Algorithm 1) to use Adam is analogous to RelaySGD with momentum (subsection D.2). All Adam state is updated locally. We use the standard Adam implementation of PyTorch 1.18.

## D.4 $D^2$ with momentum

We made slight modifications to the $D^2$ algorithm from Tang et al. [36] to allow time-varying learning rates and local momentum. The version we use is listed as Algorithm 2. Note that $D^2$ requires the smallest eigenvalue of the gossip matrix $\mathbf{W}$ to be $\geq -1/3$. This property is satisfied for Metropolis-Hasting matrices used on rings and double binary trees, but it was not in our Social Network Graph experiment (Figure 3). For this reason, we used the gossip matrix $(\mathbf{W} + \mathbf{I})/2$, from the otherwise-equivalent Exact Diffusion algorithm [45] on the social network graph.

## D.5 Gradient Tracking

Algorithm 3 lists our implementation of Gradient Tracking from Lorenzo and Scutari [22].

---

**Algorithm 2** $D^2$ [36] with momentum

---

**Input:** $\forall\, i$, $\mathbf{x}_i^{(0)} = \mathbf{x}^{(0)}$, learning rate $\gamma$, momentum $\alpha$, gossip matrix $\mathbf{W} \in \mathbb{R}^{n \times n}$, $\mathbf{c}_i^{(0)} = \mathbf{0} \in \mathbb{R}^d$.

1: **for** $t = 0, 1, \ldots$ **do**
2:     **for** node $i$ **in parallel**
3:         Update the local momentum buffer $\mathbf{m}_i^{(t)} = \alpha\, \mathbf{m}_i^{(t-1)} + \nabla f_i(\mathbf{x}_i^{(t)})$.
4:         Compute a local update $\mathbf{u}_i^{(t)} = -\gamma(\nabla f_i(\mathbf{x}_i^{(t)}) + \alpha\, \mathbf{m}_i^{(t)})$.
5:         Update the local model $\mathbf{x}_i^{(t+1/2)} = \mathbf{x}_i^{(t)} + \mathbf{u}_i^{(t)} + \mathbf{c}_i^{(t)}$.
6:         Average with neighbors: $\mathbf{x}_i^{(t+1)} = \sum_{j \in \mathcal{N}_i} \mathbf{W}_{ij} \mathbf{x}_j^{(t+1/2)}$.
7:         Update the local correction $\mathbf{c}_i^{(t+1)} = \mathbf{x}_i^{(t+1)} - \mathbf{x}_i^{(t)} - \mathbf{u}_i^{(t)}$.
8:     **end for**
9: **end for**

---

**Algorithm 3** Gradient Tracking [22]

---

**Input:** $\forall\, i$, $\mathbf{x}_i^{(0)} = \mathbf{x}^{(0)}$, learning rate $\gamma$, gossip matrix $\mathbf{W} \in \mathbb{R}^{n \times n}$, $\mathbf{c}_i^{(0)} = \mathbf{0} \in \mathbb{R}^d$.

1: **for** $t = 0, 1, \ldots$ **do**
2:     **for** node $i$ **in parallel**
3:         Compute a local update $\mathbf{u}_i^{(t)} = -\gamma \nabla f_i(\mathbf{x}_i^{(t)})$.
4:         Update the local model $\mathbf{x}_i^{(t+1/2)} = \mathbf{x}_i^{(t)} + \mathbf{u}_i^{(t)} + \mathbf{c}_i^{(t)}$.
5:         Average with neighbors: $\mathbf{x}_i^{(t+1)} = \sum_{j \in \mathcal{N}_i} \mathbf{W}_{ij} \mathbf{x}_j^{(t+1/2)}$.
6:         Update the correction and average: $\mathbf{c}_i^{(t+1)} = \sum_{j \in \mathcal{N}_i} \mathbf{W}_{ij} \left( \mathbf{c}_i^{(t)} - \mathbf{u}_i^{(t)} \right)$.
7:     **end for**
8: **end for**

---

### D.6 Stochastic Gradient Push with the time-varying exponential topology

Stochastic Gradient Push with the time-varying exponential topology from [2] demonstrates that decentralized learning algorithms can reduce communication in a data center setting where each node could talk to each other node. Algorithm 4 lists our implementation of this algorithm.

---

**Algorithm 4** Stochastic Gradient Push with time-varying exponential topology [2]

---

**Input:** $\forall\, i$, $\mathbf{x}_i^{(0)} = \mathbf{x}^{(0)}$, learning rate $\gamma$, $n = 2^k$ workers, $t' = 0$.

1: **for** $t = 0, 1, \ldots$ **do**
2:     **for** node $i$ **in parallel**
3:         $\mathbf{x}_i^{(t+1/2)} = \mathbf{x}_i^{(t)} + \mathbf{u}_i^{(t)} {\color{blue} -\gamma \nabla f_i(\mathbf{x}_i^{(t)})}$.         {\color{blue}(or momentum/Adam, like RelaySGD)}
4:         **for** 2 communication steps to equalize bandwidth with RelaySGD **do**
5:             Compute an offset $o = 2^{t' \bmod k}$.
6:             Send $\mathbf{x}_i^{(t+1/2)}$ to worker $i - o$.
7:             Receive and overwrite $\mathbf{x}_i^{(t+1/2)} \leftarrow \frac{1}{2}\left( \mathbf{x}_i^{(t+1/2)} + \mathbf{x}_{i+o}^{(t+1/2)} \right)$.
8:             $t' \leftarrow t' + 1$.
9:         **end for**
10:        Set $\mathbf{x}_i^{(t+1)} = \mathbf{x}_i^{(t+1/2)}$.
11:     **end for**
12: **end for**

---

## E  Additional experiments on RelaySGD

### E.1  Rings vs double binary trees on Cifar-10

In our experiments that target data-center inspired scenarios where the network topology is arbitrarily selected by the user to save bandwidth, RelaySGD uses double binary trees to communicate. They

use the same memory and bandwidth as rings (2 models sent/received per iteration) but they delays only scale with $\log n$, enabling RelaySGD, in theory, to run with very large numbers of workers $n$. Table 11 shows that in our Cifar-10 experiments with 16 there are minor improvements from using double binary trees over rings. Our baselines DP-SGD and $D^2$, however, perform significantly better on rings than on trees, so we use those results in the main paper.

Table 11: Comparing the performance of the algorithms in Table 1 on rings and double binary trees in the high-heterogeneity setting $\alpha = 0.01$. In both topologies, workers send and receive two full models per update step. With 16 workers, RelaySGD with momentum seems to benefit from double binary trees, RelaySGD has more consistently good results on a chain. We still opt for double binary trees based on their promise to scale to many workers. Other methods do not benefit from double binary trees over rings.

| Algorithm | Ring (Chain for RelaySGD) | Double binary trees |
|---|---|---|
| **RelaySGD** | 86.5% | 84.6% |
| +**local momentum** | 88.4% | 89.1% |
| DP-SGD [19] | 53.9% | 36.0% |
| +quasi-global mom. [20] | 63.3% | 57.5% |
| $D^2$ [36] | 38.2% | did not converge |
| +local momentum | 61.0% | did not converge |

## E.2  Scaling the number of workers on Cifar-10

In this experiment (Table 12), use momentum-SGD on 16, 32 and 64 workers compare the scaling of RelaySGD to SGP [2]. We fix the parameter $\alpha$ that determines the level of data heterogeneity to $\alpha = 0.01$. Note that this level of $\alpha$ could lead to more challenging heterogeneity when there are many workers (and hence many smaller local subsets of the data), compared to when there are few workers.

Table 12:  Scaling the number of workers in heterogeneous Cifar-10. The heterogeneity level $\alpha = 0.01$ is kept constant, although it does change its meaning when the number of workers changes. RelaySGD scales at least well as Stochastic Gradient Push [2] (with equal communication budget). It is surprising that RelaySGD with 64 workers performs significantly better on a chain topology than on the double binary trees. This behavior does not match what our observations on quadratic toy-problems.

| Algorithm | Topology | 16 workers | 32 workers | 64 workers |
|---|---|---|---|---|
| All-reduce (baseline) | fully connected | 89.5% | 88.9% | 87.2% |
| RelaySGD | binary trees | 89.3% | 86.1% | 63.7% |
| | chain | 88.4% | 86.6% | 83.1% |
| Stochastic gradient push [2] | time-varying exponential [2] | 87.0% | 68.9% | 62.4% |

Table 13:  Tuned learning rates for Table 12. We tuned the learning rate for each setting on a multiplicative grid with spacing $\sqrt{2}$, and then repeated each experiment 3 times. If both repetitions diverged, we would change to a smaller learning rate in the grid. Numbers in parentheses are the 'effective' learning rates corrected according to subsection D.1.

| Algorithm | Topology | 16 workers | | 32 workers | | 64 workers | |
|---|---|---|---|---|---|---|---|
| All-reduce (baseline) | fully connected | 0.1 | (0.100) | 0.05 | (0.050) | 0.05 | (0.050) |
| RelaySGD | binary trees | 0.282 | (0.066) | 0.2 | (0.035) | 0.2 | (0.027) |
| | chain | 0.2 | (0.047) | 0.4 | (0.070) | 0.8 | (0.108) |
| Stochastic gradient push [2] | time-varying exp. | 0.025 | (0.025) | 0.025 | (0.025) | 0.0125 | (0.013) |

## E.3  Independence of heterogeneity

The benefits of RelaySGD over some other methods shows most when workers have heterogeneous training objectives. Figure 7 compares several algorithms with varying levels of data heterogeneity on synthetic quadratics on a ring topology with 32 workers. Like $D^2$, RelaySGD converges linearly, and does not require more steps when the data becomes more heterogeneous. Note that, even though

RelaySGD operates on a chain network instead of a ring, it is as fast as $D^2$. On other topologies, such as a star topology, or on trees, RelaySGD can even be faster than $D^2$ (see Appendix E.4), while maintaining the same independence of heterogeneity.

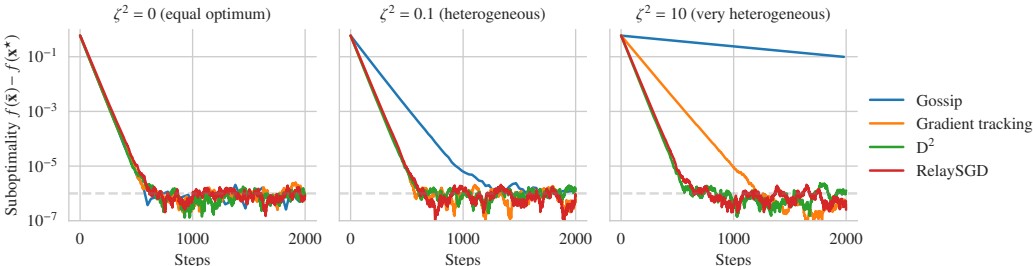

Figure 7: Random quadratics on *ring* networks of size 32 with varying data heterogeneity $\zeta^2$ and all other theoretical quantities fixed. To simulate stochastic noise, we add random normal noise to each gradient update. For each method, the learning rate is tuned to reach suboptimality $\leq 10^{-6}$ the fastest. RelaySGD operates on a chain network instead of a ring. Like $D^2$, it does not require more steps when the worker's objectives are more heterogeneous.

## E.4   Star topology

On star-topologies, the set of neighbors of worker 0 is $\{1, 2, \ldots, n\}$ and the set of neighbors for every other worker is just $\{0\}$. While $D^2$ and RelaySGD are equally fast in the synthetic experiments on *ring* topologies in subsection E.3, RelaySGD is significantly faster on *star* topologies as illustrates by Figure 8.

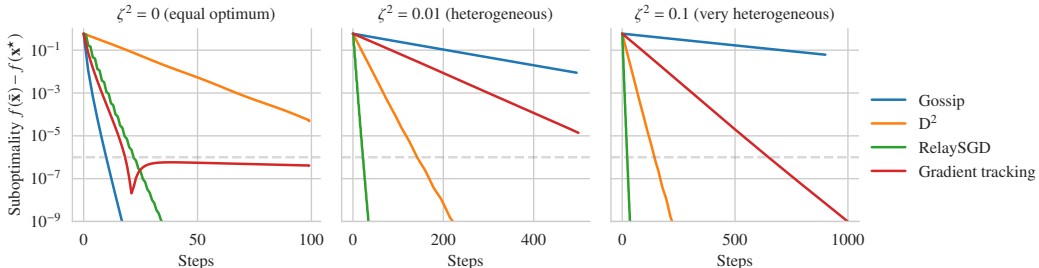

Figure 8: Random quadratics on *star* networks of size 32 with varying data heterogeneity $\zeta^2$ and all other theoretical quantities fixed. For each method, the learning rate is tuned to reach suboptimality $\leq 10^{-6}$ the fastest. Like $D^2$, RelaySGD does not require more steps when the worker's objectives are more heterogeneous. Note that for $\zeta^2 = 0$ (left figure), our tuning procedure found a learning rate where Gradient Tracking does converge to $< \leq 10^{-6}$, but does not converge linearly. It would with a lower learning rate.

## F   RelaySum for distributed mean estimation

We conceptually separate the optimization algorithm RelaySGD from the communication mechanism RelaySum that uniformly distributes updates across a peer-to-peer network. We made this choice because we envision other applications of the RelaySum mechanism outside of optimization for machine learning. To illustrate this point, this section introduces RelaySum for Distributed Mean Estimation (Algorithm 5).

In distributed mean estimation, workers are connected in a network just as in our optimization setup, but instead of models gradients, they receive samples $\hat{\mathbf{d}}^{(t)} \sim \mathcal{D}$ of the distribution $\mathcal{D}$ at timestep $t$. The workers estimate the mean $\bar{\mathbf{d}}$ the mean of $\mathcal{D}$, and we measure their average squared error to the true mean.

**Algorithm 5** RelaySum for Distributed Mean Estimation

**Input:** $\forall\, i,\; \mathbf{x}_i^{(0)} = \mathbf{0}, \mathbf{y}_i^{(0)} = \mathbf{0}, s_i^{(0)} = 0;\; \forall\, i,j,\; \mathbf{m}_{i \to j}^{(-1)} = \mathbf{0}$, tree network

1: **for** $t = 0, 1, \dots$ **do**
2:     **for** node $i$ **in parallel**
3:         **for** each neighbor $j \in \mathcal{N}_i$ **do**
4:             Get a sample $\hat{\mathbf{d}}_i^{(t)} \sim \mathcal{D}$.
5:             Send $\mathbf{m}_{i \to j}^{(t)} = \hat{\mathbf{d}}_i^{(t)} + \sum_{k \in \mathcal{N}_i \setminus j} \mathbf{m}_{k \to i}^{(t-1)}$.
6:             Send $c_{i \to j}^{(t)} = 1 + \sum_{k \in \mathcal{N}_i \setminus j} c_{k \to i}^{(t-1)}$.
7:             Receive $\mathbf{m}_{j \to i}^{(t)}$ and $c_{j \to i}^{(t)}$ from node $j$.
8:         **end for**
9:         Update the sum of samples $\mathbf{y}_i^{(t+1)} = \mathbf{y}_i^{(t)} + \hat{\mathbf{d}}_i^{(t)} + \sum_{j \in \mathcal{N}_i} \mathbf{m}_{j \to i}^{(t)}$.
10:        Update the sum of counts $s_i^{(t+1)} = s_i^{(t)} + 1 + \sum_{j \in \mathcal{N}_i} c_{j \to i}^{(t)}$.
11:        Output average estimate $\mathbf{x}_i^{(t)} = \mathbf{y}_i^{(t)} / s_i^{(t)}$
12:     **end for**
13: **end for**

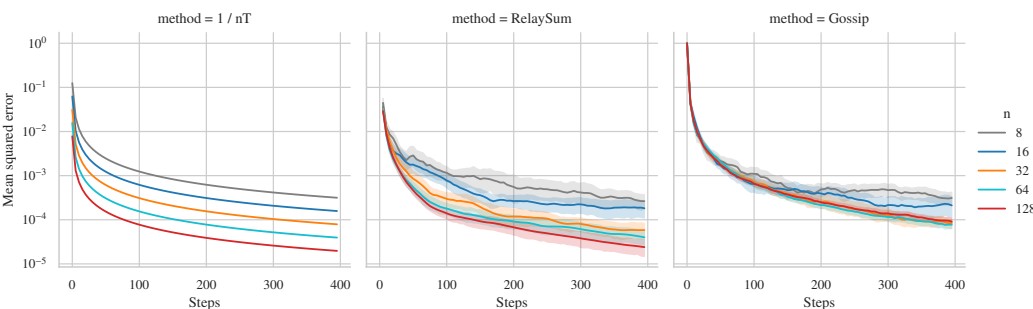

Figure 9: RelaySum for Distributed Mean Estimation compured to a gossip-based baseline, on a ring topology (chain for RelaySGD). Workers receive samples from a normal distribution $\mathcal{N}(1, 1)$ with mean 1. RelaySum, using Algorithm 5 achieves a variance reduction of $\mathcal{O}\left(\frac{1}{nT}\right)$.

In algorithm 5, the output estimates $\mathbf{x}_i^{(t)}$ of a worker $i$ is a uniform average of all samples that can reach a worker $i$ at that timestep. This algorithm enjoys variance reduction of $\mathcal{O}\left(\frac{1}{nT}\right)$, a desirable property that is in general not shared by gossip-averaging-based algorithms on arbitrary graphs.

In Figure 9, we compare this algorithm to a simple gossip-based baseline.

## G   Alternative optimizer based on RelaySum

Apart from RelaySGD presented in the main paper, there are other ways to build optimization algorithms based on the RelaySum communication mechanism. In this section, we describe RelaySGD/Grad (Algorithm 6), an alternative to RelaySGD that does uses the RelaySum mechanism on *gradient updates* rather than on *models*.

RelaySGD/Grad distributes each update uniformly over all workers in a finite number of steps. This means that worker's models differ by only a finite number of $\mathcal{O}(\tau_{\max} maxn)$ that are scaled as $\frac{1}{n}$. With this property, it achieves tighter consensus than typical gossip averaging, and it also works well in deep learning. Contrary to RelaySGD, however, this algorithm is not fully independent of data heterogeneity, due to the delay in the updates. When the data heterogeneity $\zeta^2 > 0$, RelaySGD/Grad does not converge linearly, but its suboptimality saturates at a level that depends on $\zeta^2$.

The sections below study this alternative algorithm in detail, both theoretically and experimentally. The key differences between RelaySGD and RelaySGD/Grad are:

|  | RelaySGD | RelaySGD/Grad |
|---|---|---|
| Provably independent of data heterogeneity $\zeta^2$ | yes | no |
| Distributes updates exactly uniform in finite steps | no | yes |
| Loses energy of gradient updates (subsection D.1) | yes | no |
| Works experimentally with momentum / Adam | yes | no |
| Robust to lost messages + can support workers joining/leaving | yes | no |

---

**Algorithm 6** RelaySGD/Grad

---

**Input:** $\forall\, i,\ \mathbf{x}_i^{(0)} = \mathbf{x}^{(0)};\ \forall\, i, j, \mathbf{m}_{i \to j}^{(-1)} = \mathbf{0}$, learning rate $\gamma$, tree network
1: **for** $t = 0, 1, \ldots$ **do**
2:     **for** node $i$ **in parallel**
3:         $\mathbf{u}_i^{(t)} = -\gamma \nabla f_i(\mathbf{x}_i^{(t)}, \xi_i^{(t)})$
4:         **for** each neighbor $j \in \mathcal{N}_i$ **do**
5:             Send $\mathbf{m}_{i \to j}^{(t)} = \mathbf{u}_i^{(t)} + \sum_{k \in \mathcal{N}_i \setminus j} \mathbf{m}_{k \to i}^{(t-1)}$.
6:             Receive $\mathbf{m}_{j \to i}^{(t)}$ from node $j$.
7:         **end for**
8:         $\mathbf{x}_i^{(t+1)} = \mathbf{x}_i^{(t)} + \frac{1}{n} \left( \mathbf{u}_i^{(t)} + \sum_{j \in \mathcal{N}_i} \mathbf{m}_{j \to i}^{(t)} \right)$
9:     **end for**
10: **end for**

---

### G.1 Theoretical analysis of RelaySGD/Grad

In this section we provide the theoretical analysis for RelaySGD/Grad. As the proof and analysis is very similar to [16], we only provide the case for the convex objective.

### G.1.1 Proof of RelaySGD/Grad for the convex case

Let $\mathbf{x}^\star$ be the minimizer of $f$ and define the following iterates

- $r_t := \mathbb{E} \left\| \bar{\mathbf{x}}^{(t)} - \mathbf{x}^\star \right\|^2$,

- $e_t := f(\bar{\mathbf{x}}^{(t)}) - f(\mathbf{x}^\star)$,

- $\Xi_t := \frac{1}{n} \sum_{i=1}^n \left\| \mathbf{x}_i^{(t)} - \bar{\mathbf{x}}^{(t)} \right\|^2$.

**Proposition X.** *Let function $F_i(\mathbf{x}, \xi)$, $i \in [n]$ be L-smooth (Assumption A) with bounded noise at the optimum (Assumption H). Then for any $\mathbf{x}_i \in \mathbb{R}^d$,*

$$\mathbb{E}_{\xi_1^t, \ldots, \xi_n^t} \left\| \frac{1}{n} \sum_{i=1}^n (\nabla f_i(\mathbf{x}_i^{(t)}) - \nabla F_i(\mathbf{x}_i^{(t)}, \xi_i^{(t)})) \right\|^2 \leq \frac{3}{n}(L^2 \Xi_t + 2L e_t + \bar{\sigma}^2)$$

*Proof.* In this proof we ignore the superscript $t$ as it does not raise embiguity.

$$\mathbb{E}_{\xi_1,\dots,\xi_n} \left\| \frac{1}{n} \sum_{i=1}^{n} (\nabla f_i(\mathbf{x}_i) - \nabla F_i(\mathbf{x}_i, \xi_i)) \right\|^2 \leq \frac{1}{n^2} \sum_{i=1}^{n} \mathbb{E}_{\xi_i} \|\nabla f_i(\mathbf{x}_i) - \nabla F_i(\mathbf{x}_i, \xi_i)\|^2$$

$$= \frac{1}{n^2} \sum_{i=1}^{n} \mathbb{E}_{\xi_i} \|\nabla f_i(\mathbf{x}_i) - \nabla F_i(\mathbf{x}_i, \xi_i) \pm \nabla F_i(\bar{\mathbf{x}}, \xi_i) \pm \nabla f_i(\bar{\mathbf{x}}) \pm \nabla F_i(\mathbf{x}^\star, \xi_i) \pm \nabla f_i(\mathbf{x}^\star)\|^2$$

$$\leq \frac{3}{n^2} \sum_{i=1}^{n} \mathbb{E}_{\xi_i} \left( \|\nabla f_i(\mathbf{x}_i) - \nabla f_i(\bar{\mathbf{x}}) + \nabla F_i(\bar{\mathbf{x}}, \xi_i) - \nabla F_i(\mathbf{x}_i, \xi_i)\|^2 \right.$$
$$\left. + \|\nabla f_i(\bar{\mathbf{x}}) - \nabla f_i(\mathbf{x}^\star) + \nabla F_i(\mathbf{x}^\star, \xi_i) - \nabla F_i(\bar{\mathbf{x}}, \xi_i)\|^2 + \|\nabla f_i(\mathbf{x}^\star) - \nabla F_i(\mathbf{x}^\star, \mathbf{x}_i)\|^2 \right)$$

$$\leq \frac{3}{n^2} \sum_{i=1}^{n} \mathbb{E}_{\xi_i} (\|\nabla F_i(\mathbf{x}_i, \xi_i) - \nabla F_i(\bar{\mathbf{x}}, \xi_i)\|^2 + \|\nabla F_i(\bar{\mathbf{x}}, \xi_i) - \nabla F_i(\mathbf{x}^\star, \xi_i)\|^2 + \|\nabla F_i(\mathbf{x}^\star, \mathbf{x}_i) - \nabla f_i(\mathbf{x}^\star)\|^2)$$

$$\leq \frac{3}{n^2} \sum_{i=1}^{n} (L^2 \|\mathbf{x}_i - \bar{\mathbf{x}}\|^2 + 2L(f_i(\bar{\mathbf{x}}) - f_i(\mathbf{x}^\star)) + \sigma_i^2)$$

$\square$

**Lemma 20.** *(Descent lemma for convex objective.) If $\gamma \leq \frac{1}{10L}$, then*
$$r_{t+1} \leq (1 - \tfrac{\gamma\mu}{2}) r_t - \gamma e_t + 3\gamma L \Xi_t + \tfrac{3}{n} \gamma^2 \bar{\sigma}^2.$$

*Proof.* Throughout this proof we use $\mathbb{E} = \mathbb{E}_{\xi_1^t,\dots,\xi_n^t}$. Expand iterate $r_{t+1} = \mathbb{E} \|\bar{\mathbf{x}}^{(t+1)} - \mathbf{x}^\star\|^2$

$$\mathbb{E} \|\bar{\mathbf{x}}^{(t+1)} - \mathbf{x}^\star\|^2$$
$$= \mathbb{E} \|\bar{\mathbf{x}}^{(t)} - \tfrac{\gamma}{n} \sum_{i=1}^{n} \nabla F_i(\mathbf{x}_i^{(t)}, \xi_i^{(t)}) \pm \tfrac{\gamma}{n} \sum_{i=1}^{n} \nabla f_i(\mathbf{x}_i^{(t)}) - \mathbf{x}^\star\|^2$$
$$= \|\bar{\mathbf{x}}^{(t)} - \mathbf{x}^\star - \tfrac{\gamma}{n} \sum_{i=1}^{n} \nabla f_i(\mathbf{x}_i^{(t)})\|^2 + \mathbb{E} \|\tfrac{\gamma}{n} \sum_{i=1}^{n} \nabla F_i(\mathbf{x}_i^{(t)}, \xi_i^{(t)}) - \tfrac{\gamma}{n} \sum_{i=1}^{n} \nabla f_i(\mathbf{x}_i^{(t)})\|^2$$
$$+ 2\mathbb{E}\langle \bar{\mathbf{x}}^{(t)} - \mathbf{x}^\star - \tfrac{\gamma}{n} \sum_{i=1}^{n} \nabla f_i(\mathbf{x}_i^{(t)}), \tfrac{\gamma}{n} \sum_{i=1}^{n} \nabla F_i(\mathbf{x}_i^{(t)}, \xi_i^{(t)}) - \tfrac{\gamma}{n} \sum_{i=1}^{n} \nabla f_i(\mathbf{x}_i^{(t)}) \rangle$$
$$= \|\bar{\mathbf{x}}^{(t)} - \mathbf{x}^\star - \tfrac{\gamma}{n} \sum_{i=1}^{n} \nabla f_i(\mathbf{x}_i^{(t)})\|^2 + \mathbb{E} \|\tfrac{\gamma}{n} \sum_{i=1}^{n} \nabla F_i(\mathbf{x}_i^{(t)}, \xi_i^{(t)}) - \tfrac{\gamma}{n} \sum_{i=1}^{n} \nabla f_i(\mathbf{x}_i^{(t)})\|^2$$

The second term is bounded by Proposition X. Consider the first term

$$\|\bar{\mathbf{x}}^{(t)} - \mathbf{x}^\star - \tfrac{\gamma}{n} \sum_{i=1}^{n} \nabla f_i(\mathbf{x}_i^{(t)})\|^2$$
$$\leq \|\bar{\mathbf{x}}^{(t)} - \mathbf{x}^\star\|^2 + \gamma^2 \underbrace{\|\tfrac{1}{n} \sum_{i=1}^{n} \nabla f_i(\mathbf{x}_i^{(t)})\|^2}_{=:T_1} - 2\gamma \underbrace{\langle \bar{\mathbf{x}}_t - \mathbf{x}^\star, \tfrac{1}{n} \sum_{i=1}^{n} \nabla f_i(\mathbf{x}_i^{(t)}) \rangle}_{=:T_2}.$$

First consider $T_1$,

$$T_1 = \|\tfrac{1}{n} \sum_{i=1}^{n} (\nabla f_i(\mathbf{x}_i^{(t)}) - \nabla f_i(\bar{\mathbf{x}}^{(t)}) + \nabla f_i(\bar{\mathbf{x}}^{(t)}) - \nabla f_i(\mathbf{x}^\star))\|^2$$
$$\leq \tfrac{2L^2}{n} \sum_{i=1}^{n} \|\mathbf{x}_i^{(t)} - \bar{\mathbf{x}}^{(t)}\|^2 + \tfrac{2}{n} \sum_{i=1}^{n} \|\nabla f_i(\bar{\mathbf{x}}^{(t)}) - \nabla f_i(\mathbf{x}^\star)\|^2$$
$$\overset{(5)}{\leq} \tfrac{2L^2}{n} \sum_{i=1}^{n} \|\mathbf{x}_i^{(t)} - \bar{\mathbf{x}}^{(t)}\|^2 + \tfrac{4L}{n} \sum_{i=1}^{n} (f_i(\bar{\mathbf{x}}^{(t)}) - f_i(\mathbf{x}^\star) - \langle \bar{\mathbf{x}}^{(t)} - \mathbf{x}^\star, \nabla f_i(\mathbf{x}^\star) \rangle)$$
$$= \tfrac{2L^2}{n} \sum_{i=1}^{n} \|\mathbf{x}_i^{(t)} - \bar{\mathbf{x}}^{(t)}\|^2 + 4L(f(\bar{\mathbf{x}}^{(t)}) - f(\mathbf{x}^\star))$$
$$= 2L^2 \Xi_t + 4L e_t.$$

Consider $T_2$,

$$T_2 = \tfrac{1}{n} \sum_{i=1}^{n} (\langle \bar{\mathbf{x}}^{(t)} - \mathbf{x}_i^{(t)}, \nabla f_i(\mathbf{x}_i^{(t)}) \rangle + \langle \mathbf{x}_i^{(t)} - \mathbf{x}^\star, \nabla f_i(\mathbf{x}_i^{(t)}) \rangle)$$
$$\geq \tfrac{1}{n} \sum_{i=1}^{n} \left( f_i(\bar{\mathbf{x}}^{(t)}) - f_i(\mathbf{x}_i^{(t)}) - \tfrac{L}{2} \|\bar{\mathbf{x}}^{(t)} - \mathbf{x}_i^{(t)}\|^2 + \langle \mathbf{x}_i^{(t)} - \mathbf{x}^\star, \nabla f_i(\mathbf{x}_i^{(t)}) \rangle \right)$$
$$\geq \tfrac{1}{n} \sum_{i=1}^{n} \left( f_i(\bar{\mathbf{x}}^{(t)}) - f_i(\mathbf{x}_i^{(t)}) - \tfrac{L}{2} \|\bar{\mathbf{x}}^{(t)} - \mathbf{x}_i^{(t)}\|^2 + f_i(\mathbf{x}_i^{(t)}) - f_i(\mathbf{x}^\star) + \tfrac{\mu}{2} \|\mathbf{x}_i^{(t)} - \mathbf{x}^\star\|^2 \right)$$
$$= f(\bar{\mathbf{x}}^{(t)}) - f(\mathbf{x}^\star) + \tfrac{1}{n} \sum_{i=1}^{n} \left( \tfrac{\mu}{2} \|\mathbf{x}_i^{(t)} - \mathbf{x}^\star\|^2 - \tfrac{L}{2} \|\bar{\mathbf{x}}^{(t)} - \mathbf{x}_i^{(t)}\|^2 \right)$$
$$\geq f(\bar{\mathbf{x}}^{(t)}) - f(\mathbf{x}^\star) + \tfrac{1}{n} \sum_{i=1}^{n} \left( \tfrac{\mu}{4} \|\bar{\mathbf{x}}^{(t)} - \mathbf{x}^\star\|^2 - \tfrac{\mu+L}{2} \|\bar{\mathbf{x}}^{(t)} - \mathbf{x}_i^{(t)}\|^2 \right)$$
$$\geq e_t + \tfrac{\mu}{4} r_t - L \Xi_t$$

where the first inequality and the second inequality uses the $L$-smoothness and $\mu$-convexity of $f_i$.

Combine both $T_1$, $T_2$ and Proposition X we have

$$r_{t+1} \leq r_t + \gamma^2(2L^2\Xi_t + 4Le_t) - 2\gamma(e_t + \tfrac{\mu}{4}r_t - L\Xi_t) + \tfrac{3}{n}\gamma^2(L^2\Xi_t + 2Le_t + \bar{\sigma}^2)$$
$$= (1 - \tfrac{\gamma\mu}{2})r_t - 2\bar{\gamma}(1 - 5L\gamma)e_t + \gamma L(5\gamma L + 2)\Xi_t + \tfrac{3}{n}\gamma^2\bar{\sigma}^2.$$

In addition if $\gamma \leq \frac{1}{10L}$, then

$$r_{t+1} \leq (1 - \tfrac{\gamma\mu}{2})r_t - \gamma e_t + 3\gamma L\Xi_t + \tfrac{3}{n}\gamma^2\bar{\sigma}^2.$$

$\square$

**Lemma 21.** *Bound the consensus distance as follows*

$$\Xi_t \leq 3\gamma^2\tau_{\max}\sum_{t'=[t-\tau_{\max}]^+}^{t-1}\left(2L^2\Xi_{t'} + 4Le_{t'} + (\bar{\sigma}^2 + \bar{\zeta}^2)\right).$$

*Furthermore, multiply with a non-negative sequence $\{w_t\}_{t\geq 0}$ and average over time gives*

$$\frac{1}{W_T}\sum_{t=0}^{T}w_t\Xi_t \leq \frac{1}{6LW_T}\sum_{t=0}^{T}w_te_t + 6\gamma^2\tau_{\max}^2(\bar{\sigma}^2 + \bar{\zeta}^2)$$

*where $W_T := \sum_{t=0}^{T}w_t$ and $\gamma \leq \frac{1}{10L\tau_{\max}}$.*

*Proof.* Throughout this proof we use $\mathbb{E} = \mathbb{E}_{\xi_1^t,\dots,\xi_n^t}$. Denote $[x]^+ := \max\{x, 0\}$. For all $i \in [n]$,

$$\mathbb{E}\|\mathbf{e}_i^t\|^2 = \mathbb{E}\|\tfrac{\gamma}{n}\sum_{j=1}^{n}\sum_{t'=[t-\tau_{\max ij}]^+}^{t-1}\nabla F_j(\mathbf{x}_j^{(t')}, \xi_j^{(t')}) \pm \nabla f_j(\mathbf{x}_j^{(t')})\|^2$$

$$\leq \tfrac{\gamma^2}{n}\sum_{j=1}^{n}\mathbb{E}\|\sum_{t'=[t-\tau_{\max ij}]^+}^{t-1}\nabla F_j(\mathbf{x}_j^{(t')}, \xi_j^{(t')}) \pm \nabla f_j(\mathbf{x}_j^{(t')})\|^2$$

$$\leq \tfrac{\gamma^2\tau_{\max}}{n}\sum_{j=1}^{n}\sum_{t'=[t-\tau_{\max}]^+}^{t-1}\mathbb{E}\|\nabla F_j(\mathbf{x}_j^{(t')}, \xi_j^{(t')}) \pm \nabla f_j(\mathbf{x}_j^{(t')})\|^2$$

$$= \tfrac{\gamma^2\tau_{\max}}{n}\sum_{j=1}^{n}\sum_{t'=[t-\tau_{\max}]^+}^{t-1}\mathbb{E}\|\nabla F_j(\mathbf{x}_j^{(t')}, \xi_j^{(t')}) - \nabla f_j(\mathbf{x}_j^{(t')})\|^2$$

$$+ \underbrace{\tfrac{\gamma^2\tau_{\max}}{n}\sum_{j=1}^{n}\sum_{t'=[t-\tau_{\max}]^+}^{t-1}\|\nabla f_j(\mathbf{x}_j^{(t')})\|^2}_{=:T_3}$$

We can apply Proposition X to the first term

$$\tfrac{\gamma^2\tau_{\max}}{n}\sum_{j=1}^{n}\sum_{t'=[t-\tau_{\max}]^+}^{t-1}\mathbb{E}\|\nabla F_j(\mathbf{x}_j^{(t')}, \xi_j^{(t')}) - \nabla f_j(\mathbf{x}_j^{(t')})\|^2 \leq 3\gamma^2\tau_{\max}\sum_{t'=[t-\tau_{\max}]^+}^{t-1}(L^2\Xi_{t'} + 2Le_{t'} + \bar{\sigma}^2).$$

The second term $T_3$ can be bounded by adding $0 = \pm\nabla f_j(\bar{\mathbf{x}}^{(t')}) \pm \nabla f_j(\mathbf{x}^\star)$ inside the norm

$$T_3 \leq \tfrac{\gamma^2\tau_{\max}}{n}\sum_{j=1}^{n}\sum_{t'=[t-\tau_{\max}]^+}^{t-1}\|\nabla f_j(\mathbf{x}_j^{(t')}) \pm \nabla f_j(\bar{\mathbf{x}}^{(t')}) \pm \nabla f_j(\mathbf{x}^\star)\|^2$$

$$\leq \tfrac{3\gamma^2\tau_{\max}}{n}\sum_{j=1}^{n}\sum_{t'=[t-\tau_{\max}]^+}^{t-1}\left(L^2\|\mathbf{x}_j^{(t')} - \bar{\mathbf{x}}^{(t')}\|^2 + \|\nabla f_j(\bar{\mathbf{x}}^{(t')}) - \nabla f_j(\mathbf{x}^\star)\|^2 + \|\nabla f_j(\mathbf{x}^\star)\|^2\right)$$

$$= 3\gamma^2\tau_{\max}\sum_{t'=[t-\tau_{\max}]^+}^{t-1}\left(L^2\Xi_{t'} + \tfrac{1}{n}\sum_{j=1}^{n}\|\nabla f_j(\bar{\mathbf{x}}^{(t')}) - \nabla f_j(\mathbf{x}^\star)\|^2 + \bar{\zeta}^2\right)$$

$$\overset{(5)}{\leq} 3\gamma^2\tau_{\max}\sum_{t'=[t-\tau_{\max}]^+}^{t-1}\left(L^2\Xi_{t'} + 2L(f(\bar{\mathbf{x}}^{(t')}) - f(\mathbf{x}^\star)) + \bar{\zeta}^2\right)$$

Therefore

$$\mathbb{E}\|\mathbf{e}_i^t\|^2 \leq 3\gamma^2\tau_{\max}\sum_{t'=[t-\tau_{\max}]^+}^{t-1}(2L^2\Xi_{t'} + 4Le_{t'} + (\bar{\sigma}^2 + \bar{\zeta}^2)).$$

Average over $i$ on both sides and note the right hand side does not depend on index $i$,

$$\Xi_t = \tfrac{1}{n}\sum_{i=1}^{n}\|\mathbf{e}_i^t\|^2 \leq 3\gamma^2\tau_{\max}\sum_{t'=[t-\tau_{\max}]^+}^{t-1}\left(2L^2\Xi_{t'} + 4Le_{t'} + \bar{\sigma}^2\right).$$

Multiply both sides by $w_t$ and sum over $t$ gives

$$\frac{1}{W_T}\sum_{t=0}^{T}w_t\Xi_t \leq \frac{3\gamma^2\tau_{\max}^2}{W_T}\sum_{t=0}^{T}w_t\left(2L^2\Xi_t + 4Le_t + \bar{\sigma}^2\right)$$

$$= \frac{6\gamma^2L^2\tau_{\max}^2}{W_T}\sum_{t=0}^{T}w_t\Xi_t + \frac{12\gamma^2L\tau_{\max}^2}{W_T}\sum_{t=0}^{T}w_te_t + 3\gamma^2\tau_{\max}(\bar{\sigma}^2 + \bar{\zeta}^2)$$

where $W_T := \sum_{t=0}^{T} w_t$. Rearrage the terms and let $\gamma \leq \frac{1}{10L\tau_{\max}}$ give

$$\frac{1}{W_T} \sum_{t=0}^{T} w_t \Xi_t \leq \frac{1}{1 - 6\gamma^2 L^2 \tau_{\max}^2} \left( \frac{12\gamma^2 L \tau_{\max}^2}{W_T} \sum_{t=0}^{T} w_t e_t + \frac{3\gamma^2 \tau_{\max}^2}{n} (\bar{\sigma}^2 + \bar{\zeta}^2) \right)$$

$$\leq \frac{1}{6LW_T} \sum_{t=0}^{T} w_t e_t + 6\gamma^2 \tau_{\max}^2 (\bar{\sigma}^2 + \bar{\zeta}^2)$$

$\square$

**Theorem XI.** *For convex objective, we have*

$$\frac{1}{T+1} \sum_{t=0}^{T} \left( f(\bar{\mathbf{x}}^{(t)}) - f(\mathbf{x}^\star) \right) \leq 4 \left( \frac{3\bar{\sigma}^2 r_0}{n(T+1)} \right)^{\frac{1}{2}} + 4 \left( \frac{6\tau_{\max} \sqrt{L(\bar{\sigma}^2 + \bar{\zeta}^2)} r_0}{T+1} \right)^{\frac{2}{3}} + \frac{10L(\tau_{\max}+1)r_0}{T+1}.$$

*where $r_0 = \|\mathbf{x}^0 - \mathbf{x}^\star\|^2$.*

**Remark 22.** *For target accuracy $\varepsilon > 0$, then $\frac{1}{T+1} \sum_{t=0}^{T} \left( f(\bar{\mathbf{x}}^{(t)}) - f(\mathbf{x}^\star) \right) < \varepsilon$ after*

$$\mathcal{O} \left( \frac{\bar{\sigma}^2 r_0}{n\varepsilon^2} + \frac{\tau_{\max} \sqrt{L(\bar{\sigma}^2 + \bar{\zeta}^2)} r_0}{\varepsilon^{3/2}} + \frac{10L(\tau_{\max}+1)r_0}{\varepsilon} \right)$$

*iterations. This result is similar to [16, Theorem 2] except that here we replace spectral gap $p$ with the inverse of maximum delay $\frac{1}{\tau_{\max}}$.*

*Proof.* Consider Lemma 20 and multiply both sides with $\frac{w_t}{\gamma}$ and average over time

$$\frac{1}{W_T} \sum_{t=0}^{T} w_t e_t \leq \frac{1}{W_T} \sum_{t=0}^{T} (\frac{w_t}{\gamma} r_t - \frac{w_t}{\gamma} r_{t+1}) + \frac{3L}{W_T} \sum_{t=0}^{T} w_t \Xi_t + \frac{3\gamma}{nW_T} \sum_{t=0}^{T} w_t \bar{\sigma}^2$$

$$\leq \frac{1}{W_T} \sum_{t=0}^{T} (\frac{w_t}{\gamma} r_t - \frac{w_t}{\gamma} r_{t+1}) + \frac{1}{2W_T} \sum_{t=0}^{T} w_t e_t + 18\gamma^2 \tau_{\max}^2 L(\bar{\sigma}^2 + \bar{\zeta}^2) + \frac{3\gamma \bar{\sigma}^2}{n}$$

where the second inequality comes from Lemma 21. Then

$$\frac{1}{2W_T} \sum_{t=0}^{T} w_t e_t \leq \frac{1}{W_T} \sum_{t=0}^{T} (\frac{w_t}{\gamma} r_t - \frac{w_t}{\gamma} r_{t+1} + \frac{3\bar{\sigma}^2}{n} \gamma + 18\tau_{\max}^2 L(\bar{\sigma}^2 + \bar{\zeta}^2)\gamma^2).$$

We can further consider

$$\frac{3L}{W_T} \sum_{t=0}^{T} w_t \Xi_t = \frac{1}{2W_T} \sum_{t=0}^{T} w_t e_t + 18\tau_{\max}^2 L(\bar{\sigma}^2 + \bar{\zeta}^2)\gamma^2$$

$$\leq \frac{1}{W_T} \sum_{t=0}^{T} (\frac{w_t}{\gamma} r_t - \frac{w_t}{\gamma} r_{t+1} + \frac{3\bar{\sigma}^2}{n} \gamma + 36\tau_{\max}^2 L(\bar{\sigma}^2 + \bar{\zeta}^2)\gamma^2) =: \Psi_T.$$

Taking $\{w_t = 1\}_{t \geq 0}$, then

$$\Psi_T \leq \frac{r_0}{\gamma(T+1)} + \frac{3\bar{\sigma}^2}{n} \gamma + 36\tau_{\max}^2 L(\bar{\sigma}^2 + \bar{\zeta}^2)\gamma^2.$$

Apply Lemma 16 we have

$$\Psi_T \leq 2 \left( \frac{3\bar{\sigma}^2 r_0}{n(T+1)} \right)^{\frac{1}{2}} + 2 \left( \frac{6\tau_{\max} \sqrt{L(\bar{\sigma}^2 + \bar{\zeta}^2)} r_0}{T+1} \right)^{\frac{2}{3}} + \frac{dr_0}{T+1}.$$

where $d = \max\{10L, 10L\tau_{\max}\} \leq 10L(\tau_{\max}+1)$ and at the same time

$$\frac{1}{2(T+1)} \sum_{t=0}^{T} e_t \leq 2 \left( \frac{3\bar{\sigma}^2 r_0}{n(T+1)} \right)^{\frac{1}{2}} + 2 \left( \frac{6\tau_{\max} \sqrt{L(\bar{\sigma}^2 + \bar{\zeta}^2)} r_0}{T+1} \right)^{\frac{2}{3}} + \frac{dr_0}{T+1}$$

$$\frac{3L}{T+1} \sum_{t=0}^{T} \Xi_t \leq 2 \left( \frac{3\bar{\sigma}^2 r_0}{n(T+1)} \right)^{\frac{1}{2}} + 2 \left( \frac{6\tau_{\max} \sqrt{L(\bar{\sigma}^2 + \bar{\zeta}^2)} r_0}{T+1} \right)^{\frac{2}{3}} + \frac{dr_0}{T+1}$$

$\square$

Table 14: Comparing RelaySGD/Grad with RelaySGD on Cifar-10 [17] with the VGG-11 architecture. We vary the data heterogeneity $\alpha$ [20] between 16 workers. For low-heterogeneity cases and without momentum, RelaySGD/Grad sometimes performs better than RelaySGD.

| Algorithm | Topology | $\alpha = 1.00$ (most homogeneous) | | $\alpha = 0.1$ | | $\alpha = .01$ (most heterogeneous) | |
|---|---|---|---|---|---|---|---|
| All-reduce (baseline) | fully connected | 87.0% | ⊢⊣→ | 87.0% | ⊢⊣→ | 87.0% | ⊢⊣→ |
| +momentum | | 90.2% | ⊩→ | 90.2% | ⊩→ | 90.2% | ⊩→ |
| RelaySGD | chain | 87.3% | ⊩→ | 87.2% | ⊩→ | 86.5% | ⊢→ |
| +local momentum | | 89.5% | ⊩→ | 89.2% | ⊩→ | 88.4% | ⊩→ |
| RelaySGD/Grad | chain | 88.8% | ⊩→ | 88.5% | ⊩→ | 83.5% | ⊢─H─→ |
| +local momentum | | 86.9% | ⊩→ | 87.8% | ⊩→ | 68.6% | ⊢⊣→ |

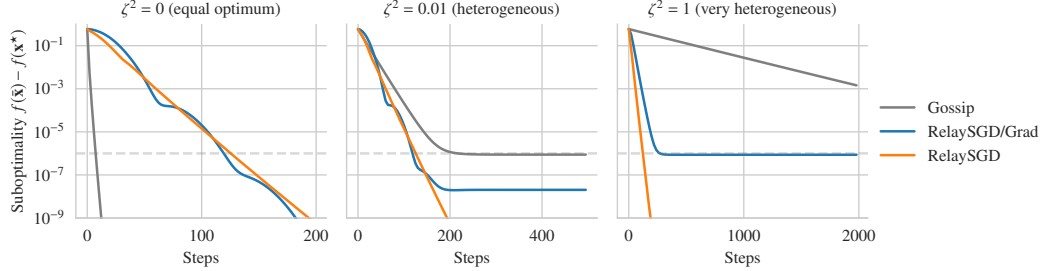

Figure 10: Comparing RelaySGD/Grad against RelaySGD on random quadratics with varying levels of heterogeneity $\zeta^2$, without stochastic noise, on a ring/chain of 32 nodes. Learning rates are tuned to reach suboptimality $\leq 10^{-6}$ as quickly as possible. In contrast to RelaySGD, RelaySGD/Grad with a fixed learning rate does not converge linearly. Compared to DP-SGD (Gossip), RelaySGD/Grad is still less sensitive to data heterogeneity.

## G.2 Empirical analysis of RelaySGD/Grad

In Table 14, we compare RelaySGD/Grad to RelaySGD on deep-learning based image classification on Cifar-10 with VGG-11. Without momentum, and with low levels of heterogeneity, RelaySGD/Grad sometimes outperforms RelaySGD.

Figure 10 illustrates a key difference between RelaySGD/Grad and RelaySGD. While RelaySGD behaves independently of heterogeneity, and converges linearly with a fixed step size, RelaySGD/Grad reaches a plateau based on the learning rate and level of heterogeneity.