# OpenReview forum: "RelaySum for Decentralized Deep Learning on Heterogeneous Data"
_NeurIPS.cc/2021/Conference — NeurIPS 2021 Poster_

### Official Review · Reviewer_ja4W · 2021-06-28

**Rating:** 6
**Confidence:** 4

**Summary:**

A method to reduce communication costs when performing distributed training in networks where communication between nodes requires more than one hop. The trade off is increased memory usage and delayed communication (parameters arrive at a later iteration).

=== Review Update ===

Thank you for the clarifications. In particular, it is true that all virtual topologies used by modern distributed ML systems are sparse, so I retract my comment regarding "comparison on fully-connected" topologies as it certainly does not make sense.

After reading both the authors' response and the other reviewers' comments, I've decided to keep my score as-is.

**Limitations And Societal Impact:**

No issues

**Main Review:**

Originality
- Delayed optimization over a distributed network is not a new idea, see Distributed Delayed Stochastic Optimization (Duchi and Agarwal, 2011). This paper should have been cited and discussed in the related work, though I still lean towards acceptance in spite of this issue.
- The paper's main idea is that messages are aggregated in a way that the memory and network costs are independent of network diameter (i.e. no need to maintain multiple delayed messages "in-flight"). I believe this is an original contribution.

Quality
- Wide coverage across multiple experimental factors: simulation vs real data, multiple competing baselines, various network toplogies, covers both SGD and Adam, and at least one model from CV (VGG, ResNet) and NLP (Bert).
- Interesting and insightful experiments that go beyond mere benchmark-chasing
- Theoretical proofs are sufficient and welcome, though I do not believe there is innovation in the proof techniques

Clarity
- Clear organization of ideas, writing and illustrations are a cut above the typical NeurIPS paper
- Minor issue: the main writing should state clearly that VGG is used for CIFAR10 experiments. This fact is only briefly mentioned in figure captions.

Significance
- Reducing communication in distributed training is a crucial topic, especially when model sizes have grown several orders of magnitude over the last few years
- However, even in Federated Learning, it is not clear to me if applications truly benefit from multiple communication hops. The issue is that topologies are usually "virtual" in peer-to-peer FL applications - it is up to the implementer to decide whether to use a fully connected topology (which does not need delayed communication) or not. Thus, the network topology is not an immutable constraint; it is merely a design choice.
- In connection with the previous point: there is one ablation experiment that is missing from the paper, and that is to compare the method on different non-fully-connected topologies vs the state of the art on a fully-connected one, under realistic assumptions for peer to peer applications. Such an experiment would have made the paper much stronger


**Time Spent Reviewing:**

1.5

---

> ### Author Response · Authors · 2021-08-07
> **Reply to your comments**
>
> Dear Reviewer, thank you for your encouraging feedback and thoughtful comments. We hope to address your concerns below.
>
> __Sparse vs fully-connected topologies__
>
> While sparse communication is indeed often a design choice, there are certainly some real-world settings where sparse topologies are a physical constraint, such as near-field wireless communication, UAVs, and other peer-to-peer networks.
>
> Further, even when the underlying network is fully connected (such as in a datacenter), it is often advantageous to *sparsify* it. The time required to communicate all-to-all in one hop scales linearly with the number of workers, and this does not scale. This is why popular all-reduce protocols (the foundation of current distributed deep learning) always first convert a given fully connected network into a sparse [tree or ring](https://developer.nvidia.com/blog/massively-scale-deep-learning-training-nccl-2-4/).
>
> Given that sparse topologies are already prevalent in real-world deep learning, our work answers the question: “what is the best *efficient, robust, and fast* algorithm for such a setting?”
>
> __Requested speed comparison to all-reduce (ablation)__
>
> All our experiments currently feature all-reduce as a baseline. This baseline implicitly uses multiple communication hops per SGD iteration to uniformly average contributions from all workers over a sparsified topology. In contrast, RelaySGD uses only one communication hop per iteration while still nearly matching the performance of all-reduce.
>
> Whether this reduction in communication hops translates to wall-clock speedups depends heavily on the communication and computation hardware, and we consider this out-of-scope. We do, however, compare our method to SGP (Assran et al. 2019), and in their paper, they empirically show wall-clock benefits over state-of-the-art all-reduce protocols. Since we compare favorably to SGP, those benefits should also hold for RelaySGD.
>
> __Relation to delayed optimization__
>
> Thank you for pointing us to the very relevant reference (Duchi and Agarwal, 2011). Closest to our work is “Locally Averaged Delayed Architecture” in Section 4. However, a key difference between our setups is that, in RelaySGD, every node not only computes gradients but also updates its own model. This decentralized (as opposed to federated or distributed) setting ensures that all nodes are equal and there is no single point of failure such as a master node. However, this also means that our gradients are not only delayed but also computed on models belonging to different nodes. Each such node may potentially drift away from its neighbors and requires careful analysis as we discuss next.
>
> __Proof techniques__
>
> The key challenge in our setting is to ensure that the models across the nodes do not drift away from each other. This is hampered by two factors: 1) the data across the nodes is heterogeneous, and 2) each node locally averages delayed (stale) parameters adding to the instability. Overcoming these is non-trivial and requires novel extensions of existing techniques.
>
> __VGG__
>
> We motivate the choice for an architecture without BatchNorm in line 243, but we should make the use of VGG-11 more explicit. Thank you for pointing this out.

---

### Official Review · Reviewer_zW5E · 2021-07-08

**Rating:** 6
**Confidence:** 4

**Summary:**

This work presents a novel decentralized deep learning algorithm on heterogeneous data. Specifically, the authors develop a RelaySum protocol for handling the information propagation in the networked systems with spanning trees, and RelaySGD for optimizing the learning problems. Different from the popular gossip averaging, RelaySum enables to relay messages through the whole network without decaying each node’s weight at every hop. The authors theoretically prove that RelaySGD is not affected by differences in workers’ data distributions, and that its convergence does not rely on the spectral gap of the graph, but instead on the network diameter. To validate the proposed algorithm, a few baselines and different datasets are used for the comparison. The empirical results show that RelaySGD outperforms baselines in terms of accuracy and convergence speed.

**Limitations And Societal Impact:**

The authors have not adequately addressed the limitations and potential negative societal impact of their work. It would be great for the authors to add one paragraph to briefly discuss.

**Main Review:**

This paper is easy to follow and well written. The investigated topic is particularly critical as decentralized learning has been demanding in the era of big data. However, several major issues came up to my mind when I reviewed the paper carefully.
1.	Issue of topology. The authors mentioned in line 121 that the key insight of this work is that, on tree networks , the RelaySGD update rule can be implemented while using the same communication volume per step as gossip averaging, using additional memory linear in the number of a worker’s direct neighbors. It looks like the RelaySGD has specifically been designed for such a structure of networks. Though the authors in lines 137 and 138 have emphasized that RelaySGD can be used in any communication graph if a corresponding spanning tree is constructed. However, throughout the paper, I have not seen any techniques of converting any communication graph into a spanning tree. Additionally, all experimental results have been shown in terms of spanning trees for RelaySGD. It is difficult to evaluate if RelaySGD really works for other communications graphs.
Further, as the additional memory is linear in the number of a worker’s direct neighbors, it means that for a large-scale network, the computational overhead will increase exponentially?
2.	Unfair comparison in experimental results. As shown in Table 1, the topologies considered for different algorithms are different. In the empirical results, the accuracies were obtained for different algorithms with different topologies, which rendered the unfair comparison. Probably as the authors suggested, some conversion can be done to turn any communication topology into a spanning tree, but the empirical results presented in the paper cannot convince myself that the proposed RelaySGD has superiority over existing methods. For example, in Table 3, RelaySGD w/ momentum seems to perform better than SGP w/ momentum. But it is difficult to tell if the different is just caused by the difference between topologies. Moreover, why for SGP w/ momentum and D^2 w/momentum (or DP-SGD w/ momentum), do they also need different topologies in this work?
3.	Confusing and/or problematic analysis. (a) In the supplementary materials, the authors introduced Assumption H for the bounded noise at the optimum. While the similar property has been defined in Assumption B. Any relation between these two? Why did the authors define such an assumption for the optimum point, which has not typically been seen in the previous works. (b) The complexity for strongly convex case in Theorem looks problematic. As the $\epsilon$ has been used to signify the complexity, why did the authors still include the time $T$ in the complexity, which hasn’t been seen in any previous works? That way, the exact time $T$ required to achieve the $\epsilon$-accuracy is not obtained. (c) In Assumption I, the authors defined $M$ to be 0. Why is this case? I think Assumption I has been quite generic in previous works. That way, could the authors directly define Assumption I at the beginning? Why is the motivation to define two slightly different assumptions in the work? (d) In the proof, a matrix is denoted by a capital bold letter. However, in Lemma 11 and Lemma 12, the authors just used capital letter, which can be confusing. Also, in terms the norm, the authors at the beginning have defined that $||\cdot||$ is spectral norm for a matrix while 2-norm for a vector. It is easy to know that $||\cdot||_F$ represents Frobenius norm. What is $||B||_2$? Also, Lemma 9 – Lemma 12 are trivial and typically the authors can only mention in the proof. There is no need to give formal statements for these. (e) In line 183, the authors mentioned that the dominant term is a function of $\mathcal{O}(1/\epsilon^2)$, while in line 191, the leading term became $\mathcal{O}(1/\epsilon)$. Is it a typo or anything else?
4.	In lines 39 and 40, the authors said that the slow distribution of updates not only slows down training, but also makes decentralized learning sensitive to heterogeneity in workers’ data distributions. Why is this true? The second half may not be true. If each agent in a networked system has quite different data distributions, without effective techniques, even if the distribution of updates is fast, the performance is still poor. Since each agent tries to converge to its own optimal solution, the consensus among them enables a suboptimal solution to the global objective function. Therefore, regardless of the distribution rate of updates, decentralized learning is always sensitive to the heterogeneity.
5.	The dependence of RelaySGD/Grad on the data heterogeneity is a bit surprising to me. Though the table before Algorithm 6 illustrates the difference between RelaySGD and RelaySGD/Grad, it is confusing to me why the authors have introduced the RelaySGD/Grad in the work. It is also not robust to lost messages and workers joining/leaving. What is the motivation here? Are there any scenarios in which RelaySGD/Grad has superiority?
Overall, the paper in the current form still requires substantial changes, particularly for the experimental results and some theoretical complexity analysis. I hope my comments can help the authors address these potential issues.

============Review Update=============

After carefully reading the rebuttal from the authors, and other reviewers' comments, I decided to raise my score. The responses from the authors clarified most of my major comments. Hopefully the authors can include the corresponding changes in the final draft to make it solid.

**Time Spent Reviewing:**

8

---

> ### Author Response · Authors · 2021-08-07
> **Clarifications regarding experimental design and theory**
>
> Dear Reviewer, thank you for your review. The thorough feedback is much appreciated.
>
> __Topologies and spanning trees__
>
> As you mentioned, RelaySGD can only be used on tree topologies. To run RelaySGD on general topologies, we construct a spanning tree of the network using the standard [Spanning Tree Protocol](https://en.wikipedia.org/wiki/Spanning_Tree_Protocol), once before training. We show the efficacy of this protocol combined with RelaySGD in Figure 3. See also line 87 in our paper.
>
> __Fairness of the experiments__
>
> We went out of our way to design fair and meaningful experiments. This was also noted by Reviewer ja4W who mentions our “interesting and insightful experiments that go beyond mere benchmark-chasing”. We motivate our design choices below and we will clarify this in the manuscript, too.
>
> In our deep learning experiments, we ran each algorithm with each compatible topology and found that the algorithms work best on *different topologies*. For example, DP-SGD works better on rings over chains or double binary trees, and SGP works best in time-varying exponential graphs. See Tables 1,2 and 11 of the paper. We summarize these observations below.
>
> |          | Ring                 | Chain (spanning tree of ring)  | Double binary trees                  | Time-varying exp. (Assran et al. 2019) |
> |----------|----------------------|--------------------------------|--------------------------------------|----------------------------------------|
> | DP-SGD   | __Best result__          | Strictly worse than ring       | Worse result than ring (Tables 1, 11) | Unsupported                            |
> | D^2      | __Best result__          | Strictly worse than ring       | Worse result than ring (Tables 1, 11) | Unsupported                            |
> | SGP      | Equivalent to DP-SGD | Equivalent to DP-SGD           | Equivalent to DP-SGD                 | __Best result__                            |
> | RelaySGD | Unsupported          | Worse than double binary trees | __Best result__                          | Unsupported                            |
>
> Thus, when running our experiments (e.g. Table 2 or 3), we used the best topology for each of the algorithms for fairest comparison. Further, we ensured that every algorithm has exactly equal communication (see lines 235, 241). Finally, note that SGP on static symmetric graphs (such as rings or trees) is exactly equivalent to DP-SGD.
>
> > RelaySGD outperforms SGP, but could this difference be caused by the different topologies rather than the algorithms?
>
> We can answer this using the table. RelaySGD on double binary trees > SGP on time-varying graphs > SGP on double-binary trees.
>
> > Why does SGP outperform D² and DP-SGD?
>
> This is indeed because of the time-varying topology. SGP on a ring is equivalent to DP-SGD, and does not perform as well. Despite strong theoretical guarantees, D² empirically suffers from instability ([Lin et al. 2021](https://arxiv.org/abs/2102.04761)), and on the same ring topology, it diverged.
>
>
> __Memory linear in the number of direct neighbors__
>
> Indeed, RelaySGD requires memory proportional to the number of direct neighbors. However, gossip-based decentralized methods are attractive mainly when a network is *sparse* and each worker has only a constant number of neighbors (cf. SGP (Assran et al. 2019)). In such sparse networks, the memory overhead is constant and may be acceptable. Particularly, for double binary trees, the overhead is only 2 extra model copies regardless of the size of the network.
>
> __Correctness of the theoretical analysis__
>
> Thank you for pointing out typos and notational inconsistencies. We want to emphasize that these are easily fixable as we note below. Our conclusions and theorems remain unchanged.
>
> - (a) The key difference between Assumptions B and H is that the weaker Assumption H only works for convex objectives because it only assumes bounded noise at one point (the optimum). Assumption B, on the other hand, assumes bounded noise everywhere and can also be used in the non-convex setting. Assumption H is becoming increasingly popular, see for example (Needell et al., 2016; Bottou et al., 2018; Gower et al., 2019; Koloskova et al., 2020).
> - (b) This is a typo. The exponential term of T should be replaced by a term of $\log(1/\epsilon)$. The proof remains the same as shown in Appendix A.5.
> - (c) We will replace Assumption B with Assumption I to simplify the proof and improve consistency. Similar convergence results can be obtained under both assumptions and both assumptions are common in the literature.
> - (d) We will capitalize the matrices in the two lemmas. The $|| \textbf{B} ||_2$ in the proof refers to the spectral norm $|| \textbf{B} ||$. This is a convention of Schatten p-norm notation. We will make notations more consistent by converting $|| \textbf{B} ||_2$ to $|| \textbf{B} ||$.
> - (e) This is a typo. The leading term should be $\mathcal{O}(\frac{\sigma^2}{n \epsilon^2})$. This term cannot be improved.
>
> __Sensitivity to heterogeneity of worker's data distributions__
>
> You mentioned our statement, "The slow distribution of updates not only slows down training but also makes decentralized learning sensitive to heterogeneity in workers’ data distributions." (lines 39/40)
>
> This statement is indeed confusing and should be clarified. We meant to say that, because of the slow mixing of the updates, the heterogeneous information of the workers is not sufficiently diffused across the workers. Each worker is more influenced by its close neighbors than by workers far away.
>
> __RelaySGD/Grad__
>
> The paper's core is the Relay**Sum** mechanism for uniform averaging of messages from all workers in a sparse network. Relay**SGD** is one way (the best we found) to turn this mechanism into an optimization algorithm. RelaySGD/Grad is another valid algorithm based on the same mechanism. We did not include this algorithm in the main paper but only in the Appendix to avoid confusion. Just like in our experiments on distributed mean estimation with RelaySum (Appendix F), we illustrate that the RelaySum mechanism could be used in more ways than fit in this short paper.
>
> __Limitations and societal impact__
>
> We address additional memory as the main limitation of this method around lines 133 and 147. We do not see strong societal implications since decentralized learning is already an established topic. We would be happy to include or emphasize any specific limitations or societal impact.
>
> __References__
>
> - Assran et al. *Stochastic gradient push for distributed deep learning.* ICML 2019.
> - Needell et al. *Stochastic gradient descent, weighted sampling, and the randomized Kaczmarz algorithm.* Mathematical Programming, 2016.
> - Bottou et al. *Optimization methods for large-scale machine learning.* SIAM Review, 2018
> - Gower et al. *SGD: General analysis and improved rates*. ICML 2019.
> - Koloskova et al. *A unified theory of decentralized SGD with changing topology and local updates.* ICML 2020.

---

### Official Review · Reviewer_D5b5 · 2021-07-16

**Rating:** 6
**Confidence:** 4

**Summary:**

This paper proposes a decentralized SGD method, RelaySGD, for faster mixing in distributed deep learning, with the aim of improving performance on nodes with heterogeneous data sources. Convergence guarantees are provided for smooth non-convex functions under the weak growth condition, as well as for smooth strongly convex functions under the weak growth condition. The method relies on undirected, static, spanning-tree graphs.

**Limitations And Societal Impact:**

No obvious negative societal limitations or impact.

**Main Review:**

The clarity of this paper can be improved, but ultimately, I found this work interesting and sufficiently significant. I do however have some questions about the correctness of the theory and significance of the results that I hope the authors can address in their rebuttal.

- The proposed RelaySGD is equivalent (analytically) to running gossip SGD over an all-to-all graph with fixed delay along each edge based on the number of hops between nodes in your spanning tree. This may result in faster mixing than regular gossip, and the experiments are convincing. However, precisely demonstrating this result would strengthen the paper, for example by constructing an augmented graph (see Chapter 10 of "Convex optimization in signal processing and communications" 2010 by Nedic and Ozdaglar) with the appropriate mixing weights, and comparing the spectral radius of that augmented graph to the spectral radius of a regular doubly-stochastic gossip matrix over various topologies.

On the theory:
- I may be missing something, but the matrix W does not look row-stochastic. Consider \tau=0, i=0, j=1 in a chain. Then [W]_{0, n+1} = 1/n and the rest of the row is equal to 0. Therefore, the rows of W do not sum 1.
Note that for the analysis you can simply apply known results in the literature for gossip with augmented graphs (e.g., see the reference about), hence I am not concerned about the empirical convergence of the method.

Practicality:
- Compared to regular gossip, the method requires each node to store one memory buffer for each neighbor, but has same communication cost, as parcels are summed before communicating. The memory overhead is not great, but as long as the number of neighbours remains small, this should be find and comparable to adaptive gradient methods like Adam.

**Time Spent Reviewing:**

3

---

> ### Author Response · Authors · 2021-08-07
> **Clarifications on theory**
>
> Dear Reviewer, thank you for your thoughtful comments.
>
> Before mainly discussing the paper's *theory* below, let us emphasize that our work represents a major step in making decentralized deep learning practical. Decentralized deep learning with non-iid data is notoriously difficult, and many algorithms with theoretical guarantees have poor practical performance (Lin et al. 2021).
>
> __Analysis using augmented graphs__
>
> Thank you for the pointer. Indeed, our consensus result (Lemma 1) can be derived using the augmented graph framework (Nedić and Ozdaglar, 2010) as well. However, this alone is insufficient to yield convergence rates for RelaySGD. Lemma 1 only shows consensus to some *weighted* average. Using this Lemma to prove optimization of the *unweighted* objective, and deriving concise rates is non-trivial and forms the rest of our theoretical contribution. Further, we show that RelaySGD is unaffected by data heterogeneity (a property not enjoyed by DP-SGD or SGP).
>
> __Row-stochasticity of W__
>
> Let us clarify that $\mathbf{W}$ (line 154) *is* row stochastic with your example of a chain topology of size 2. The temporally unrolled averaging step in Relay looks like
>
> $$\mathbf{W} \mathbf{Y}^{(t)} = \begin{pmatrix} {\color{red}\frac{1}{2}} & 0 & 0 & \frac{1}{2}\\\\  0 & {\color{red}\frac{1}{2}} & \frac{1}{2} & 0 \\\\ {\color{blue} 1} & 0 & 0 & 0 \\\\  0 & {\color{blue} 1} & 0 & 0 \\\\ \end{pmatrix} \begin{pmatrix} \mathbf{x}_0^{(t)} \\\\ \mathbf{x}_1^{(t)} \\\\ \mathbf{x}_0^{(t-1)} \\\\ \mathbf{x}_1^{(t-1)} \\\\ \end{pmatrix}.$$
>
> The top rows of $\mathbf{W}$ correspond to uniform averages of differently delayed contributions from each worker. They contain one entry $\frac{1}{n}$ for each of the $n$ workers (adding up to 1). The self-contributions in red are missing from your description, and we will clarify that $\tau_{ii}=0$ in the text. You could interpret $\mathbf{W}$ as an augmented graph such as in (Nedić and Ozdaglar, 2010).
>
> If this doubt about the correctness of the theoretical contribution impacted your assessment, we would kindly ask you to reconsider your score.
>
> __Memory__
>
> We agree that RelaySGD's additional memory is acceptable only if the number of directly connected neighbors is small. We would like to emphasize that gossip algorithms are, in general, only attractive in this setting. Even for algorithms where memory does not increase with the number of neighbors, communication time does. A worker with many neighbors would form a communication bottleneck in any decentralized algorithm.
>
> __References__
>
> - Nedić and Ozdaglar. *Convergence rate for consensus with delays.* Journal of Global Optimization 2010.
> - Lin et al. *Quasi-Global Momentum: Accelerating Decentralized Deep Learning on Heterogeneous Data*. ICML 2021.

---

> > ### Comment · Reviewer_D5b5 · 2021-09-02
> > **Response to Authors**
> >
> > I have read the author response and the other reviews.
> >
> > There are certainly some limitations to the work. As pointed out in other reviews as well: 1) the novelty is somewhat hampered by the fact that RelaySGD is (analytically) equivalent to delayed gossip over all-to-all graphs, 2) RelaySGD incurs additional memory overhead proportional to the number of neighbours per worker, 3) RelaySGD requires constructing spanning trees, which complicates the training procedure
> >
> > However, in spite of these limitations, I feel that the proposed idea is somewhat creative and would be happy for this idea to be disseminated in the field: in my opinion, the works indirectly asks the question of whether it is best to route messages or perform gossip.
> >
> > I still think a direct spectral analysis of this question on the rate of average consensus would significantly strengthen the work, but still lean on the slightly positive side for this work.

---

### Decision · Program_Chairs · 2021-09-27

**Decision:**

Accept (Poster)

**Comment:**

The paper presents a new communication protocol for decentralized learning when the nodes have significant data heterogeneity. The reviewers all agreed about several of the positives of the current draft: good presentation, extensive experiments, interesting ideas. The main concern was the algorithmic novelty of the scheme (similar to tree-reduce, and gossip with delays), and the memory overhead proportional to the number of neighbors. However, it was agreed during discussions that this work carries enough technical depth to be interesting to the related communities (decentralized/fed learning).